# Distributional Equivalence in Linear Non-Gaussian Latent-Variable Cyclic Causal Models: Characterization and Learning

**Haoyue Dai**[1]    **Immanuel Albrecht**[2]    **Peter Spirtes**[1]    **Kun Zhang**[1,3]

[1]Carnegie Mellon University    [2]FernUniversität in Hagen    [3]MBZUAI

## Abstract

Causal discovery with latent variables is a fundamental task. Yet most existing methods rely on strong structural assumptions, such as enforcing specific indicator patterns for latents or restricting how they can interact with others. We argue that a core obstacle to a general, structural-assumption-free approach is the lack of an equivalence characterization: without knowing *what* can be identified, one generally cannot design methods for *how* to identify it. In this work, we aim to close this gap for linear non-Gaussian models. We establish the graphical criterion for when two graphs with arbitrary latent structure and cycles are *distributionally equivalent*, that is, they induce the same observed distribution set. Key to our approach is a new tool, *edge rank* constraints, which fills a missing piece in the toolbox for latent-variable causal discovery in even broader settings. We further provide a procedure to traverse the whole equivalence class and develop an algorithm to recover models from data up to such equivalence. To our knowledge, this is the first equivalence characterization with latent variables in any parametric setting without structural assumptions, and hence the first structural-assumption-free discovery method. Code and an interactive demo are available at `https://equiv.cc`.

## 1 Introduction

At the core of scientific inquiry lies causal discovery, the task of learning causal relations from observational data (Spirtes et al., 2000; Pearl, 2009). In many real-world scenarios, the variables of interest can be unobserved. For instance, in psychology, personality traits are hidden behind survey responses, and in biology, crucial regulators may be unobserved due to technical inaccessibility. Discovering the causal structure with these latent variables, referred to as latent-variable causal discovery, is essential for understanding and reasoning, yet remains a challenging task.

Latent-variable causal discovery has seen significant development over the past three decades. A milestone was the Fast Causal Inference (FCI) algorithm (Spirtes, 1992), which exploits conditional independence (CI) constraints under hidden confounding. However, FCI is typically not regarded as a method of latent-variable causal discovery, as it focuses solely on causal relations among observed variables, with no intension or capability to identify those among latent variables. In fact, though FCI is already maximally informative under nonparametric CI constraints (Richardson & Spirtes, 2002; Zhang, 2008a), it is still not informative enough for recovering latent structure.

This limitation has motivated the development of many recent approaches that go beyond CI constraints, typically by introducing parametric assumptions, such as linearity (Silva et al., 2003; Dong et al., 2024), non-Gaussianity (Hoyer et al., 2008; Jin et al., 2024), mixture models (Kivva et al., 2021), and distribution shifts (Zhang et al., 2024). Within each setting, a rich array of techniques has emerged. For example, in the linear non-Gaussian setting alone, methods have been developed based on overcomplete independent component analysis (OICA) (Salehkaleybar et al., 2020), regression (Tashiro et al., 2014), Bayesian estimation (Shimizu & Bollen, 2014), independence testing (Xie et al., 2020), cumulants (Robeva & Seby, 2021), independent subspace analysis (Dai et al., 2024), and many more.

However, despite this prosperity, most methods share a clear limitation: they rely on structural assumptions, often about how latent variables are indicated and how they can interact with others. Common examples include measurement models where observed variables have to be pure measure-

ments of latents (Silva & Scheines, 2004; Zhang et al., 2018); hierarchical models that prohibit effects from observed variables (Choi et al., 2011; Huang et al., 2020); sufficient number of pure children per latent (Squires et al., 2022; Jin et al., 2024); and assumptions like triangle- or bow-freeness (Dong et al., 2024; Wang & Drton, 2023). In addition, most methods also assume acyclicity, even though feedback loops are common in real systems. These assumptions, often overly strong and untestable, not only limit applicability but also complicate method selection for practitioners.

A pressing question naturally arises: after decades of progress, is it possible now to have a general structural-assumption-free approach for latent-variable causal discovery that, like FCI, allows arbitrary relations among latent and observed variables, yet goes beyond FCI's limited informativeness?

One core obstacle, we argue, is the lack of a general equivalence characterization with latent variables. Equivalence is a notion fundamental to causal discovery: when different causal models induce the same observed distribution set (known as *distributional equivalence*), no method can, or should, distinguish among them, without extra information like interventions or sparsity constraints. The expected discovery output is thus the entire *equivalence class*, the best one can hope to identify from data. In practice, equivalence can also be defined more coarsely, depending on the specific constraints used. One example is *Markov equivalence*, capturing when models entail the same CI constraints. A well-known and nice result is that in causally sufficient, acyclic, and nonparametric models, Markov equivalence coincides with distributional equivalence (Spirtes et al., 2000); the resulting equivalence class is represented by a completed partially directed acyclic graph (CPDAG).

In the presence of cycles or latent variables, however, equivalence characterization becomes more complex. For example, the nice coincidence between Markov and distributional equivalences breaks down, even with only cycles (Spirtes, 1994; Mooij & Claassen, 2020), or only latent variables (Verma & Pearl, 1991; Richardson et al., 2023), let alone both. The resulting equivalence classes, be it Markov (Richardson & Spirtes, 2002; Claassen & Mooij, 2023) or distributional (Nowzohour et al., 2017; Evans, 2018), also become far more complex. Such complications carry over to parametric settings: for cycles alone, distributional equivalence has been characterized in linear non-Gaussian and Gaussian models (Lacerda et al., 2008; Ghassami et al., 2020); yet for latent variables, no characterization of any kind, whether distributional or constraint specific, is currently known to us. The closest result (Adams et al., 2021) gives conditions for when a linear non-Gaussian acyclic model can be uniquely identified, but leaves open describing the equivalence when such identifiability fails.

All such complications from latent variables and cycles have so far prevented a general equivalence characterization, which is exactly what obstruct progress towards a structural-assumption-free method. The need for such a characterization is yet clear: without knowing *what* can be identified, one generally cannot design methods for *how* to identify it. This is echoed in history: PC algorithm followed CPDAGs; FCI's guarantee followed maximal ancestral graphs (MAGs) (Richardson & Spirtes, 2002).

Our goal in this work is hence to overcome these challenges and establish a general equivalence notion with latent variables and cycles. We focus on linear non-Gaussian models, a parametric setting that has received much attention. Under this setting, we address three questions: **1)** When are two graphs with arbitrary latent variables and cycles equivalent? **2)** How can one traverse the entire equivalence class? **3)** How can one recover latent-variable models up to equivalence from data?

Centered around these three questions, our contributions are summarized as follows:

1. We present a general equivalence notion that allows arbitrary latent structure and cycles in linear non-Gaussian models. This is the first such result known to us in any parametric setting (§2).
2. We introduce a new tool, *edge rank* constraints. It contributes a missing piece to the broader toolbox for latent-variable causal discovery, with potential use across many settings (§3).
3. We characterize equivalence graphically and provide procedures to traverse the entire class. Results are cleaner than expected. We provide an interactive demo at `https://equiv.cc` (§4).
4. We develop an efficient algorithm to recover the equivalence class from data, which is, to our knowledge, the first structural-assumption-free method for latent-variable causal discovery (§5).

## 2 PROBLEM SETUP

In this section, we lay the groundwork for our study. In §2.1, we define the notion of distributional equivalence in linear non-Gaussian latent-variable causal models. Then in §2.2, we introduce the idea of irreducibility to rule out trivial cases, clearing the way for the main results to come.

## 2.1 Preliminaries for Linear non-Gaussian Models

**Notations on matrices.** For a matrix $M$, we let $M_{i,j}$ be its $(i, j)$-th entry. For two index sets $A, B$, we let $M_{A,B} = (M_{a,b})_{a \in A, b \in B}$ be the submatrix of $M$ with rows indexed by $A$ and columns indexed by $B$. We let $M_{A,:}$ be the rows in $M$ indexed by $A$, and similarly $M_{:,B}$ for the columns. For a finite set $A$, we denote its cardinality by $|A|$. We denote by $\mathrm{Scale}(d)$ the set of all $d \times d$ diagonal matrices with nonzero diagonal entries, and by $\mathrm{Perm}(d)$ the set of all $d \times d$ permutation matrices. For a permutation $\pi : V \to V$ on a ground set $V$, we denote $\pi(F) \coloneqq \{\pi(i) : i \in F\}$ for any set $F \subseteq V$, and extend this notation to families of sets by $\pi(\mathcal{F}) \coloneqq \{\pi(F) : F \in \mathcal{F}\}$ for $\mathcal{F} \subset 2^V$.

**Notations on graphs.** Throughout, by a *digraph* we refer to a directed graph that may contain cycles but no self-loops (edges from a vertex to itself). In a digraph $\mathcal{G}$, let $V(\mathcal{G})$ be its vertex set. For vertices $a, b$, we say $a$ is a *parent* of $b$ and $b$ is a *child* of $a$, denoted by $a \in \mathrm{pa}_{\mathcal{G}}(b)$ and $b \in \mathrm{ch}_{\mathcal{G}}(a)$, when $a \to b$ is an edge in $\mathcal{G}$, written $a \to b \in \mathcal{G}$; $a$ is an *ancestor* of $b$ and $b$ is a *descendant* of $a$, denoted by $a \in \mathrm{an}_{\mathcal{G}}(b)$ and $b \in \mathrm{de}_{\mathcal{G}}(a)$, when $a = b$ or there is a directed path $a \to \cdots \to b$ in $\mathcal{G}$. These notations extend to sets: e.g., for a vertex set $A$, $\mathrm{an}_{\mathcal{G}}(A) \coloneqq \bigcup_{a \in A} \mathrm{an}_{\mathcal{G}}(a)$.

**Linear non-Gaussian (LiNG) causal models.** We consider a *linear non-Gaussian model* associated with a digraph $\mathcal{G}$, in which random variables $V = (V_1, \cdots, V_{|V|})^\top$, corresponding to the vertices of $\mathcal{G}$, are generated according to the structural equation:
$$V = BV + E, \tag{1}$$
where $E = (E_1, \cdots, E_{|V|})^\top$ consists of mutually independent, non-constant, non-Gaussian exogenous noise terms. The matrix $B \in \mathcal{B}(\mathcal{G})$ is a weighted *adjacency matrix* (whose entries represent *direct causal effects*) that follows $\mathcal{G}$, where $\mathcal{B}(\mathcal{G})$, all adjacency matrices that *follow* $\mathcal{G}$, is defined as:
$$\mathcal{B}(\mathcal{G}) \coloneqq \{B \in \mathbb{R}^{|V| \times |V|} : B_{V_j, V_i} \neq 0 \implies V_i \to V_j \in \mathcal{G}\}. \tag{2}$$
Assuming $I - B$ is invertible, solving for Equation (1) gives an equivalent mixing form:
$$V = (I - B)^{-1} E =: AE, \tag{3}$$
where $A$ is called the weighted *mixing matrix*. The entry $A_{V_j, V_i}$ represents the *total causal effect* from $V_i$ to $V_j$. All mixing matrices that follow $\mathcal{G}$, denoted by $\mathcal{A}(\mathcal{G})$, is defined as:
$$\mathcal{A}(\mathcal{G}) \coloneqq \{(I - B)^{-1} : B \in \mathcal{B}(\mathcal{G}), \ I - B \text{ invertible}\}. \tag{4}$$

**Latent-variable LiNG models.** Let the vertices $V$ of a digraph $\mathcal{G}$ be partitioned as $V = L \cup X$, where $L$ denotes *latent* (unobserved) variables and $X$ denotes *observed* variables. A *latent-variable model* is specified by the tuple $(\mathcal{G}, X)$, with latent variables $L$ omitted when clear from context.

Given a full mixing matrix $A \in \mathcal{A}(\mathcal{G})$, the submatrix $A_{X,:} \in \mathbb{R}^{|X| \times |V|}$ maps exogenous noise terms to the observed variables. The collection of such wide rectangular mixing matrices is defined as:
$$\mathcal{A}(\mathcal{G}, X) \coloneqq \{A_{X,:} : A \in \mathcal{A}(\mathcal{G})\}. \tag{5}$$
Accordingly, the induced *observed distribution set* of $\mathcal{G}$ on $X$, that is, the set of all distributions over $X$ that can arise from a LiNG model over $(\mathcal{G}, X)$, denoted $\mathcal{P}(\mathcal{G}, X)$, is given by:
$$\mathcal{P}(\mathcal{G}, X) \coloneqq \{p(X) : X = AE, \ A \in \mathcal{A}(\mathcal{G}, X), \ E \in \mathrm{NG}(|V|)\}, \tag{6}$$
where $p(X)$ denotes the probability distribution of the random vector $X$, and $\mathrm{NG}(d)$ denotes the set of all $d$-dim random vectors with mutually independent, non-constant, and non-Gaussian components.

We are now ready to formalize the central notion of this work: distributional equivalence.

**Definition 1 (Distributional equivalence).** Let $\mathcal{G}$ and $\mathcal{H}$ be two digraphs with possibly different vertices, and $X \subseteq V(\mathcal{G}) \cap V(\mathcal{H})$ be the shared observed variables. We say $\mathcal{G}$ and $\mathcal{H}$ are *distributionally equivalent* (or for short, *equivalent*) on $X$, denoted $\mathcal{G} \overset{X}{\sim} \mathcal{H}$, when $\mathcal{P}(\mathcal{G}, X) = \mathcal{P}(\mathcal{H}, X)$.

The equivalence (Definition 1) captures when two models yield identical observed distribution set, i.e., observationally indistinguishable. With this notion in place, next we clean up some trivialities.

## 2.2 Irreducibility: To First Rule Out Trivial Cases of Equivalence

To study identifiability, let us first see what is inherently non-identifiable. For instance, one can freely add latent vertices that are not ancestors of any observed variables $X$ to a digraph $\mathcal{G}$ without affecting $\mathcal{P}(\mathcal{G}, X)$, yielding trivially equivalent models. Identifying those latents is both impossible and meaningless. To rule out such trivialities, we introduce the notion of *irreducibility*.

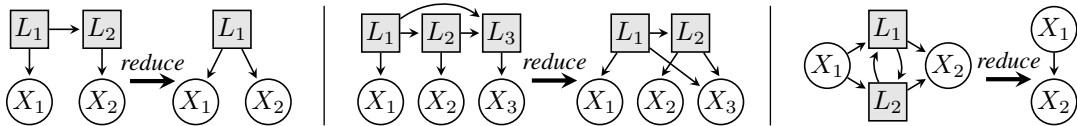

Figure 1: Examples of reducing models to their irreducible forms via the procedure in Proposition 2. Throughout, white circles denote observed variables and grey squares denote latent variables.

**Definition 2 (Irreducibility).** We say a latent-variable model $(\mathcal{G}, X)$ is *irreducible*, when there exists no digraph $\mathcal{H}$ with $|V(\mathcal{H})| < |V(\mathcal{G})|$ such that $\mathcal{G} \overset{X}{\sim} \mathcal{H}$.

Irreducibility captures when an observed distribution set cannot arise from any other model with fewer latent variables. We now present a simple graphical condition for this property.

**Proposition 1 (Graphical condition for irreducibility).** *A model $(\mathcal{G}, X)$ is irreducible, if and only if for each non-empty set $l \subseteq L$, $|\operatorname{ch}_{\mathcal{G}}(l) \setminus l| \geq 2$, i.e., it has more than one child outside.*

Note that when $\mathcal{G}$ is acyclic, it suffices to check each single $L_i \in L$, consistent with the condition previously derived by Salehkaleybar et al. (2020). The proof of Proposition 1, along with others, is provided in Appendix B. The key idea here is that any violation of the condition leads to proportional columns in mixing matrices $\mathcal{A}(\mathcal{G}, X)$, so that the observed distributions can be equivalently generated by a smaller graph with these columns merged to one. Conversely, identifiability results of OICA (Eriksson & Koivunen, 2004) suggest that as long as in the absence of such proportional columns, the mixing matrix is identifiable up to column scaling and permutation, so the number of latents is identifiable.

We next provide an explicit procedure for reducing an arbitrary model to its irreducible form.

**Proposition 2 (Procedure of reduction to the irreducible form).** *Given any latent-variable model $(\mathcal{G}, X)$, the following procedure outputs a digraph $\mathcal{H}$ such that $\mathcal{H} \overset{X}{\sim} \mathcal{G}$ and $(\mathcal{H}, X)$ is irreducible.*

*Step 1. Initialize $\mathcal{H}$ as $\mathcal{G}$.*

*Step 2. Remove vertices $V(\mathcal{H}) \setminus \operatorname{an}_{\mathcal{H}}(X)$ from $\mathcal{H}$, i.e., remove latents who have no effects on $X$.*

*Step 3. Identify the maximal redundant latents in the remaining latent vertices:*

$$\operatorname{mrl} := \{l \subseteq V(\mathcal{H}) \setminus X : |l| > 0, |\operatorname{ch}_{\mathcal{H}}(l) \setminus l| < 2, \text{ and } \forall l' \supsetneq l, |\operatorname{ch}_{\mathcal{H}}(l') \setminus l'| \geq 2\}. \quad (7)$$

*Step 4. For each $l \in \operatorname{mrl}$, let $c$ be the exact child in $\operatorname{ch}_{\mathcal{H}}(l) \setminus l$; for each parent $p \in \operatorname{pa}_{\mathcal{H}}(l) \setminus l \setminus \{c\}$, add an edge $p \to c$ into $\mathcal{H}$ if not already present; finally, remove $l$ vertices from $\mathcal{H}$.*

Illustrative examples of this reduction are shown in Figure 1. This reduction lets us, without loss of generality, restrict attention to irreducible models for the remainder, as arbitrary models are equivalent if and only if their irreducible forms are equivalent. Note that irreducibility is not a structural assumption as discussed in §1, but rather a canonicalization to eliminate trivialities. As a side note, applying the reduction in Proposition 2 does not increase the number of edges or cycles.

## 3 DEVELOPING GRAPHICAL TOOLS FOR CHARACTERIZING EQUIVALENCE

In the previous section, we defined distributional equivalence and irreducibility to rule out trivial unidentifiable cases, so we can focus solely on irreducible models in what follows. Then, when are two irreducible models equivalent? In this section, we tackle this question step by step.

Specifically, in §3.1 we first show that distributional equivalence reduces to an algebraic condition on mixing matrices, and further to a graphical condition involving a concept familiar to the community: *path ranks*, given by max-flow-min-cuts in digraphs. Although familiar, path ranks are difficult to work with due to their global, non-local nature, as we illustrate in §3.2. To overcome this, we introduce a new tool: *edge ranks*, a local, edge-level constraint that complements path ranks and is easier to manipulate. This new tool, developed in §3.3, not only enables our final result to come in the next section, but also enriches the broader rank-based picture beyond our specific setting.

### 3.1 EQUIVALENCE VIA PATH RANKS

We start by examining the algebra behind distributional equivalence. By Definition 2, all equivalent irreducible models must have the same number of latents. This follows from OICA, which guarantees

exact recovery of the number of (nontrivial) latent variables. Hence, in what follows, when considering the equivalence of two irreducible models $(\mathcal{G}, X)$ and $(\mathcal{H}, X)$, we can, without loss of generality, denote their latent variables by a same set of labels, so that $V(\mathcal{G}) = V(\mathcal{H}) = X \cup L$.

We then observe that distributional equivalence can be rephrased in terms of the mixing matrices: two models are equivalent if and only if for every mixing matrix one model can generate, the other can also generate a version of it up to column scaling and permutation, and vice versa, due to the scaling and permutation closedness of exogenous noise terms. Formally,

**Lemma 1** (**Equivalence via mixing matrices closure**). *Two irreducible models are equivalent, written $\mathcal{G} \overset{X}{\sim} \mathcal{H}$, if and only if $\overline{\mathcal{A}(\mathcal{G}, X)} = \overline{\mathcal{A}(\mathcal{H}, X)}$, where for a set of matrices $\mathcal{A} \subseteq \mathbb{R}^{m \times d}$, we let:*

$$\overline{\mathcal{A}} := \{APD : \ A \in \mathcal{A}, \ P \in \mathrm{Perm}(d), \ D \in \mathrm{Scale}(d)\}, \tag{8}$$

*that is, the closure of $\mathcal{A}$ up to column scaling and permutation.*

Then, what are exactly these mixing matrices, namely, $\mathcal{A}(\mathcal{G}, X)$? As defined in Equations (2) to (5), it arises from a mapping over the free parameters in adjacency matrices. Concretely, each entry of the mixing matrix is a rational function: the numerator polynomial reflects "total causal effects" between variables, and the denominator polynomial accounts for "global cycle discounts", which is simply $1$ when the digraph is acyclic. In cyclic cases, there is a small pathological locus where denominators vanish, that is, where $I - B$ becomes singular and cycles "cancel themselves." But as we will show in the proof, this does not affect our results. So for now, let us progress with the Zariski closure of $\mathcal{A}(\mathcal{G}, X)$, an algebraic variety that can be defined by finitely many *equality constraints*.

We now study these constraints. One fundamental class of them is the so-called *rank constraints*, which admits a nice graphical interpretation in terms of *max-flow-min-cut* in digraphs, defined below:

**Definition 3** (**Path ranks**). *In a digraph $\mathcal{G}$, for two sets of vertices $Z, Y \subseteq V(\mathcal{G})$, the path rank $\rho_{\mathcal{G}}(Z, Y)$ is defined as the maximum number of vertex-disjoint directed paths from $Y$ to $Z$ in $\mathcal{G}$. By (Menger, 1927), this max-flow quantity can also be defined by its min-cut version:*

$$\rho_{\mathcal{G}}(Z, Y) := \min_{\boldsymbol{c} \subseteq V(\mathcal{G})} \{|\boldsymbol{c}| : \boldsymbol{c}\text{'s removal from } \mathcal{G} \text{ ensures no directed path from } Y \backslash \boldsymbol{c} \text{ to } Z \backslash \boldsymbol{c}\}. \tag{9}$$

These purely graphical quantities can be read off from the mixing matrices by examining the matrix ranks of corresponding submatrices, which is the well-known (path) rank constraint:

**Lemma 2** (**Path rank constraints in mixing matrices**). *In a digraph $\mathcal{G}$, for any two sets of vertices $Z, Y \subseteq V(\mathcal{G})$ that need not be disjoint, the following equality holds for generic choice of $A \in \mathcal{A}(\mathcal{G})$:*

$$\mathrm{rank}(A_{Z,Y}) = \rho_{\mathcal{G}}(Z, Y). \tag{10}$$

*Here,* rank *denotes the usual matrix rank, and "generic" means the equality holds almost everywhere except for a Lebesgue measure zero set where coincidental lower matrix ranks occur.*

Rank constraints bridge algebra in matrices with geometry in digraphs. They were initially proved for acyclic graphs only (Lindström, 1973; Gessel & Viennot, 1985), and later generalized by (Talaska, 2012). They are powerful: as we will show in the proof, rank constraints alone, together with a column permutation, suffice to determine equivalence. We directly state the result below:

**Lemma 3** (**Equivalence via path ranks**). *Two irreducible models are distributionally equivalent, written $\mathcal{G} \overset{X}{\sim} \mathcal{H}$, if and only if there exists a permutation $\pi$ over the vertices $V(\mathcal{G})$, such that*

$$\rho_{\mathcal{G}}(Z, Y) = \rho_{\mathcal{H}}(Z, \pi(Y)) \quad \text{for all } Z \subseteq X \text{ and } Y \subseteq V(\mathcal{G}). \tag{11}$$

From Lemma 1 to Lemma 3, so far we have arrived at a first purely graphical view of equivalence.

## 3.2 The Complexity of Manipulating Path Ranks

In §3.1 we have arrived at Lemma 3, a purely graphical characterization of equivalence, which, perhaps surprisingly, is expressed in terms of a familiar concept: path ranks. However, this is only a start and far from operational: verifying it requires searching over all vertex permutations and all $(Z, Y)$ pairs, which quickly becomes intractable due to their factorial and exponential growth, let alone the costly graph traversal required for each single path rank computation. As an analogy to the acyclic, causally sufficient case, Lemma 3 is like saying "having all the same d-separations," whereas what we seek is something simpler and more local, like "same adjacencies and v-structures."

Figure 2: An illustration of path ranks, edge ranks, and their duality. Left: a digraph $\mathcal{G}$ with vertices $V$ partitioned to $Y$, $C$, and $Z$, shown by different colors. The path rank $\rho_{\mathcal{G}}(Z, Y) = 2$, with $C$ being a min-cut. Right: the dual edge rank $r_{\mathcal{G}}(V \backslash Y, V \backslash Z) = 4$, given by the maximum bipartite matching from $V \backslash Z$ to $V \backslash Y$, i.e., from $Y \cup C$ to $Z \cup C$, with four matched edges highlighted in red. Four corresponding nonzero entries placed on diagonal, also in red, confirm $\mathrm{mrank}(Q^{(\mathcal{G})}_{V \backslash Y, \, V \backslash Z}) = 4$. One may examine the duality in Theorem 1: w.l.o.g. let $m \leq n$, there is $m - 2 = m + n + 2 - n - 4$.

Then, does a simpler local condition naturally follow from Lemma 3? Unfortunately, not quite. Path ranks are hard to work with due to their global nature: they summarize the size of "bottlenecks", but say nothing about which paths are involved or how they interact. Each single edge may lie on multiple bottlenecks, so even a small local alteration to a digraph may trigger unpredictable global changes in path ranks. Conversely, with latent variables, seemingly very different digraphs can still share the same path ranks. We illustrate such complexity with the following example.

**Example 1** (**Complexity of viewing equivalence via path ranks**). Consider the digraph $\mathcal{G}$ on the left of Figure 2, with vertices partitioned into $Y$, $C$, and $Z$. Obviously, the path rank $\rho_{\mathcal{G}}(Z, Y) = 2$. Now, suppose vertices $\{C_1, C_2\}$ become latent and all others remain observed. What models are equivalent? This is not obvious anymore. It usually takes some thought to realize that adding edges or cycles within $C$, or removing one or two edges from $C$ to $Z$, still preserves path ranks as in Lemma 3. What about the $Y$ to $C$ structure then? This is more subtle: when $n > 2$, it must remain fixed; but when $n = 2$, $C$ is no longer a unique bottleneck, and suddenly, $Y$ can point freely to both $C$ and $Z$.

Things become even less intuitive when other variables are latent. For example, with $m = n = 4$, if $\{C_1, C_2\}$ are latent, there are 17 digraphs in the equivalence class (view them online). When $\{Y_1, Y_2\}$ or $\{Y_1, C_1\}$ are latent, this number comes to $872$ (view) and $1,024$ (view), respectively. Note that all this comes from a well structured digraph; arbitrary structures only lead to greater complexity. $\triangle$

Example 1 illustrates the complexity of path ranks in inferring graph structures. In fact, this complexity is well recognized in literature: despite various techniques developed to estimate path ranks from data (Dai et al., 2022; Sturma et al., 2024), and well-studied counterparts in the linear Gaussian (Sullivant et al., 2010) and discrete settings (Chen et al., 2024b) or even with selection bias (Dai et al., 2025b), when it comes to structure learning from ranks, usually restrictive structural assumptions are required to ensure clean interpretation to where and how these paths can be.

All observations above motivate a question: is there a more local, graph-manipulable alternative to path ranks, not just for building equivalence in this work but also as a piece in the broader toolbox? Interestingly, the answer is yes, and we develop such a tool next: edge ranks.

### 3.3 EDGE RANKS: A NEW TOOL IN THE RANK-BASED PICTURE

We now introduce a new tool: *edge ranks*. As the name suggests, edge ranks directly operate on edges in digraphs, which is more local and accessible in contrast to the paths used in path ranks. For intuition, one may refer to Figure 2, which illustrates all the concepts and results below.

Let us first define edge ranks, similar to how we define path ranks previously in Definition 3:

**Definition 4** (**Edge ranks**). In a digraph $\mathcal{G}$, for two sets of vertices $Z, Y \subseteq V(\mathcal{G})$, the *edge rank* $r_{\mathcal{G}}(Z, Y)$ is defined as the size of the maximum bipartite matching from $Y$ to $Z$ via edges in $\mathcal{G}$, where self-matches ($a$ to $a$ for $a \in Y \cap Z$) are allowed. Edge ranks also admit a min-cut version:

$$r_{\mathcal{G}}(Z, Y) := \min_{\boldsymbol{z} \subseteq Z, \, \boldsymbol{y} \subseteq Y, \, \boldsymbol{z} \cup \boldsymbol{y} \supseteq Z \cap Y} \{|\boldsymbol{z}| + |\boldsymbol{y}| : \text{ there is no edge from } Y \backslash \boldsymbol{y} \text{ to } Z \backslash \boldsymbol{z} \text{ in } \mathcal{G}\}. \quad (12)$$

In parallel to how path ranks correspond to matrix ranks of mixing submatrices (cf. Lemma 2), the pure graphical quantities of edge ranks also have their algebraic counterpart. This time, it is not the mixing matrices at play, but directly the adjacencies.

For clarity, let us introduce a new matrix notation, $Q$, in addition to the already familiar notations of $B$ and $A$, and a new notion of *matching ranks*, in addition to the already familiar matrix ranks.

**Definition 5 (Support matrix).** For a digraph $\mathcal{G}$, its binary *support matrix* in shape $|V(\mathcal{G})| \times |V(\mathcal{G})|$, denoted $Q^{(\mathcal{G})}$, is given by:

$$Q^{(\mathcal{G})}_{V_j, V_i} = \text{`} \times \text{'} \text{ if } V_i = V_j \text{ or } V_i \to V_j \in \mathcal{G}, \text{ and } 0 \text{ otherwise.} \tag{13}$$

**Definition 6 (Matching rank of a matrix).** The *matching rank* of a matrix $M \in \mathbb{K}^{m \times n}$ is given by:

$$\mathrm{mrank}(M) := \max_{P \in \mathrm{Perm}(n)} \sum\nolimits_{i=1,\cdots,\min(m,n)} \mathbb{1}((MP)_{i,i} \neq 0). \tag{14}$$

In simple terms, the matching rank of a matrix, denoted $\mathrm{mrank}$, is the maximum number of nonzero entries that can be positioned on the diagonal by permuting its columns (or rows).

We can now give the edge rank constraints, as a counterpart to path rank constraints (cf. Lemma 2). Unlike the algebraic efforts required there, this result follows immediately from definition:

**Lemma 4 (Edge rank constraints in support matrices).** *In a digraph $\mathcal{G}$, for any two sets of vertices $Z, Y \subseteq V(\mathcal{G})$ that need not be disjoint, the following equality holds:*

$$\mathrm{mrank}(Q^{(\mathcal{G})}_{Z,Y}) = r_{\mathcal{G}}(Z, Y). \tag{15}$$

So far, we have defined both path ranks and edge ranks, which at first glance appear so different: graphically, one is global, focusing on paths, while the other is local, operating on edges; algebraically, one is tied to weighted mixing matrices, the other to binary support matrices. However, despite these apparent differences, a surprising and elegant duality exists between them:

**Theorem 1 (Duality between path ranks and edge ranks).** *In a digraph $\mathcal{G}$ with vertices $V$, for any two sets of vertices $Z, Y \subseteq V$ that need not be disjoint, the following equality holds:*

$$\min(|Z|, |Y|) - \rho_{\mathcal{G}}(Z, Y) = |V| - \max(|Z|, |Y|) - r_{\mathcal{G}}(V \backslash Y, V \backslash Z). \tag{16}$$

This duality is powerful: it suggests that every statement phrased in terms of path ranks and its variants, including the familiar *d-separation* and *t-separation*, can be equivalently rephrased in terms of edge ranks. It reveals that, despite the very different graphical objects involved in the two ranks, they offer complementary perspectives on a same notion in the digraph, namely, *bottleneck*, which captures how dependencies arise in observed data, and is thus central to causal discovery.

In fact, this duality has long been studied in the matroid community (Kőnig, 1931; Perfect, 1968; Ingleton & Piff, 1973), while only the path rank side has been well known in causal discovery. We thus introduce edge ranks here, filling the other side to the rank-based toolbox. It is not that edge ranks are always better, but having both perspectives is beneficial. Within this work, edge ranks indeed lead to simpler derivations. For instance, let us rephrase Lemma 3 using edge ranks below:

**Lemma 5 (Equivalence via edge ranks).** *Two irreducible models are distributionally equivalent, written $\mathcal{G} \overset{X}{\sim} \mathcal{H}$, if and only if there exists a permutation $\pi$ over the vertices $V(\mathcal{G})$, such that*

$$r_{\mathcal{G}}(Z, Y) = r_{\mathcal{H}}(\pi(Z), Y) \quad \text{for all } Z, Y \subseteq V(\mathcal{G}) \text{ with } L \subseteq Y. \tag{17}$$

As we will see in the next section, this formulation paves the way to our final criterion for equivalence. To conclude this section, we provide a side-by-side comparison of two ranks (Table 1; Appendix C.1).

## 4 THE GRAPHICAL CHARACTERIZATION OF DISTRIBUTIONAL EQUIVALENCE

In previous sections, through a step-by-step breakdown of equivalence, we have arrived at a key result, Lemma 5, which, notably, is framed by a new tool we introduced: edge ranks. Building on this foundation, in this section, we provide our final graphical criterion for distributional equivalence, and present a transformational characterization that enables traversal of all digraphs in the equivalence class.

We first study the task of deciding whether two given models are equivalent. For this purpose, although Lemma 5 offers a more local condition for each rank check, it still requires a large number of total checks: one must go through all sets $Y \supseteq L$, which amounts to all subsets $x \subseteq X$. As noted in our earlier analogy (§3.2), this remains akin to "same d-separations," instead of a practical criterion like "same adjacencies and v-structures." Then, does Lemma 5 yield such a practical criterion?

Fortunately, this time, the answer is yes. Unlike the complexities encountered with path ranks in §3.2, edge ranks allow Lemma 5 to admit a nice local decomposition: instead of checking all subsets $x \subseteq X$, it suffices to check each singleton $X_i \in X$ independently. This yields our final graphical criterion:

**Theorem 2** (**Graphical criterion for distributional equivalence**). *In a digraph $\mathcal{G}$, we define the "children bases" of a vertex set $Y \subseteq V(\mathcal{G})$ as vertex sets that admit perfect edge matchings from $Y$:*

$$\mathrm{bases}_\mathcal{G}(Y) \coloneqq \{Z \subseteq \mathrm{ch}_\mathcal{G}(Y) \cup Y : r_\mathcal{G}(Z, Y) = |Z| = |Y|\}. \tag{18}$$

*Then, two irreducible models $(\mathcal{G}, X)$ and $(\mathcal{H}, X)$ are distributionally equivalent, if and only if there exists a permutation $\pi$ over the vertices $V(\mathcal{G})$, such that the following conditions hold:*

$$\begin{cases} \mathrm{bases}_\mathcal{G}(L) = \pi(\mathrm{bases}_\mathcal{H}(L)), & and \\ \mathrm{bases}_\mathcal{G}(L \cup \{X_i\}) = \pi(\mathrm{bases}_\mathcal{H}(L \cup \{X_i\})) & for\ each\ X_i \in X. \end{cases} \tag{19}$$

To interpret this criterion, let us consider the causally sufficient case where $L = \varnothing$. In this case, each $\mathrm{bases}_\mathcal{G}(\{X_i\})$ is just $X_i$ with its children. Then, Theorem 2 immediately reduces to the classical result of exact digraph identification up to permutation (Lacerda et al., 2008). Interestingly, that result has recently been revisited also from a bipartite matching view used here (Sharifian et al., 2025).

Having established Theorem 2 as an efficient criterion for determining equivalence, we now turn to another task of traversing all digraphs in an equivalence class. For this purpose, however, a determining criterion alone offers little guidance. Again, we recall the analogy with Markov equivalence. Note that except for the criterion of "same adjacencies and v-structures," there is an alternative characterization: "two acyclic digraphs are equivalent if and only if one can reach the other via a sequence of *covered edge reversals*," known as "Meek conjecture" (Meek, 1997). Such a transformational characterization offers a natural way for equivalence class traversal. In light of it, we next develop such a transformational characterization, analogous to "Meek conjecture" for our setting.

We start with the permutation part in Theorem 2, which corresponds to row permutations to the support matrix $Q^{(\mathcal{G})}$. Such permutations must result in valid support matrices, i.e., ones with nonzero diagonals. By cycle decomposition of permutations, this leads to an observation: disjoint cycles in the digraph can be freely reversed without affecting equivalence. Formally:

**Lemma 6** (**Admissible cycle reversals**). *For a digraph $\mathcal{G}$, let $\mathcal{C}$ be any collection of vertex-disjoint simple cycles in $\mathcal{G}$. Define a new digraph $\mathcal{H}$ where for each edge $V_i \to V_j \in \mathcal{G}$:*

1. *If $V_i \to V_j$ is on a cycle in $\mathcal{C}$, then include $V_j \to V_i$ in $\mathcal{H}$;*
2. *Otherwise, if $V_j$ is on a cycle in $\mathcal{C}$ with the predecessor $V_k \to V_j$, then include $V_i \to V_k$ in $\mathcal{H}$;*
3. *Otherwise, simply include $V_i \to V_j$ in $\mathcal{H}$.*

*Then, with this new $\mathcal{H}$, the equivalence $\mathcal{G} \overset{X}{\sim} \mathcal{H}$ still holds, for every $X \subseteq V(\mathcal{G})$.*

This result was also shown by (Lacerda et al., 2008). It highlights that in the linear non-Gaussian setting, cycles do not introduce substantial complexity. One may illustrate it using examples in Figure 3.

We then examine a more subtle part in Theorem 2, concerning edge rank equivalence, that is, when all the involved perfect bipartite matchings via edges are unchanged. Intuitively, it is about how edges are structurally "crucial" for maintaining matchings. This leads to the following criterion about edge additions or deletions, corresponding to flipping entries in the support matrix:

**Lemma 7** (**Admissible edge additions/deletions**). *Let $(\mathcal{G}, X)$ be an irreducible model. For any edge $V_i \to V_j$ not currently in $\mathcal{G}$, adding it to $\mathcal{G}$ preserves equivalence on $X$ if and only if:*

$$r_\mathcal{G}(V_i\text{'s nonchildren}\backslash\{V_j\}, L\backslash\{V_i\}) \ < \ r_\mathcal{G}(V_i\text{'s nonchildren}, L\backslash\{V_i\}), \tag{20}$$

*where $V_i$'s nonchildren denotes $V(\mathcal{G})\backslash \mathrm{ch}_\mathcal{G}(V_i)\backslash\{V_i\}$, i.e., zero entries in support column $Q^{(\mathcal{G})}_{:,V_i}$. Conversely, an edge can be deleted if and only if it can be re-added by this criterion.*

In layman's term, Lemma 7 says that an edge $V_i \to V_j$ can be added, only when in the bipartite graph from latents to all vertices currently not $V_i$'s children (including $V_j$), $V_j$ stands as a "pillar" across the maximum matchings; in matroid terms, it is a *coloop*. Then, since $V_j$ is already a "pillar", adding this edge will not be noticed by any $Y$ containing latent variables. Note that both $V_i$ and $V_j$ may be in $X$ or $L$: edges can be added within each or in either direction. Let us examine an example.

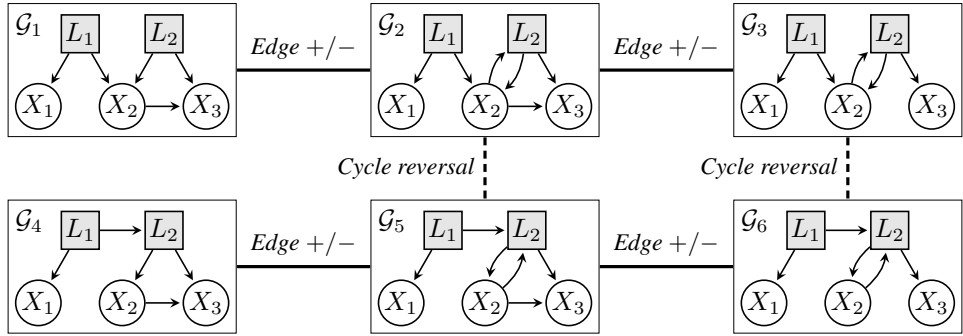

Figure 3: An example distributional equivalence class consisting of 6 digraphs up to $L$-relabeling.

**Example 2** (**Illustrating edge additions via Lemma 7**). We consider the digraph $\mathcal{G}_1$ in Figure 3, and check why the edge $X_2 \to L_2$ can be added. From $L\backslash\{X_2\} = \{L_1, L_2\}$ to $X_2$'s nonchildren $\{L_1, L_2, X_1\}$, there is a full matching of size 2, with $(L_1, L_2)$ matched to either $(L_1, L_2)$ or $(X_1, L_2)$. Since $L_2$ appears in both as a "pillar", adding $X_2 \to L_2$ preserves edge ranks. In contrast, $X_2 \to L_1$ cannot be added, which, for instance, will change $r_{\mathcal{G}_1}(\{L_1, L_2, X_1\}, \{L_1, L_2, X_2\})$ from 2 to 3. $\triangle$

We have introduced two graphical operations that preserve equivalence, namely, cycle reversals and edge additions/deletions. Remarkably, these two operations are not only sufficient but also necessary: together, they fully characterize equivalence. This brings us to our transformational characterization:

**Theorem 3** (**Transformational characterization of the equivalence class**). *Two irreducible models* $(\mathcal{G}, X)$ *and* $(\mathcal{H}, X)$ *are equivalent if and only if* $\mathcal{G}$ *can be transformed into* $\mathcal{H}$, *up to L-relabeling, via a sequence of admissible cycle reversals and edge additions/deletions, as defined in Lemmas 6 and 7.*

*Here, "up to L-relabeling" means there exists a relabeling of L in* $\mathcal{H}$ *yielding a digraph* $\mathcal{H}'$ *such that* $\mathcal{G}$ *reaches* $\mathcal{H}'$ *via the sequence. Moreover, at most one cycle reversal is needed in this sequence.*

Thanks to this transformational characterization, Theorem 3 offers a natural way to traverse an equivalence class by e.g., running BFS or DFS over the space of digraphs connected via admissible operations. Such equivalence class structures are illustrated by Figure 3, Figure 5 (Appendix C.2), and more in our online demo. Note that this traversal can be further accelerated in implementation, by traversing each vertex's children independently in parallel (Lemmas 9 and 12; Appendix B).

Finally, let us return once more to the analogy with Markov equivalence. We have now established counterparts of both "same adjacencies and v-structures" and "Meek conjecture". A natural question is then whether a counterpart of the CPDAG, an informative presentation of the equivalence class, can also be developed. The answer is again yes. We show that within each cycle-reversal configuration, there exists a unique maximal equivalent digraph of which all others are subgraphs. We further provide efficient criteria to construct this maximal digraph, and to determine edges invariant across the equivalence class (similar to arrows in a CPDAG). Due to space limit, this result is presented in Theorem 4 (Appendix C.3). To conclude this section, we provide a side-by-side overview that places our results with their analogues across various classical settings (Table 2; Appendix C.5).

## 5 Algorithm and Evaluation

In this section, we develop a structural-assumption-free algorithm to recover the underlying causal models from observed data up to distributional equivalence. We name this algorithm as general latent-variable Linear Non-Gaussian causal discovery (glvLiNG). Evaluation results are also provided.

**Algorithm.** The glvLiNG pipeline consists of three main steps: it first runs OICA on data to estimate a mixing matrix $\tilde{A}$, then constructs a digraph $\tilde{\mathcal{G}}$ to realize rank patterns in $\tilde{A}$, and finally, starting from $\tilde{\mathcal{G}}$, traverses the equivalence class using the procedure introduced in Theorem 3. Under the assumptions of access to an oracle OICA and faithfulness (no coincidental low ranks in the mixing matrix beyond those structurally entailed; formally stated in Assumption 1 at Appendix A), glvLiNG is guaranteed to recover the entire class of irreducible models equivalent to the ground-truth model.

Proofs and detailed formulations of the glvLiNG algorithm are deferred to Appendix A for page limit. Here, we briefly highlight the core second step: constructing a digraph to realize the observed ranks.

The main challenge lies in this second step, a rank realization task. While the satisfiability nature of this task may suggest brute-force solutions like integer programming, glvLiNG instead offers a more efficient constraint-based approach. Specifically, it proceeds in two phases. Phase 1 recovers edges from latent variables $L$ to all variables $V$, which reduces to a bipartite realization problem known in matroid theory. Phase 2 is more delicate: recover edges from observed variables $X$ to $V$. This may seem combinatorially complex at first glance, since *all* ranks induced by *all* subsets of $X$ must be jointly satisfied (Lemma 3). Fortunately, as we have shown in Theorem 2, these global constraints admit a local decomposition, allowing each single $X_i$'s outgoing edges to be recovered independently. To recover these edges, we give an explicit construction (Lemma 10 in Appendix A) based directly on querying ranks in the OICA mixing matrix, with no need for solving complex constraint systems.

**Evaluation.** We evaluate our approach from five aspects: 1) quantifying the sizes of equivalence classes, 2) assessing glvLiNG's runtime, 3) benchmarking existing methods under oracle inputs, 4) evaluating glvLiNG's performance in simulations, and 5) applying glvLiNG to a real-world dataset.

For 1), we quantify the sizes of equivalence classes, in order to provide an illustrative sense of the uncertainty in latent-variable models. We exhaustively partition digraphs with up to 6 vertices under various latent configurations. For example, there are $1,027,080$ weakly connected digraphs with 5 vertices, of which $26,430$ are acyclic. When the first 2 vertices are latent, $480,640$ of these digraphs yield irreducible models, which finally form 783 equivalence classes. Full statistics are shown in Table 3.

For 2), we assess the efficiency gain enabled by glvLiNG's constraint-based design. We compare the execution time against a linear programming baseline for constructing digraphs to satisfy ranks of oracle OICA mixing matrices. Results confirm substantial speedup: glvLiNG solves cases with $n = 10$ vertices in under 5s, while the baseline takes hours beyond $n = 5$. Full results in Table 4.

For 3), we examine how existing methods behave under structural misspecification by applying them to arbitrary latent-variable models possibly beyond their assumptions. We evaluate LaHiCaSl (Xie et al., 2024) and PO-LiNGAM (Jin et al., 2024), given oracle access to their required tests. Both methods tend to produce overly sparse graphs and misidentify over half of the edges. Full results in Table 5.

For 4), we evaluate glvLiNG with existing methods under finite samples. We simulate data from random irreducible models, varying numbers of observed and latent variables, graph density, and sample size. We observe that glvLiNG performs particularly better than baselines on denser graphs and stays more robust to latent dimensionality, likely due to avoiding model misspecification, while baselines perform better on sparser graphs. Full setup and results are provided in Appendix D.4.

For 5), we apply glvLiNG to a real-world dataset of daily stock returns (Jan 2000-Jun 2005) from 14 major Hong Kong companies spanning banking, real estate, utilities, and commerce. glvLiNG recovers meaningful patterns, such as major banks acting as central causal sources. The two latent variables recovered seem also to admit plausible interpretations. Full results are in Appendix D.5.

**Final remarks.** We conclude with a reflection on the use of OICA in glvLiNG. While one may be concerned about OICA's known inefficiency in practice, we would like to note that the main focus of this work is to characterize distributional equivalence. The glvLiNG algorithm serves more as a proof of concept, showing that such equivalence is indeed recoverable without any structural assumption.

That said, we do see two promising directions for future improvement. 1) For estimation, several existing methods allow partial access to rank information in the mixing matrix without explicitly running OICA. They could be integrated into glvLiNG. 2) For algorithmic efficiency, while glvLiNG already scales well, further pruning is possible. For instance, Theorem 3 implies that ancestral relations among observed variables are identifiable, which may help reduce the search space.

## 6  CONCLUSION AND LIMITATIONS

In this work, we provide a graphical characterization of distributional equivalence for linear non-Gaussian latent-variable models. Based on it, we develop a constraint-based algorithm, glvLiNG, that recovers the underlying model up to equivalence from data without any structural assumptions. Central to our approach is the introduction of edge rank constraints, a new tool in the rank-based picture. One limitation is the use of OICA in glvLiNG, as discussed above. Future directions include developing OICA-free algorithms, and extending new tools to broader settings like linear Gaussian systems.

## ACKNOWLEDGMENT

We would like to acknowledge the support from NSF Award No. 2229881, AI Institute for Societal Decision Making (AI-SDM), the National Institutes of Health (NIH) under Contract R01HL159805, and grants from Quris AI, Florin Court Capital, MBZUAI-WIS Joint Program, and the Al Deira Causal Education project. We also thank the anonymous reviewers for their helpful suggestions.

**Large Language Models Usage:** We used large language models only to aid or polish writing, at the sentence level.

**Ethics Statement:** This paper presents work whose goal is to advance the field of causal discovery. We do not see any ethical or societal concerns that need to be disclosed.

**Reproducibility Statement:** We provide code for our algorithm, glvLiNG, along with an interactive demo for traversing equivalence classes, available at `https://equiv.cc`.

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

# Appendix

## Table of Contents

## A   DETAILS OF THE GLVLING ALGORITHM

### A.1   BASICS OF TRANSVERSAL MATROIDS

Before proceeding, let us introduce some basic concepts from matroid theory that will be used later. Throughout, we define matroids in terms of binary matrices, interpreted as adjacency matrices of bipartite graphs where columns point to rows. The matroid is defined over row indices, corresponding to what is known as a *transversal matroid*. For more, one may refer to (Oxley, 2006).

**Definition 7** (**Basics of transversal matroid**)**.** Let $Q \in \{0,1\}^{m \times n}$ be a binary matrix, interpreted as the adjacency matrix of a bipartite graph where columns $[n]$ point to rows $E := [m]$, where $E$ is called the *ground set*. For simplicity, for each row set $Z \subseteq E$ we denote its rank as:

$$r(Z) := \text{mrank}(Q_{Z,:}), \tag{A.1}$$

though with a slight notation abuse to the letter $r$ we used previously for edge ranks (Definition 4). Here, $\text{mrank}$ is the matching rank we defined in Definition 6. This rank function $r$ turns $E$ into a transversal matroid presented by $Q$. We record the following basic concepts of this matroid, together with some useful properties, written directly in terms of $Q$ and $r$:

**Independent/dependent sets.**

$$\text{Ind}(Q) \;\coloneqq\; \{Z \subseteq E: \; r(Z) = |Z|\},$$
$$\text{Dep}(Q) \;\coloneqq\; 2^E \setminus \text{Ind}(Q). \tag{A.2}$$

*Note.* $Z$ is independent if and only if the rows $Z$ admit a matching into the columns; dependent sets are those with a matching deficiency.

**Bases.**

$$\text{bases}(Q) \;\coloneqq\; \{B \subseteq E: \; B \in \text{Ind}(Q),\, |B| = r(E)\}. \tag{A.3}$$

*Note.* Bases are the maximal independent sets: all its subsets are independent, and all its proper supersets are dependent. Bases are the maximum-cardinality independent sets (all have size $r(E)$). Bases uniquely determine the matroid.

**Circuits.**

$$\text{circuits}(Q) \;\coloneqq\; \{C \subseteq E: \; C \in \text{Dep}(Q) \text{ and } \forall\, C' \subsetneq C,\, C' \in \text{Ind}(Q)\}. \tag{A.4}$$

*Note.* Circuits are the minimal dependent sets: $r(C) = |C| - 1$ and every proper subset is independent. Every dependent set contains a circuit as subset. Circuits do not necessarily have the same cardinalities. Circuits uniquely determine the matroid.

**Cocircuits.**

$$\text{cocircuits}(Q) \;\coloneqq\; \{D \subseteq E: \; r(E \setminus D) = r(E) - 1 \text{ and } \forall\, D' \subsetneq D,\, r(E \setminus D') = r(E)\}. \tag{A.5}$$

*Note.* Cocircuits are the minimal rank-dropping blockers: removing $D$ lowers the full rank (by exactly one), while removing any proper subset does not. Cocircuits meet every basis:

$$|D \cap B| \geq 1, \quad \forall\, D \in \text{cocircuits}(Q),\, B \in \text{bases}(Q). \tag{A.6}$$

Equivalently, $D$ is a minimal set intersecting all bases. By minimal we mean, for any cocircuit $D$, no proper subset of $D$ can be a cocircuit.

When cocircuits meet circuits, the intersection size is never $1$. In particular,

$$|D \cap C| \in \{0, 2, 3, \cdots\}, \quad \forall\, D \in \text{cocircuits}(Q),\, C \in \text{circuits}(Q). \tag{A.7}$$

Equivalently, $D$ is a minimal set not intersecting any circuit with size $1$.

Cocircuits uniquely determine the matroid.

**Coloops.**

$$\text{coloops}(Q) \;\coloneqq\; \{e \in E: \; r(E \setminus \{e\}) = r(E) - 1\}. \tag{A.8}$$

*Note.* A coloop is an element whose presence always increases rank by $1$. A coloop is an element that is in every basis. For each element $e \in E$, the following are equivalent:

$$e \in \text{coloops}(Q) \iff \{e\} \in \text{cocircuits}(Q) \iff e \text{ is in every basis} \iff e \text{ is in no circuit.} \tag{A.9}$$

Coloops *do not* determine the matroid.

**Flats.**

$$\text{flats}(Q) \;\coloneqq\; \{F \subseteq E: \; \forall\, x \in E \backslash F,\, r(F \cup \{x\}) = r(F) + 1\}. \tag{A.10}$$

*Note.* Flats are the $\subseteq$-maximal sets that have a given rank $r(F)$. The family of flats uniquely determines a matroid.

**Fundamental circuit with respect to a basis.** For any basis $B$ and any $e \in E \setminus B$, there is a unique circuit $C_B(e)$ such that

$$e \;\in\; C_B(e) \;\subseteq\; B \cup \{e\}, \tag{A.11}$$

called the fundamental circuit of $e$ w.r.t. $B$. Moreover, for every $f \in C_B(e) \backslash \{e\}$, the set $B \backslash \{f\} \cup \{e\}$ is a basis. Every circuit is a fundamental circuit to some $B$ and $e$.

**Fundamental cocircuit with respect to a basis.** For any basis $B$ and any $e \in B$, there is a unique cocircuit $D_B(e)$ such that

$$e \;\in\; D_B(e) \;\subseteq\; (E \setminus B) \cup \{e\}, \tag{A.12}$$

called the fundamental cocircuit of $e$ w.r.t. $B$. Moreover, for every $f \in D_B(e) \setminus \{e\}$, the set $B \backslash \{e\} \cup \{f\}$ is a basis. Every cocircuit is a fundamental cocircuit to some $B$ and $e$.

Having introduced these basics of transvesal matroids, below we explain our algorithm in detail.

## A.2 Algorithm Overview

Let us first formally define the faithfulness assumption required by the glvLiNG algorithm:

**Assumption 1 (Faithfulness).** Let $d_X := |X|$ be the number of observed variables. We assume that in the true mixing matrix $A_{X,:}$ that generates data (as defined in Equation (5)), all $d_X \times d_X$ and $(d_X - 1) \times (d_X - 1)$ minors exhibit matrix ranks consistent with the corresponding path ranks entailed by the true causal graph (as characterized in Lemma 2).

In other words, there is no coincidental parameter cancellation in the data generating process that would lead to matrix ranks lower than those structurally entailed by the graph. Note that such faithfulness assumption, often also referred to as the genericity assumption, is standard in the literature (Adams et al., 2021). It holds almost everywhere in the parameter space except for a Lebesgue measure zero set where coincidental lower ranks occur.

We next elaborate on our glvLiNG algorithm. The core to the glvLiNG algorithm is to query matrix ranks from the mixing matrix estimated from data using overcomplete ICA (OICA), and then construct a binary support matrix (corresponding to a digraph) that satisfies these matrix ranks.

Let $p(X)$ be a data distribution generically generated by an unknown latent-variable model $(\mathcal{G}, X)$, that is, $p(X) \in \mathcal{P}(\mathcal{G}, X)$. Without loss of generality we assume that $(\mathcal{G}, X)$ is irreducible. Let $\tilde{A} \in \mathbb{R}^{|X| \times |V|}$ be a mixing matrix estimated on $p(X)$ by OICA, and for now, we index the rows and columns of $\tilde{A}$ by $X$ and $V$, respectively. By the identifiability of OICA, $\tilde{A}$ is the true mixing matrix up to column permutation and scaling. Further, with the duality between path ranks and edge ranks (Theorem 1), we have that, there exists a permutation $\pi$ of $V$, such that for all $Z \subseteq X$ and $Y \subseteq V$, the following equality holds:

$$\text{rank}(\tilde{A}_{Z,Y}) = \rho_{\mathcal{G}}(Z, \pi(Y)) = |Z| + |Y| - |V| + r_{\mathcal{G}}(V \backslash \pi(Y), V \backslash Z). \tag{A.13}$$

In other words, there exists an unknown binary matrix $Q \in \{0, 1\}^{|V| \times |V|}$, whose matching ranks $\text{mrank}(Q_{Z,Y})$ can be queried for any $Z, Y \subseteq V$ with $L \subseteq Y$, despite its exact entry values being unknown. Such a matrix must exist, with one specific matrix, $Q^{(\mathcal{G})}$ with rows permuted by $\pi$, being an example. As long as one can recover this matrix $Q$, one can then permute its rows to place nonzero entries on the diagonal, and the resulting matrix must exactly be some support matrix for a digraph $\mathcal{H}$ with $\mathcal{G} \overset{X}{\sim} \mathcal{H}$, by Lemma 5. Such a row permutation to have nonzero diagonals must also exists, as $\text{mrank}(Q) = |V|$, by setting $Z, Y$ both to $\varnothing$ in Equation (A.13).

Our problem then reduces to: how can one recover such a $Q$ matrix from rank queries? We express this problem in a more general formulation, as follows:

---

**Key Problem: Matrix Recovery to Satisfy Matching Rank Queries**

Let $Q \in \{0, 1\}^{m \times n}$ be a binary matrix whose entries are unknown. Let the columns be partitioned as $[n] = L \cup X$, where $L$ is a fixed subset. For any $Z \subseteq [m]$ and any $Y \subseteq [n]$ satisfying $L \subseteq Y$, one may query the matching rank of the submatrix $Q_{Z,Y}$, which, for simplicity, is denoted as an oracle function:

$$r(Z, Y) := \text{mrank}(Q_{Z,Y}). \tag{A.14}$$

**The task is:** using only access to this oracle, construct a binary matrix $H \in \{0, 1\}^{m \times n}$ such that for all $Z \subseteq [m]$ and $Y \subseteq [n]$ with $L \subseteq Y$, the following condition holds:

$$\text{mrank}(H_{Z,Y}) = r(Z, Y). \tag{A.15}$$

That is, $H$ is required to satisfy the matching rank oracle on all valid $(Z, Y)$ pairs.

---

Clearly, the key problem posed above is essentially a satisfiability problem, and can be solved via brute-force methods such as linear programming. However, in what follows, we present a significantly more efficient and structured procedure. Note that the matching rank queries in the problem are equivalent to providing the transversal matroids on each submatrix $Q_{:,Y}$. Thus, for convenience, throughout the rest of this appendix, we may freely use the matroid language introduced in Definition 7.

**Overview of our procedure:** Our reconstruction procedure consists of two phases:

Phase 1: Impute the columns in $H$ indexed by $L$ to satisfy the matroid $\mathrm{bases}(Q_{:,L})$. Equivalently, this is to construct a bipartite graph that realizes a given transversal matroid.

Phase 2: Impute the remaining columns indexed by $X$ such that all matroids induced by $Q_{:,L\cup\{x\}}$ for $x \subseteq X$ are satisfied. Although this may appear combinatorially complex at first glance, we show that each singleton column in $X$ can in fact be imputed independently.

## A.3 PHASE 1: BIPARTITE GRAPH REALIZATION

Let us first formulate the problem of Phase 1:

---

**Problem of Phase 1: Bipartite Graph Realization of Transversal Matroids**

Let $Q \in \{0,1\}^{m\times l}$ be a binary matrix with unknown entries, but known matroid $\mathrm{bases}(Q)$.
**The task is:** Construct a binary matrix $H \in \{0,1\}^{m\times l}$ such that:
$$\mathrm{bases}(H) = \mathrm{bases}(Q). \tag{A.16}$$

That is, construct an example bipartite graph, represented by $H$, to realize a transversal matroid, given by $\mathrm{bases}(Q)$.

---

By duality, this problem is equivalent to reconstructing the digraph representation of the strict gammoid that is the dual of the transversal matroid based on the seminal paper by (Mason, 1972) and dualizing the result using the Fundamental Lemma by (Ingleton & Piff, 1973).

The bases of the dual matroid $Q^*$ are given by $\mathrm{bases}(Q^*) = \{E\backslash B \mid B \in \mathrm{bases}(Q)\}$. The $\alpha$-system for $Q^*$ is defined as the bipartite graph with the following incidence relation

$$I_{Q^*} = \{(e, (F,i)) \in E \times (\mathrm{flats}(Q^*), \mathbb{N}) \mid F \in \mathrm{flats}(Q^*),\, e \in F,\, i \in \mathbb{N},\, 1 \le i \le \alpha_{Q^*}(F)\}, \tag{A.17}$$

where

$$\alpha_{Q^*}(F) = |F| - r_{Q^*}(F) - \bigcup_{G\in\mathrm{flats}(Q^*),\, G\subsetneq F} \alpha_{Q^*}(G). \tag{A.18}$$

Since $Q$ is a transversal matroid, $Q^*$ is a strict gammoid, and therefore all $\alpha_{Q^*}(F) \ge 0$, $\sum_{F\in\mathrm{flats}(Q^*)} \alpha_{Q^*}(F) = |E| - r_{Q^*}(E) = r_Q(E)$, and the $\alpha$-system for $Q^*$ has a maximal matching that covers all $(F,i)$, and each such matching has the property that the set of unmatched elements from $E$ forms a basis $T$ of $Q^*$.

Now fix such a maximal matching, let $T \subseteq$ be the unmatched basis, and let $(F_e, i_e)$ be the vertex that is matched to $e$ for all $e \in E\backslash T$. Define the digraph $D = (V, A)$ with $V = E$ and $A = \{(u,v) \in E \times E \mid u \notin T,\, v \in F_u,\, v \ne u\}$. The digraph $D$ represents the strict gammoid $Q^*$ (Mason, 1972). Using the fundamental lemma (Ingleton & Piff, 1973), we obtain that

$$H = \{(e,t) \in E \times (E\backslash T) \mid e \in F_t\} \tag{A.19}$$

represents the transversal matroid $Q$.

## A.4 PHASE 2: AUGMENTING A BIPARTITE GRAPH FOR MATROID EXTENSIONS

Having imputed the values in $H_{:,L}$, we then impute the remaining $X$ columns:

---

**Problem of Phase 2: Augmenting a Bipartite Graph to Realize Matroid Extensions**

Let $Q \in \{0,1\}^{m \times n}$ be a binary matrix with unknown entries, and columns partitioned as $[n] = L \cup X$ for a fixed $L$. For every subset $\boldsymbol{x} \subseteq X$, the matroid $\mathrm{bases}(Q_{:,L \cup \boldsymbol{x}})$ is known.

Let $H \in \{0,1\}^{m \times n}$ be a partially imputed matrix such that $\mathrm{bases}(H_{:,L}) = \mathrm{bases}(Q_{:,L})$ is already satisfied, while the columns indexed by $X$ remain unassigned.

**The task is:** Fill in the remaining columns of $H$ such that for all $\boldsymbol{x} \subseteq X$,

$$\mathrm{bases}(H_{:,L \cup \boldsymbol{x}}) = \mathrm{bases}(Q_{:,L \cup \boldsymbol{x}}). \tag{A.20}$$

That is, augment the bipartite graph, represented by $H_{:,L}$, by adding more "sources" $X$, so that matroids are realized for all subsets of $X$ extended with $L$.

---

One may first question whether such an imputation is possible, since overall the already assigned $H_{:,L}$ only realizes the matroid equality $\mathrm{bases}(H_{:,L}) = \mathrm{bases}(Q_{:,L})$, but the exact entry values recovery $H_{:,L} = Q_{:,L}$ is not guaranteed (and also impossible).

We show that such an imputation is indeed possible, via the following result:

**Lemma 8** (**How the transversal matroid changes when augmenting more sources**). *For two binary matrices $Q_1 \in \{0,1\}^{m \times n_1}$ and $Q_2 \in \{0,1\}^{m \times n_2}$, we denote by $[Q_1|Q_2] \in \{0,1\}^{m \times (n_1 + n_2)}$ the matrix obtained by horizontally concatenating $Q_1$ with $Q_2$. Then, we have:*

$$\mathrm{Ind}([Q_1|Q_2]) \;=\; \{Z_1 \cup Z_2 : Z_1 \in \mathrm{Ind}(Q_1),\ Z_2 \in \mathrm{Ind}(Q_2)\}. \tag{A.21}$$

*In other words, two matrices' matroids sufficiently determine the matroid of their augmentation.*

With Lemma 8, every $\mathrm{bases}(H_{:,L \cup \boldsymbol{x}})$ equals $\mathrm{bases}([Q_{:,L}|H_{:,\boldsymbol{x}}])$, and thus the imputation is possible.

Then, how to solve for this imputation? At the first glance, one may have concern on the complexity: in contrast to solving Phase 1's realization problem for only one matroid induced by $L$, now we need to realize combinatorially many matroids induced by $L \cup \boldsymbol{x}$ for all $\boldsymbol{x} \subseteq X$. When trying to impute a single column $H_{:,X_i}$, this column can appear in many subsets $\boldsymbol{x} \ni X_i$.

Interestingly, all these subsets can be disentangled: one do not need to explicitly realize each $\boldsymbol{x} \subseteq X$. Instead, it suffices to just realize each singleton augmentation for $X_i \in X$ independently.

We show this by the following result:

**Lemma 9** (**Reducing all union equivalence checks to singleton checks**). *Let $Q, H \in \{0,1\}^{m \times n}$ be two binary matrices with columns partitioned as $[n] = L \cup X$ for a fixed $L$. Then, the condition*

$$\mathrm{bases}(Q_{:,L \cup \boldsymbol{x}}) = \mathrm{bases}(H_{:,L \cup \boldsymbol{x}}), \quad \forall \boldsymbol{x} \subseteq X, \tag{A.22}$$

*holds, if and only if the condition*

$$\begin{cases} \mathrm{bases}(Q_{:,L}) = \mathrm{bases}(H_{:,L}), \ and \\ \mathrm{bases}(Q_{:,L \cup \{X_i\}}) = \mathrm{bases}(H_{:,L \cup \{X_i\}}), \quad \forall X_i \in X, \end{cases} \tag{A.23}$$

*holds.*

With Lemma 9, the remaining problem of Phase 2 can be reduced to solving for each singleton augmentation, formulated as follows:

> **Problem of Phase 2 (Reduced): Augmenting a Bipartite Graph with a Singleton Source**
>
> Let $H \in \{0,1\}^{m \times (l+1)}$ be a partially known binary matrix whose columns are partitioned as $[l+1] = L \cup \{x\}$ for a fixed $L$. The two matroids $\mathrm{bases}(H)$ and $\mathrm{bases}(H_{:,L})$ are known, while the exact values at the column $H_{:,x}$ remain unknown.
>
> **The task is:** Fill in the column $H_{:,x}$ to satisfy the given matroid $\mathrm{bases}(H)$.
>
> That is, augment the bipartite graph, represented by $H_{:,L}$, by adding one extra "source" $x$, so that it extends the current transversal matroid correctly.

To solve this singleton augmentation problem, one could at worst exhaustively try all $2^m$ possible fillings. In what follows, however, we present a more efficient, deterministic construction:

**Lemma 10** (**Constructing a particular solution for singleton augmentation**). *Let $H \in \{0,1\}^{m \times (l+1)}$ be a binary matrix with columns partitioned as $[l+1] = L \cup \{x\}$. Define*

$$D := \{i \in \{1, \ldots, m\} \mid \forall C \in \mathrm{circuits}(H) : i \in C \Rightarrow C \backslash \{i\} \notin \mathrm{Ind}(H_{:,L})\}. \tag{A.24}$$

*Define a new matrix $H'$ where $H'_{:,L} = H_{:,L}$ and the column $x$ is replaced by $H'_{i,x} = 1$ if $i \in D$ and $0$ otherwise. Then, the whole matroid remains unchanged after this column $x$ replacement:*

$$\mathrm{bases}(H') = \mathrm{bases}(H). \tag{A.25}$$

With this result, we complete the final step of the Key Problem defined above and can obtain a binary matrix $H$ that satisfies all rank constraints.

Proofs of Lemmas 8 to 10 are all given in Appendix B.

To interpret $H$ as a digraph representation, we perform a final row permutation to place nonzeros along the diagonal. Standard algorithms such as the $n$-rooks method may be used for the row permutation. We keep the column indices of $L \cup X$ fixed and reindex the rows to match the same ordered list $L \cup X$. The resulting matrix encodes a model distributionally equivalent to the underlying model that general the data. One may then run BFS/DFS using Theorem 3 to obtain the whole equivalence class.

With all above, we conclude the algorithm part.

## B  PROOFS OF MAIN RESULTS

Note that we present the proofs in an order that differs slightly from their appearance, arranged instead according to their logical dependencies.

### B.1  PROOFS OF IRREDUCIBILITY RESULTS

The irreducibility results rely on the identifiability of (overcomplete) independent component analysis (ICA). So we first restate them here.

A linear irreducible ICA model can be described by the equation

$$X = AE, \tag{B.1}$$

where $E = (E_1, \cdots, E_m)^\top$ are unknown mutually independent random variables, namely *sources*, and $X = (X_1, \cdots, X_p)^\top$ are observed random variables, namely *mixtures*. $A \in \mathbb{R}^{p \times m}$, namely the *mixing matrix*, is constrained to have no pairwise proportional columns (including zero columns). The tuple $(A, E)$ is called an *irreducible ICA representation* of $X$.

**Lemma 11** (Identifiability of ICA; (Eriksson & Koivunen, 2004))**.** *Let $(A, E)$ and $(B, S)$ be two irreducible ICA representations of a $p$-dim random vector $X$, where $A \in \mathbb{R}^{p \times m}$ and $B \in \mathbb{R}^{p \times n}$. If every component of $E$ follows a non-Gaussian distribution, then the following properties hold:*

    *1. $m = n$.*

2. *Every column of $A$ is proportional to some column of $B$, and vice versa.*

3. *Every component of $S$ follows a non-Gaussian distribution*[1].

**Proposition 1** (**Graphical condition for irreducibility**). *A model $(\mathcal{G}, X)$ is irreducible, if and only if for each non-empty set $l \subseteq L$, $|\operatorname{ch}_{\mathcal{G}}(l) \setminus l| \geq 2$, i.e., it has more than one child outside.*

*Proof of Proposition 1.* Due to the identifiability of OICA, irreducibility is equivalent to that there are no proportional columns in the mixing matrix, which, with Lemma 2, is that

$$\rho_{\mathcal{G}}(X, v) \geq 2, \quad \forall v \subseteq V \text{ with } |v| \geq 2. \tag{B.2}$$

When $v$ contains 2 or more vertices from observed $X$, this condition is naturally satisfied. So we only need to consider $v$ that contains at most one vertex from $X$ and at least one vertex from $L$.

When $v$ contains only $L$ vertices, the violation of Equation (B.2) leads to the graphical criterion. When $v$ contains one $X$ vertex, say, $X_i$, and Equation (B.2) is violated, it means the min-cut from $v$ to $X$ is simply $\{X_i\}$, which implies that the min-cut from the remaining latent vertices $v \setminus \{X_i\}$ to $X$ is either $\varnothing$ or $\{X_i\}$. This also leads to the graphical criterion.

□

**Proposition 2** (**Procedure of reduction to the irreducible form**). *Given any latent-variable model $(\mathcal{G}, X)$, the following procedure outputs a digraph $\mathcal{H}$ such that $\mathcal{H} \overset{X}{\sim} \mathcal{G}$ and $(\mathcal{H}, X)$ is irreducible.*

*Step 1. Initialize $\mathcal{H}$ as $\mathcal{G}$.*

*Step 2. Remove vertices $V(\mathcal{H}) \setminus \operatorname{an}_{\mathcal{H}}(X)$ from $\mathcal{H}$, i.e., remove latents who have no effects on $X$.*

*Step 3. Identify the maximal redundant latents in the remaining latent vertices:*

$$\operatorname{mrl} := \{l \subseteq V(\mathcal{H}) \setminus X : |l| > 0, \ |\operatorname{ch}_{\mathcal{H}}(l) \setminus l| < 2, \text{ and } \forall l' \supsetneq l, \ |\operatorname{ch}_{\mathcal{H}}(l') \setminus l'| \geq 2\}. \tag{7}$$

*Step 4. For each $l \in \operatorname{mrl}$, let $c$ be the exact child in $\operatorname{ch}_{\mathcal{H}}(l) \setminus l$; for each parent $p \in \operatorname{pa}_{\mathcal{H}}(l) \setminus l \setminus \{c\}$, add an edge $p \to c$ into $\mathcal{H}$ if not already present; finally, remove $l$ vertices from $\mathcal{H}$.*

*Proof of Proposition 2.* This graphical operation directly translates the operation to merge all maximally proportional columns in the mixing matrix into single columns. This ensures the irreducible $\mathcal{H}$.

Note that by removing maximally redundant latents, the added edges in step 4 will not be removed later, i.e., for each $l$ operated in step 4,

$$|\operatorname{ch}_{\mathcal{H}}(l) \setminus l| = 1, \text{ and } (\operatorname{ch}_{\mathcal{H}}(l) \cup \operatorname{pa}_{\mathcal{H}}(l) \setminus l) \cap (\cup \operatorname{mrl}) = \varnothing. \tag{B.3}$$

This ensures the well-defined graphical operation.

□

### B.2 Proof from Rank Equivalence to Distributional Equivalence

**Lemma 3** (**Equivalence via path ranks**). *Two irreducible models are distributionally equivalent, written $\mathcal{G} \overset{X}{\sim} \mathcal{H}$, if and only if there exists a permutation $\pi$ over the vertices $V(\mathcal{G})$, such that*

$$\rho_{\mathcal{G}}(Z, Y) = \rho_{\mathcal{H}}(Z, \pi(Y)) \quad \text{for all } Z \subseteq X \text{ and } Y \subseteq V(\mathcal{G}). \tag{11}$$

---

[1]But note that unlike 2., $S$ may still be unreachable from $E$ via only permutation and scaling. In other words, the model is *identifiable*, but not *unique*. See Example 2 of Eriksson & Koivunen (2004).

*Proof of Lemma 3.* We first show the "⇒" direction.

By Lemma 2 there is a generic choice $A \in \mathcal{A}(\mathcal{G})$ that realizes the rank structure for all $Z, Y \subseteq V(\mathcal{G})$: $\mathrm{rank}(A_{Z,Y}) = \rho_{\mathcal{G}}(Z,Y)$. With Lemma 1 we obtain from $\mathcal{G} \overset{X}{\sim} \mathcal{H}$ with $A \in \overline{\mathcal{A}(\mathcal{G},X)} = \overline{\mathcal{A}(\mathcal{H},X)}$ that there is a matrix $B \in \mathcal{A}(\mathcal{H},X)$, a permutation matrix $P$, and a scaling matrix $D$ such that $A_{X,:} = BPD$. Let $\pi$ be the column-permutation that corresponds to the matrix $P$.

Now, let $Z \subseteq X$ and $Y \subseteq V(\mathcal{G}) = V(\mathcal{H})$, then we have the desired equation

$$\rho_{\mathcal{G}}(Z,Y) = \mathrm{rank}(A_{Z,Y}) = \mathrm{rank}((BPD)_{Z,Y}) = \mathrm{rank}((BP)_{Z,Y}) \tag{B.4}$$

$$= \mathrm{rank}(B_{Z,\pi(Y)}) = \rho_{\mathcal{H}}(Z, \pi(Y)). \tag{B.5}$$

We then show the "⇐" direction.

For any $Z \subseteq X$, the partial application $\rho_{\mathcal{G}}(Z, \_)$ is the rank function of a strict gammoid $M_{\mathcal{G},Z} = \Gamma(\mathcal{G}, Z, V(\mathcal{G}))$ defined by the digraph $\mathcal{G}$ on the ground set $V(\mathcal{G})$ with the set of terminals $Z$. Analogously, $\rho_{\mathcal{H}}(Z, \_)$ is the rank function a strict gammoid $M_{\mathcal{H},Z}$ defined by the digraph $\mathcal{H}$ on the same ground set and the same set of terminals.

Equation (11) implies that $\pi$ is an isomorphism between $M_{\mathcal{G},Z}$ and $M_{\mathcal{H},Z}$, for all $Z \subseteq X$ simultaneously. Clearly, $\pi$ is also an isomorphism between the dual matroids $M_{\mathcal{G},Z}^*$ and $M_{\mathcal{H},Z}^*$.

It follows from the Fundamental Lemma in (Ingleton & Piff, 1973) that the support matrices $Q^{(\mathcal{G})}$ and $Q^{(\mathcal{H})}$ define isomorphic families of transversal matroids that are represented by the corresponding row-sub-matrices of all $(I - B_{\mathcal{G}})$ with sufficiently general weights and $B_{\mathcal{G}} \in \mathcal{B}(\mathcal{G})$; and $(I - B_{\mathcal{H}})$ with sufficiently general weights and $B_{\mathcal{H}} \in \mathcal{B}(\mathcal{H})$, respectively.

In (Brylawski, 1975) it is shown that every transversal matroid may be represented by vectors that lie in the faces of a simplex such that for every minimal non-trivial combination of the zero vector by a set of representing vectors, this set lies on a common simplex face with rank one less than the cardinality of the set of vectors. Over $\mathbb{R}$ such vectors can be found almost surely by taking the incidence matrix of the transversal system and choosing a random value for each nonzero entry. All matrices over $\mathbb{R}$ that represent the transversal matroid can be produced by this procedure.

Choosing random values for the nonzero entries of $Q^{(\mathcal{G})}$ (and $Q^{(\mathcal{H})}$) gives almost surely a matrix that represents the respective family of transversal matroids in such a general simplex position, with nonzero entries on the diagonal. By row scaling, this matrix can be brought into the desired form $(I - B)$ where all diagonal entries are equal to 1. Scaling the columns and rows of a matrix by nonzero factors does not alter the family of matroids represented by a matrix, and does not change whether a matrix is in general simplex position.

The matrices in $A \in \mathcal{A}(\mathcal{G},X)$ (and $\mathcal{A}(\mathcal{H},X)$) are row-restrictions of inverses of diagonal-1-scaled versions of matrices in general simplex position;

$$A = (SR)_{X,:}^{-1} = R_{X,:}^{-1} S^{-1}, \tag{B.6}$$

where $R$ is a randomized valuation of $Q^{(\mathcal{G})}$, and $S$ is a diagonal matrix consisting of the multiplicative inverses of the diagonal entries of $R$.

Let $P$ be the permutation matrix for $\pi$, and let $T$ be the diagonal matrix consisting of the multiplicative inverses of the diagonal entries of $PR$. Then there is

$$A' = (TPR)_{X,:}^{-1} \in \mathcal{A}(\mathcal{H},X). \tag{B.7}$$

Since $S$ and $T$ are invertible diagonal matrices, we have

$$A' = R_{X,:}^{-1} S^{-1} S P^{-1} T^{-1} = A S P^{-1} T^{-1} = A P^{-1} (S_P T^{-1}), \tag{B.8}$$

where $S_P$ is a diagonal matrix with $(S_P)_{i,i} = S_{\pi(i),\pi(i)}$.

So $A'$ arises from $A$ by permuting and scaling columns, thus $\overline{\mathcal{A}(\mathcal{G},X)} = \overline{\mathcal{A}(\mathcal{H},X)}$. Finally, with Lemma 1 we have $\mathcal{G} \overset{X}{\sim} \mathcal{H}$.

$\square$

### B.3 PROOFS OF MATROID-PRESERVING COLUMN AUGMENTATION: A CORE COMPONENT

#### B.3.1 CONSTRUCTING PARTICULAR SOLUTIONS TO A COLUMN AUGMENTATION

We begin with the two auxiliary lemmas introduced in Appendix A, namely, Lemmas 8 and 10.

**Lemma 8** (**How the transversal matroid changes when augmenting more sources**). *For two binary matrices $Q_1 \in \{0,1\}^{m \times n_1}$ and $Q_2 \in \{0,1\}^{m \times n_2}$, we denote by $[Q_1|Q_2] \in \{0,1\}^{m \times (n_1+n_2)}$ the matrix obtained by horizontally concatenating $Q_1$ with $Q_2$. Then, we have:*

$$\mathrm{Ind}([Q_1|Q_2]) = \{Z_1 \cup Z_2 : Z_1 \in \mathrm{Ind}(Q_1),\ Z_2 \in \mathrm{Ind}(Q_2)\}. \tag{A.21}$$

*In other words, two matrices' matroids sufficiently determine the matroid of their augmentation.*

*Proof of Lemma 8.* Straightforward. Let $N_1 = [n_1]$ and $N_2 = [n_2]$ be the column indices. For the "$\subseteq$" direction, we just consider for each $Z \in \mathrm{Ind}([Q_1|Q_2])$, its matched sources in $N_1 \cup N_2$ can be split back into $N_1$ and $N_2$. For the "$\supseteq$" direction, $Z_1$ can be matched into $N_1$ and $Z_2 \backslash Z_1$ can be matched into $N_2$, which, when put together, is still independent, since $N_1$ and $N_2$ are disjoint.

$\square$

**Lemma 10** (**Constructing a particular solution for singleton augmentation**). *Let $H \in \{0,1\}^{m \times (l+1)}$ be a binary matrix with columns partitioned as $[l+1] = L \cup \{x\}$. Define*

$$D := \{i \in \{1,\ldots,m\} \mid \forall C \in \mathrm{circuits}(H) : i \in C \Rightarrow C \backslash \{i\} \notin \mathrm{Ind}(H_{:,L})\}. \tag{A.24}$$

*Define a new matrix $H'$ where $H'_{:,L} = H_{:,L}$ and the column $x$ is replaced by $H'_{i,x} = 1$ if $i \in D$ and $0$ otherwise. Then, the whole matroid remains unchanged after this column $x$ replacement:*

$$\mathrm{bases}(H') = \mathrm{bases}(H). \tag{A.25}$$

*Proof of Lemma 10.* We show that $H$ and $H'$ represent the same transversal matroid by comparing their independent families.

**In case that** $\mathrm{Ind}(H_{:,L}) = \mathrm{Ind}(H)$**,** then $D$ is the set of coloops of $H_{:,L}$. Since every maximal partial transversal of $H_{:,L}$ already contains each $d \in D$, the bases of $H'$ are precisely the bases of $H_{:,L}$, so the matroids for $H$, $H_{:,L}$, and $H'$ are the same.

**Otherwise, if** $\mathrm{Ind}(H_{:,L}) \subsetneq \mathrm{Ind}(H)$**,** then $H_{:,L}$ has a maximal partial transversal that can be extended by some element $e$ with $H_{e,x} = 1$. Because the cardinality of the bases of $H$ is one more than the cardinality of the bases of $H_{:,L}$, we have that every basis $B$ with respect to $H$ can be partitioned into a basis $B_0$ of $H_{:,L}$ and an extra element $b \in B \backslash B_0$ where $H_{b,x} = 1$. Clearly $B_0 \in \mathrm{Ind}(H')$ and if $b \in D$, then $B$ is also a basis for $H'$.

Assume that $b \notin D$, then there is a circuit $C$ of $H$ with $b \in C$ such that $C \backslash \{b\} \in \mathrm{Ind}(H_{:,L})$. The corresponding partial transversal of $H_{:,L}$ can be extended by sending $\phi(b) = x$, but then this extended partial transversal proves that $C$ is independent, contradicting that $C$ is a circuit of $H$, so $b \in D$ must be the case. Thus $\mathrm{Ind}(H) \subseteq \mathrm{Ind}(H')$.

Now let $B$ be a basis for $H'$ and assume that $B \notin \mathrm{Ind}(H)$. By set inclusion, $B \notin \mathrm{Ind}(H_{:,L})$. So there is maximal partial transversal $\phi$ of $H'$ and some $b \in B$ such that

$$\phi(b) = x \text{ and } \phi[B \backslash \{b\}] \subseteq L. \tag{B.9}$$

Hence,

$$B \backslash \{b\} \in \mathrm{Ind}(H_{:,L}) \subseteq \mathrm{Ind}(H). \tag{B.10}$$

Since $B \notin \mathrm{Ind}(H)$, there is a circuit $C \subseteq B$ with $b \in C$. But then $C \backslash \{b\} \subseteq B \backslash \{b\}$ is independent in $H_{:,L}$, so $b \notin D$. This is a contradiction to $H'_{b,\phi(b)} = H'_{b,x} = 1$, because $\phi$ is a partial transversal of $H'$. Therefore $\mathrm{Ind}(H') \subseteq \mathrm{Ind}(H)$ establishing the equality $\mathrm{Ind}(H') = \mathrm{Ind}(H)$ which implies $\mathrm{bases}(H') = \mathrm{bases}(H)$ since bases are precisely the maximal independent sets. $\square$

### B.3.2 Traversing All Solutions to a Column Augmentation

Now, we have proved the two auxiliary lemmas introduced in Appendix A.

We observe that these two lemmas are centering around one problem, as formulated in the text box titled **"Problem of Phase 2 (Reduced)"**: suppose we already know both a transversal matroid itself and the transversal matroid of it after augmented with an unknown singleton column, how can we recover this unknown singleton column to satisfy these two matroids?

In Lemma 10, we have shown a particular solution. Then, what are all the possible solutions, and how can we find them? Among all solutions, is there anything special about the particular solution given in Lemma 10? Are there any other particular solutions that might enjoy other properties?

Let us first define all these solutions:

**Definition 8** (**Solution set of matroid-preserving column augmentations**). Let $Q \in \{0,1\}^{m \times n}$ be a binary matrix. For each $x \in [n]$, we denote all column vectors that can be used to replace $Q$'s column $x$ while preserving $Q$'s matroid by:

$$\mathrm{colaug}(Q, x) \coloneqq \{ D \subseteq 2^{[m]} : \mathrm{bases}([ Q_{:,[n]\setminus\{x\}} \mid \mathbb{1}_D ]) = \mathrm{bases}(Q) \}, \qquad \text{(B.11)}$$

where the name $\mathrm{colaug}$ stands for "column augmentation", $\mathbb{1}_D$ denotes a column vector with ones at entries in $D$ and zeros elsewhere, and the notation $[ \cdot \mid \cdot ]$ denotes matrices' horizontal concatenation. Apparently, $\mathrm{colaug}(Q, x)$ is non-empty, since at least the current column $Q_{:,x}$ satisfies the condition, as well as the particular column $\mathbb{1}_D$ defined in Lemma 10, which may be different from $Q_{:,x}$.

Before we study how we may traverse all solutions in $\mathrm{colaug}(Q, x)$, let us pay more attention to the particular solutions, since 1) they offer an efficient way to get a solution directly from the matroids, without the need to solving alpha systems, which leads to algorithm speedups, and 2) as we will show below, they have meaningful implications to characterize the whole solution class.

We already have a particular solution from Lemma 10, which checks when a single-element deletion from each new circuit still leads to dependent sets before augmentation. In other words, these items are those who contribute to the newly introduced circuits. Following the proof to Lemma 10, we have that these items forms not only a solution, but also a unique maximal inclusive solution:

**Corollary 1** (**Determining items that *must not* appear in any solution**). *Let $Q \in \{0,1\}^{m \times (l+1)}$ be a binary matrix with columns partitioned as $[l + 1] = L \cup \{x\}$. Let $D \subseteq [m]$ be the particular solution constructed in Lemma 10, and let $\mathrm{colaug}(Q, x) \subseteq 2^{[m]}$ be all the solutions of $x$-column augmentation defined in Definition 8. Then, the following condition holds:*

$$D = \bigcup \mathrm{colaug}(Q, x). \qquad \text{(B.12)}$$

In other words, by giving a unique maximal solution, Lemma 10 characterizes which items in $[m]$ *can* be included in some solution(s), or equivalently, which items *must not* appear in any solution.

This then naturally leads to another question: which are the items that *must* be included in all solutions, and more importantly, is there an efficient way to determine them, just like Lemma 10, without having to traverse the whole solution set? The answer is yes, by taking complements to Lemma 10:

**Corollary 2** (**Determining items that *must* appear in all solutions**). *Let $Q \in \{0,1\}^{m \times (l+1)}$ be a binary matrix with columns partitioned as $L \cup \{x\}$. We define the "difference in cocircuits" as*

$$\mathrm{diffcc}(Q, x) \coloneqq \mathrm{cocircuits}(Q) \setminus \mathrm{cocircuits}(Q_{:,L}). \qquad \text{(B.13)}$$

*Further, for any $A \subseteq 2^V$, a set of subsets of some universe $V$, we define the minimum-sized elements of $A$ (with a slightly special treat on $\varnothing$) as:*

$$\mathrm{minimum}(A) \coloneqq \begin{cases} \{a \in A : |a| = \min_{a' \in a} |a'|\}, & \text{if } A \neq \varnothing, \\ \{\varnothing\}, & \text{otherwise.} \end{cases} \qquad \text{(B.14)}$$

*The minimal-inclusion elements of $A$ (with a slightly special treat on $\varnothing$) is then:*

$$\text{minimal}(A) := \begin{cases} \{a \in A : \nexists\, a' \in A \text{ with } a' \subsetneq a\}, & \text{if } A \neq \varnothing, \\ \{\varnothing\}, & \text{otherwise.} \end{cases} \tag{B.15}$$

*Then, the following conditions always hold:*

1. *The minimum-sized new cocircuits are all particular solutions. Moreover, they are exactly those minimal-inclusion ones among all solutions, which have a same (minimum) size:*

$$\text{minimal}(\text{colaug}(Q, x)) = \text{minimum}(\text{colaug}(Q, x)) = \text{minimum}(\text{diffcc}(Q, x)). \tag{B.16}$$

2. *Non-minimum-sized new cocircuits are not solutions, that is,*

$$(\text{diffcc}(Q, x) \setminus \text{minimum}(\text{diffcc}(Q, x))) \cap \text{colaug}(Q, x) = \varnothing. \tag{B.17}$$

3. *The intersection of minimum-sized new cocircuits may not be a solution itself, but it characterizes exactly items that must appear in all solutions:*

$$\bigcap \text{minimum}(\text{diffcc}(Q, x)) = \bigcap \text{colaug}(Q, x). \tag{B.18}$$

Roughly speaking, Corollary 2 takes a complement to Corollary 1: for any valid solution, it must complete new bases (witnessed by minimal new cocircuits), so any item that appears in every such minimum new cocircuit must appear in all valid solutions.

We use an example to illustrate these definitions above.

**Example 3** (**Illustration of Definition 8**). Suppose $Q$ is a matrix with row indices $\{1, 2, 3, 4\}$ and column indices $\{\alpha, \beta, \gamma\}$, as follows.

$$Q = \begin{matrix} & \alpha & \beta & \gamma \\ 1 \\ 2 \\ 3 \\ 4 \end{matrix}\begin{bmatrix} 0 & \times & 0 \\ \times & 0 & 0 \\ \times & \times & \times \\ \times & 0 & 0 \end{bmatrix}, \qquad \text{then} \begin{cases} \text{bases}(Q) = \{\{1, 2, 3\}, \{1, 3, 4\}\}, \\ \text{circuits}(Q) = \{\{2, 4\}\}, \\ \text{cocircuits}(Q) = \{\{1\}, \{3\}, \{2, 4\}\}. \end{cases} \tag{B.19}$$

**Consider the case with $x = \alpha$.** The remaining columns are:

$$Q_{:,\{\beta,\gamma\}} = \begin{matrix} & \beta & \gamma \\ 1 \\ 2 \\ 3 \\ 4 \end{matrix}\begin{bmatrix} \times & 0 \\ 0 & 0 \\ \times & \times \\ 0 & 0 \end{bmatrix}, \qquad \text{with} \begin{cases} \text{bases}(Q_{:,\{\beta,\gamma\}}) = \{\{1, 3\}\}, \\ \text{circuits}(Q_{:,\{\beta,\gamma\}}) = \{\{2\}, \{4\}\}, \\ \text{cocircuits}(Q_{:,\{\beta,\gamma\}}) = \{\{1\}, \{3\}\}. \end{cases} \tag{B.20}$$

The particular solution (also the maximal unique solution) given by Lemma 10 and Corollary 1 is:

$$D = \{1, 2, 3, 4\}, \tag{B.21}$$

where 1 and 3 are coloops, and 2 and 4 lead to the new circuits in $Q$.

The particular solutions given by Corollary 2 are:

$$\begin{aligned} \text{minimum}(\text{diffcc}(Q, \alpha)) &:= \text{minimum}(\text{cocircuits}(Q) \setminus \text{cocircuits}(Q_{:,\{\beta,\gamma\}})) \\ &= \text{minimum}(\{\{1\}, \{3\}, \{2, 4\}\} \setminus \{\{1\}, \{3\}\}) \\ &= \text{minimum}(\{\{2, 4\}\}), \\ &= \{\{2, 4\}\}. \end{aligned} \tag{B.22}$$

In total, there are four possible columns that can replace $Q_{:,\alpha}$ without changing the matroid:

$$\text{colaug}(Q, \alpha) = \{\{2, 4\}, \{1, 2, 4\}, \{2, 3, 4\}, \{1, 2, 3, 4\}\}. \tag{B.23}$$

For example, choose $D = \{1, 2, 4\} \in \text{colaug}(Q, \alpha)$, one may verify that

$$\text{Let } Q' = \begin{array}{c} \\ 1 \\ 2 \\ 3 \\ 4 \end{array} \begin{array}{c} \overset{\alpha \quad \beta \quad \gamma}{\begin{bmatrix} \times & \times & 0 \\ \times & 0 & 0 \\ 0 & \times & \times \\ \times & 0 & 0 \end{bmatrix}}, \qquad \text{still we have } \text{bases}(Q') = \text{bases}(Q) = \{\{1,2,3\},\{1,3,4\}\}.$$

$$(B.24)$$

For now, let us not consider how these whole solutions $\text{colaug}(Q,\alpha)$ are obtained; one may just think of them as obtained by exhaustively searching over all $2^4$ possible columns.

**Consider the case with $x = \beta$.** The remaining columns are:

$$Q_{:,\{\alpha,\gamma\}} = \begin{array}{c} \\ 1 \\ 2 \\ 3 \\ 4 \end{array} \begin{array}{c} \overset{\alpha \quad \gamma}{\begin{bmatrix} 0 & 0 \\ \times & 0 \\ \times & \times \\ \times & 0 \end{bmatrix}}, \qquad \text{with } \begin{cases} \text{bases}(Q_{:,\{\alpha,\gamma\}}) = \{\{2,3\},\{3,4\}\}, \\ \text{circuits}(Q_{:,\{\alpha,\gamma\}}) = \{\{1\},\{2,4\}\}, \\ \text{cocircuits}(Q_{:,\{\alpha,\gamma\}}) = \{\{3\},\{2,4\}\}. \end{cases}$$

$$(B.25)$$

So, the maximal solution (Lemma 10 and Corollary 1) is: $D = \{1,3\}$.

The minimal solutions (Corollary 2) are: $\text{minimum}(\text{diffcc}(Q,\beta)) = \{\{1\}\}$.

And all the possible solutions are: $\text{colaug}(Q,\beta) = \{\{1\},\{1,3\}\}$.

**Consider the case with $x = \gamma$.** The remaining columns are:

$$Q_{:,\{\alpha,\beta\}} = \begin{array}{c} \\ 1 \\ 2 \\ 3 \\ 4 \end{array} \begin{array}{c} \overset{\alpha \quad \beta}{\begin{bmatrix} 0 & \times \\ \times & 0 \\ \times & \times \\ \times & 0 \end{bmatrix}}, \qquad \text{with } \begin{cases} \text{bases}(Q_{:,\{\alpha,\beta\}}) = \{\{1,2\},\{1,3\},\{1,4\},\{2,3\},\{3,4\}\}, \\ \text{circuits}(Q_{:,\{\alpha,\beta\}}) = \{\{2,4\},\{1,2,3\},\{1,3,4\}\}, \\ \text{cocircuits}(Q_{:,\{\alpha,\beta\}}) = \{\{1,3\},\{1,2,4\},\{2,3,4\}\}. \end{cases}$$

$$(B.26)$$

So, the maximal solution (Lemma 10 and Corollary 1) is: $D = \{1,3\}$.

The minimal solutions (Corollary 2) are: $\text{minimum}(\text{diffcc}(Q,\gamma)) = \{\{1\},\{3\}\}$.

And all the possible solutions are: $\text{colaug}(Q,\beta) = \{\{1\},\{3\},\{1,3\}\}$.

$$\triangle$$

So far, we have introduced the definitions about "matroid-preserving column augmentations", and have provided particular ways to construct both the unique maximal solution and the set of minimal solutions. Then, starting from these particular solutions, or starting from any solution, how can we span to the whole solution set? This is answered by the following result.

We now present the structure among all column augmentations, which describes how how the whole solutions can be traversed, and is thus important to our result about equivalence class traversal. In particular, any two solutions can reach each other by a sequence of "edge additions/deletions".

**Lemma 12 (The whole column augmentations can be traversed by edge additions/deletions).**
*For any matrix $Q \in \{0,1\}^{m \times n}$ and a column index $x \in [n]$, we define a digraph termed $\mathcal{G}_{Q,x}^{\text{aug}}$, which is a Hasse diagram with vertices being elements of $\text{colaug}(Q,x)$, and edges being:*

$$D_i \to D_j \in \mathcal{G}_{Q,x}^{\text{aug}} \iff D_i \subsetneq D_j \text{ with } |D_j \setminus D_i| = 1, \quad \forall D_i, D_j \in \text{colaug}(Q,x). \quad (B.27)$$

*Then, this digraph is weakly connected.*

We illustrate Lemma 12 by recalling Example 3:

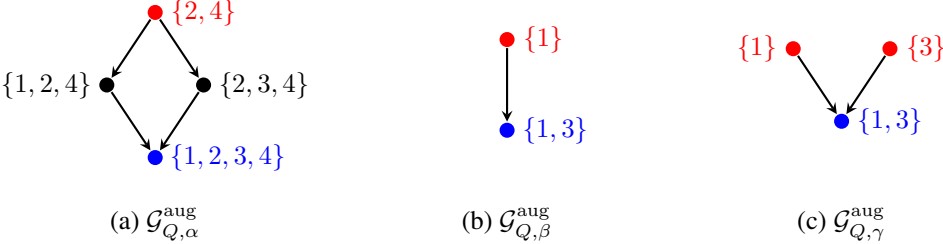

Figure 4: Example Hasse diagrams (defined in Lemma 12) over the solution sets of matroid-preserving column augmentations. Instances from the earlier Example 3 are used. In each diagram, the root vertices, corresponding to the minimal solutions (see Corollary 2) are highlighted in red, while the leaf vertex, corresponding to the unique maximal solution (see Corollary 1) are highlighted in blue.

*Proof of Lemma 12.* Let

$$D = \{i \in \{1, \ldots, m\} \mid \forall C \in \text{circuits}(Q) : i \in C \Rightarrow C \backslash \{i\} \notin \text{Ind}(Q_{:,[n]\backslash\{x\}})\}. \tag{B.28}$$

From Lemma 10, we know $D \in \text{colaug}(Q, x)$.

Let $D' \in \text{colaug}(Q, x)$, then $D' \subseteq D$, because if there is $d' \in D'$ with $d' \notin D$, then there exists a circuit $C$ in $Q$ with $d' \in C$ such that $C \backslash \{d'\}$ has a partial transversal omitting the column $x$. This partial transversal may be extended to $C$ by sending $d'$ to the column $x$, which then can be extended to a basis $B_C \supseteq C$ in the transversal matroid represented by $(Q_{:,[n]\backslash\{x\}} \mathbb{1}_{D'})$. But $Q$ cannot have a basis that contains one of its circuits, which implies that $D' \notin \text{colaug}(Q, x)$. So $D$ is the maximal element of $\text{colaug}(Q, x)$.

Let $D' \in \text{colaug}(Q, x)$ with $D' \neq D$ and let $d \in D \backslash D'$. Let $B \in \text{bases}(Q)$, then $B \in \text{bases}((Q_{:,[n]\backslash\{x\}} \mathbb{1}_{D'}))$. The maximal partial transversal $\phi$ that witnesses the independence of $B$ with respect to $(Q_{:,[n]\backslash\{x\}} \mathbb{1}_{D'})$ also witnesses the independence of $B$ with respect to $(Q_{:,[n]\backslash\{x\}} \mathbb{1}_{D'\cup\{d\}})$. So we have

$$\text{bases}(Q) = \text{bases}((Q_{:,[n]\backslash\{x\}} \mathbb{1}_{D'})) \subseteq \text{bases}((Q_{:,[n]\backslash\{x\}} \mathbb{1}_{D'\cup\{d\}})). \tag{B.29}$$

Now, let $B \in \text{bases}((Q_{:,[n]\backslash\{x\}} \mathbb{1}_{D'\cup\{d\}}))$. The maximal partial transversal witnessing $B$ with respect to $(Q_{:,[n]\backslash\{x\}} \mathbb{1}_{D'\cup\{d\}})$ is also a maximal partial transversal with respect to $(Q_{:,[n]\backslash\{x\}} \mathbb{1}_D)$, so $B \in \text{bases}((Q_{:,[n]\backslash\{x\}} \mathbb{1}_D)) = \text{bases}(Q)$. Therefore,

$$\text{bases}((Q_{:,[n]\backslash\{x\}} \mathbb{1}_{D'\cup\{d\}})) = \text{bases}(Q), \text{ and } D' \cup \{d\} \in \text{colaug}(Q, x). \tag{B.30}$$

Hence, for every $D_0, D_1 \in \text{colaug}(Q, x)$ there is a directed path to $D$ which gives

$$D_0 \to \cdots \to D \leftarrow \cdots \leftarrow D_1, \tag{B.31}$$

Therefore, $\mathcal{G}_{Q,x}^{\text{aug}}$ is weakly connected. $\qquad\square$

### B.3.3 FROM ONE COLUMN AUGMENTATION TO MULTIPLE COLUMNS' JOINT AUGMENTATION

We have now shown how one can obtain particular solutions or traverse all solutions to a single column augmentation. In what follows, we shift from one column augmentation to multiple column augmentation, which directly relates to our final graphical criterion to be shown in next section: we need to augment the whole $X$ columns, identifying their outgoing edges.

Interestingly, this seemingly combinatorially complex satisfiability problem can be decomposed locally, that is, it suffices to satisfy each singleton column augmentation independently, shown below.

**Lemma 9** (**Reducing all union equivalence checks to singleton checks**)**.** *Let* $Q, H \in \{0,1\}^{m \times n}$ *be two binary matrices with columns partitioned as* $[n] = L \cup X$ *for a fixed L. Then, the condition*

$$\text{bases}(Q_{:,L \cup \boldsymbol{x}}) = \text{bases}(H_{:,L \cup \boldsymbol{x}}), \quad \forall \boldsymbol{x} \subseteq X, \tag{A.22}$$

*holds, if and only if the condition*

$$\begin{cases} \text{bases}(Q_{:,L}) = \text{bases}(H_{:,L}), \ \text{and} \\ \text{bases}(Q_{:,L \cup \{X_i\}}) = \text{bases}(H_{:,L \cup \{X_i\}}), \quad \forall X_i \in X, \end{cases} \tag{A.23}$$

*holds.*

*Proof of Lemma 9.* Clearly, condition A.23 is a special case of condition A.22 with the choices $\boldsymbol{x} \in \{\varnothing, \{X_1\}, \dots, \{X_k\}\}$, thus if A.22 holds, then so does A.23.

If A.22 does not hold, then there is some $\boldsymbol{x} \subseteq X$ such that $\text{bases}(Q_{:,L \cup \boldsymbol{x}}) \neq \text{bases}(H_{:,L \cup \boldsymbol{x}})$. W.l.o.g. we may assume that there is some $B \in \text{bases}(Q_{:,L \cup \boldsymbol{x}})$ with $B \notin \text{bases}(H_{:,L \cup \boldsymbol{x}})$.

**If** $|\boldsymbol{x}| \leq 1$**,** then A.23 clearly does not hold.

**Now assume that** $|\boldsymbol{x}| > 1$**.** Choose $B \in \text{bases}(Q_{:,L \cup \boldsymbol{x}}) \setminus \text{bases}(H_{:,L \cup \boldsymbol{x}})$ and a partial transversal $\phi \colon B \to L \cup \boldsymbol{x}$ for $Q_{:,L \cup \boldsymbol{x}}$ such that $\delta := |\{b \in B \mid H_{b,\phi(b)} = 0\}|$ is minimal. Due to the minimality of $\delta$ (may be obtained via basis exchange), there is exactly one $b \in B$ such that $H_{b,\phi(b)} = 0$. Let

$$B' = \{b' \in B \mid \phi(b') \in L \cup \{\phi(b)\}\}, \tag{B.32}$$

then $B'$ is a basis of $Q_{:,L \cup \{\phi(b)\}}$, but $B'$ is not a basis of $H_{:,L \cup \{\phi(b)\}}$, shown below:

Assume that $B'$ is a basis of $H_{:,L \cup \{\phi(b)\}}$, then there is a partial transversal $\psi \colon B' \to L \cup \{\phi(b)\}$. Construct

$$\phi' \colon B \to H_{:,L \cup \boldsymbol{x}} \tag{B.33}$$

such that

$$\phi'(b') = \begin{cases} \psi(b') & \text{for } b' \in B', \\ \phi(b') & \text{for } b' \in B \setminus B'. \end{cases} \tag{B.34}$$

$\phi'$ is a partial transversal, because

$$\phi[B \setminus B'] \cap (L \cup \{\phi(b)\}) = \varnothing, \tag{B.35}$$

due to the definition of $B'$. Thus if $\phi'(b_0) = \phi'(b_1)$. Then:

1° Either $\{b_0, b_1\} \subseteq B'$: in this case, $\phi'(b_0) = \psi(b_0) = \psi(b_1) = \phi'(b_1)$ implies $b_0 = b_1$;

2° Otherwise we have $\{b_0, b_1\} \subseteq B \setminus B'$, and then $\phi'(b_0) = \phi(b_0) = \phi(b_1) = \phi'(b_1)$ implies $b_0 = b_1$.

But then $\phi'$ witnesses that $B$ is a basis of $H_{:,L \cup \boldsymbol{x}}$, contradicting the original assumption. Thus $\text{bases}(Q_{:,L \cup \{\phi(b)\}}) \neq \text{bases}(H_{:,L \cup \{\phi(b)\}}$ and A.23 does not hold, too.

We have then finished proof to Lemma 9. $\qquad\square$

### B.4 Proofs of the Graphical Criterion and Transformational Characterization

We first note that the graphical criterion (Theorem 2) is a direct consequence of Lemma 9, that is, instead of checking for bases of all subsets $\boldsymbol{x} \subseteq X$, we only need to check for bases for each singleton $X_i \in X$. Since Lemma 9 is already proved above, in this section, we focus on the proof of the transformational characterization (Theorem 3).

**Lemma 7** (**Admissible edge additions/deletions**). *Let $(\mathcal{G}, X)$ be an irreducible model. For any edge $V_i \to V_j$ not currently in $\mathcal{G}$, adding it to $\mathcal{G}$ preserves equivalence on $X$ if and only if:*

$$r_{\mathcal{G}}(V_i\text{'s nonchildren}\backslash\{V_j\}, \ L\backslash\{V_i\}) \ < \ r_{\mathcal{G}}(V_i\text{'s nonchildren}, \ L\backslash\{V_i\}), \tag{20}$$

*where $V_i$'s nonchildren denotes $V(\mathcal{G})\backslash \text{ch}_{\mathcal{G}}(V_i)\backslash\{V_i\}$, i.e., zero entries in support column $Q^{(\mathcal{G})}_{:,V_i}$. Conversely, an edge can be deleted if and only if it can be re-added by this criterion.*

*Proof of Lemma 7.* Let us first prove a weaker version of this result, that is, without the permutation part involved in checking equivalence (Lemma 5). Put formally, let $\mathcal{H}$ be the digraph after altering an edge $V_i \to V_j$ in $\mathcal{G}$. We study the if and only if condition (in terms of this edge) for the following

$$r_{\mathcal{G}}(Z, Y) = r_{\mathcal{H}}(Z, Y) \quad \text{for all } Z \subseteq V(\mathcal{G}) \text{ and } L \subseteq Y \subseteq V(\mathcal{G}) \tag{B.36}$$

to hold. According to Lemma 9, this condition holds if and only if a reduced version hold:

$$\text{bases}(Q^{(\mathcal{G})}_{:,L\cup\{V_i\}}) \ = \ \text{bases}(Q^{(\mathcal{H})}_{:,L\cup\{V_i\}}). \tag{B.37}$$

That is, one only need to check whether a single transversal matroid is changed. Then, when can an edge in a bipartite graph be altered while the transversal matroid induced by this bipartite graph keeps unchanged? We show the condition by the following lemma.

**Lemma 13** (**When an edge in a bipartite graph can be altered without changing the transversal matroid**). *Let $Q \in \{0,1\}^{m\times n}$ be a binary support matrix. For any $(V_j, V_i) \in [m] \times [n]$ such that $Q_{V_j, V_i} = 0$, define $H \in \{0,1\}^{m\times n}$ by $H_{V_j, V_i} = 1$ and $H_{z,y} = Q_{z,y}$ for all other entries. For convenience, denote $V_i$'s non-children in $Q$ and column indices except for $V_i$ by:*

$$\begin{aligned} R &:= \{z \in [m] : Q_{z, V_i} = 0\}; \\ Y &:= [n]\backslash\{V_i\}. \end{aligned} \tag{B.38}$$

*Then, the following conditions are equivalent to each other:*

1. *$\text{bases}(Q) = \text{bases}(H)$;*

2. *$\text{bases}(Q_{R,:}) = \text{bases}(H_{R,:})$;*

3. *$\text{mrank}(Q_{R,:}) = \text{mrank}(H_{R,:})$;*

4. *$\text{mrank}(Q_{R\backslash\{V_j\}, Y}) < \text{mrank}(Q_{R,Y})$, that is, $V_j$ is a coloop among $R$ in the transversal matroid induced by $Q_{R,Y}$, and so removing it from ground set lowers the rank (by 1).*

*Proof of Lemma 13.* We first have two immediate observations. (i) By construction, $Q_{R,\{V_i\}}$ is the zero column, whereas $H_{R,\{V_i\}}$ has a single 1 in row $V_j$. (ii) For any $Z \subseteq [m]$ with $V_j \notin Z$, the submatrices $Q_{Z,:}$ and $H_{Z,:}$ coincide, hence their matching ranks (and base behavior) coincide.

We now prove the implications among the four conditions by

$$(1) \Rightarrow (2) \Rightarrow (3) \iff (4) \Rightarrow (2) \Rightarrow (1).$$

$(1) \Rightarrow (2)$. Trivial. Taking restrictions on ground sets preserves equality of matroids.

$(2) \Rightarrow (3)$. Trivial. Same matroids have the same ranks.

$(3) \iff (4)$. Let $\nu := \text{mrank}(Q_{R,Y})$ and $\nu' := \text{mrank}(Q_{R\backslash\{V_j\}, Y})$. Note that

$$\text{mrank}(Q_{R,:}) \ = \ \text{mrank}(Q_{R,Y}) \ = \ \text{mrank}(H_{R,Y}) \ = \ \nu, \tag{B.39}$$

because the column $V_i$ is useless (full zero) for $R$ in $Q$. In $H$, the only new edge incident to $R$ is $(V_i, V_j)$; therefore any matching on $R$ in $H$ is either:

- a matching that *does not use* column $V_i$, hence has size at most $\nu$, or

- a matching that *does use* the edge $(V_i, V_j)$ and then matches the remaining $R \setminus \{V_j\}$ into $Y$, hence has size at most $1 + \nu'$.

Consequently,

$$\mathrm{mrank}(H_{R,:}) = \max\{\nu, \ 1 + \nu'\}. \tag{B.40}$$

Thus $\mathrm{mrank}(H_{R,:}) = \mathrm{mrank}(Q_{R,:})$ holds if and only if $1 + \nu' \leq \nu$, i.e., $\nu' < \nu$, which is precisely (4). This proves (3) $\iff$ (4).

(4) $\Rightarrow$ (2). Assume (4). In the transversal matroid $\mathcal{M}$ induced by $Q_{R,Y}$, the inequality $\mathrm{mrank}(Q_{R \setminus \{V_j\}, Y}) < \mathrm{mrank}(Q_{R,Y})$ means that $V_j$ is a *coloop* of $\mathcal{M}$ (see Definition 7). A standard matroid identity for coloops states that for all $Z \subseteq R \setminus \{V_j\}$,

$$\mathrm{mrank}(Q_{Z \cup \{V_j\}, Y}) = \mathrm{mrank}(Q_{Z,Y}) + 1. \tag{B.41}$$

Combining this with Equation (B.40) (applied now to *each $Z \subseteq R$*) shows that adding the edge $(V_i, V_j)$ cannot change the matching rank of *any* $Z \subseteq R$; so in particular, the bases on $R$ is unchanged: $\mathrm{bases}(Q_{R,:}) = \mathrm{bases}(H_{R,:})$.

(2) $\Rightarrow$ (1). For any $Z \subseteq [m]$, we write $Z_R := Z \cap R$ and $Z_{\mathrm{out}} := Z \setminus R$.

- If a maximum matching of $H_{Z,:}$ *does not use* $(V_i, V_j)$, then it is also a matching in $Q_{Z,:}$, and the ranks agree.

- If a maximum matching of $H_{Z,:}$ *does use* $(V_i, V_j)$, then its restriction to $Z_R$ is a maximum matching of $H_{Z_R,:}$ that uses the column $V_i$. By (2) (which holds for all subsets of $R$ as shown above), there exists a maximum matching of $Q_{Z_R,:}$ of the *same* size that avoids $V_i$. Replacing the $H$-matching on $Z_R$ by this $Q$-matching on $Z_R$ (and keeping the $Z_{\mathrm{out}}$-part unchanged) yields a matching of $Q_{Z,:}$ of the same cardinality as the original one in $H_{Z,:}$.

Hence $\mathrm{mrank}(Q_{Z,:}) = \mathrm{mrank}(H_{Z,:})$ for all $Z \subseteq [m]$, which is equivalent to $\mathrm{bases}(Q) = \mathrm{bases}(H)$.

All implications are proved, so the four conditions are equivalent. The result on deleting an edge is just the same as adding back this edge from the resulted graph. $\qquad \square$

The condition shown in Lemma 13 is exactly the condition we have in Lemma 7, and hence the weaker version without permutation (Equation (B.36)) is already proved.

The prove the full version, we only need to show that when the condition in Equation (B.37) fails, then with any permutation they still cannot be rendered equivalent. This is straightforward, since with one edge difference, the independent sets $\mathrm{Ind}(Q^{(\mathcal{G})}_{:,L \cup \{V_i\}})$ and $\mathrm{Ind}(Q^{(\mathcal{H})}_{:,L \cup \{V_i\}})$, if not equal, must admit a strict inclusion relation between them, so there is no way for these two matroids to be isomorphic.

We have now finished the proof of Lemma 7.

$\qquad \square$

**Theorem 3** (**Transformational characterization of the equivalence class**). *Two irreducible models $(\mathcal{G}, X)$ and $(\mathcal{H}, X)$ are equivalent if and only if $\mathcal{G}$ can be transformed into $\mathcal{H}$, up to $L$-relabeling, via a sequence of admissible cycle reversals and edge additions/deletions, as defined in Lemmas 6 and 7.*

*Here, "up to $L$-relabeling" means there exists a relabeling of $L$ in $\mathcal{H}$ yielding a digraph $\mathcal{H}'$ such that $\mathcal{G}$ reaches $\mathcal{H}'$ via the sequence. Moreover, at most one cycle reversal is needed in this sequence.*

*Proof of Theorem 3.* The proof to this result for traversing the equivalence class for the whole class directly relates to the helpful lemmas we have shown in Appendix B.3, i.e., how the whole solution set of column(s) augmentation is structured.

Lemma 12 is core to our result: it shows that the whole space of satisfiable column augmentations can be traversed by applying sequences of "one-edge different" operations, and these operations are exactly the "admissible edge additions/deletions" we show in Lemma 7 (for edges from $X$), and Lemma 13 (for edges from $L$). From Lemma 12, we also have a way to traverse all bipartite graphs that realizes a given transversal matroid. This can be viewed as a generalization from single-column to multi-column augmentation:

**Corollary 3** (**Traverse all bipartite graphs that realize a transversal matroid**). *Let $Q, H \in \{0,1\}^{m \times n}$ be two binary matrices. $Q$ and $H$ induce a same transversal matroid, i.e., $\mathrm{bases}(Q) = \mathrm{bases}(H)$, if and only if $Q$ can reach $H$ via a sequence of admissible edge additions/deletions defined in Lemma 13, followed by a column permutation. Moreover, similar to Corollary 1, among all matrices that can be reached from $Q$ via sequences of edge additions/deletions, there exists a unique maximal matrix whose support is the union of supports of all these reachable matrices.*

Corollary 3 directly relates to our traversal on the $Q_{:,L}^{(\mathcal{G})}$ part. But it is worth noting that unlike the independent decomposition of column augmentations for each $X_i$, here the edge additions/deletions have to be operated within the whole matrix space. We cannot simply run column augmentation for each $L_i$ and take the Cartesian product. To see this, let $Q = [[1,1]]$ with columns $\alpha, \beta$. Obviously, $\mathrm{colaug}(Q, \alpha) = \{\varnothing, \{1\}\}$, and also $\mathrm{colaug}(Q, \beta) = \{\varnothing, \{1\}\}$. However, we cannot take a product and let $Q' = [[0,0]]$, which induces a matroid different from $Q$.

Now, putting Lemma 12, Corollary 3, and Lemma 9 together, we have a way to traverse all digraphs that achieve the same matroids over the tower of all sources that include some latent vertices:

**Corollary 4** (**Traverse all digraphs that realize same matroid tower under latents**). *Let $Q, H \in \{0,1\}^{m \times n}$ be two binary matrices with columns partitioned as $[n] = L \cup X$. Then, the condition*

$$\mathrm{bases}(Q_{:,L \cup \boldsymbol{x}}) = \mathrm{bases}(H_{:,L \cup \boldsymbol{x}}), \quad \forall \boldsymbol{x} \subseteq X, \tag{B.42}$$

*holds, if and only if the $\mathrm{bases}(Q_{:,L}) = \mathrm{bases}(H_{:,L})$, (graphical criterion in Corollary 3), and for all $X_i \in X$, $\{i \in [m] : Q_{i,X_i} = 1\} \in \mathrm{colaug}(H_{:,L \cup \{X_i\}}, X_i)$ (graphical criterion in Lemma 12).*

In other words, $Q$ can reach $H$ via a sequence of admissible edge additions/deletions, followed by a permutation only among the columns in $L$. Corollary 4 directly relates to our traversal on the $Q^{(\mathcal{G})}$.

If we are to allow the equivalence up to row permutation, i.e., permuting the ground set as in Lemma 5, only a row permutation appended to the end of the operations in Corollary 4 is needed.

Finally, a treatment to ensure nonzero diagonals for digraphs.

- Since in our case we need to exclude matrices with zero diagonals, this row permutation becomes the "at most one step" within the sequence (Theorem 3), instead of at the end.

- The $L$-relabeling part, however, can still be put at the end, since to relabel the $L$ vertices in a digraph $\mathcal{G}$, it is to apply a permutation on the columns $Q_{:,L}^{(\mathcal{G})}$ first, and then to apply the same permutation back on the rows $Q_{L,:}^{(\mathcal{G})}$. This operation still ensures the nonzero diagonals. This becomes the "up to $L$-relabeling" term in Theorem 3.

We have now finished the proof of Theorem 3.

$\square$

### B.5 OTHER IMMEDIATE OR KNOWN RESULTS

We omit the proofs of the remaining results occurred in this manuscript: some of them follow immediately from the already proved results, including Lemmas 1, 4 and 5, and the others are results shown by existing work, including Lemmas 2 and 6 and Theorem 1.

## C  Discussion

### C.1  Summary: A Side-by-side Comparison Between Path Ranks and Edge Ranks

Table 1: A side-by-side comparison between path ranks and edge ranks.

| Aspect | Path rank (Matrix rank) | Edge rank (Matching rank) |
|---|---|---|
| Intuition | Algebraic independence | Combinatorial independence |
| Full rank of a $d \times d$ square matrix $M$ | $\mathrm{rank}(M) = d \iff$ the *determinant* of $M$ is nonzero | $\mathrm{mrank}(M) = d \iff$ the *permanent* of $M$ is nonzero |
| Graphical constraints in digraphs | $\rho_{\mathcal{G}}(Z, Y)$, the maximum number of vertex-disjoint directed paths from $Y$ to $Z$ (Definition 3), equals the matrix rank of generic mixing sub-matrices $A_{Z,Y}$ (Lemma 2) | $r_{\mathcal{G}}(Z, Y)$, the size of the maximum bipartite matching from $Y$ to $Z$ via direct edges (Definition 4), equals the matching rank of the support submatrix $Q_{Z,Y}^{(\mathcal{G})}$ (Lemma 4) |
| Matroid representations | Strict gammoids in digraphs (Perfect, 1968) | Transversal matroids in bipartite graphs (Ingleton & Piff, 1973) |
| Duality (Theorem 1) | $\min(|Z|, |Y|) - \rho_{\mathcal{G}}(Z, Y) = |V| - \max(|Z|, |Y|) - r_{\mathcal{G}}(V \backslash Y, V \backslash Z)$ | |

### C.2  Another Example Distributional Equivalence Class

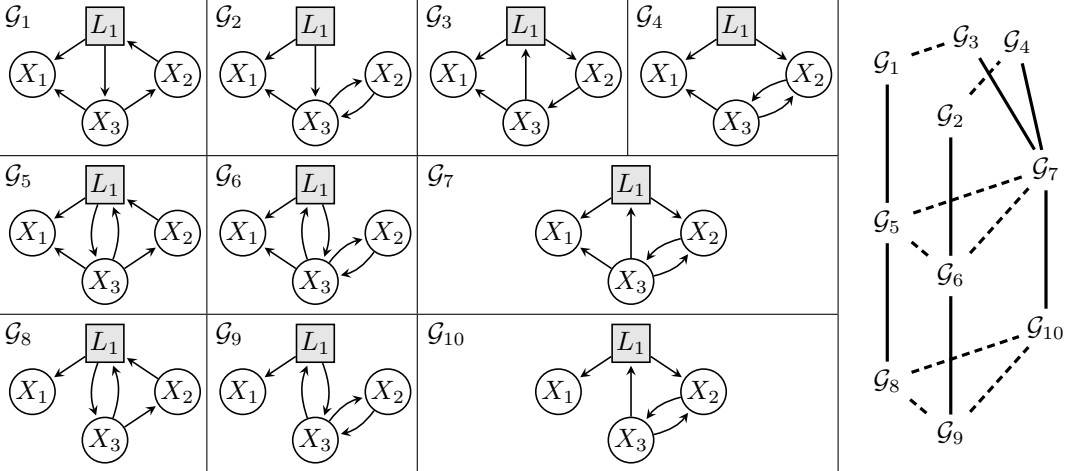

Figure 5: Left: An example distributional equivalence class consisting of 10 digraphs. Right: Transitions among these digraphs, where solid edges indicate edge additions or deletions, and dashed edges indicate cycle reversals.

We show another example distributional equivalence class in Figure 5, in addition to the Figure 3 already shown in main text. The points of this example, different from those of Figure 3, are that:

1. Partitioned by cycle reversals (removing the dashed edges in the right of Figure 5), the classes connected by only edge additions/deletions (solid edges) are not necessarily isomorphic to each other. Here, there are $3, 3, 4$ digraphs within each such class, respectively.

2. To illustrate cases where cycles intersect.

### C.3 A PRESENTATION OF THE EQUIVALENCE CLASS

In the main text, we have presented both a graphical criterion to check for equivalence (Theorem 2) and a transformational characterization to traverse the entire equivalence class (Theorem 3). These results are analogous to the "same adjacencies and v-structures" and the "covered edge reversal (Meek conjecture)" in the fully observed, acyclic, Markov equivalence setting.

However, note that in that classical setting, there is another familiar result, CPDAG, which serves as an informative presentation of the equivalence class. This naturally raises the question: can we construct an analogous presentation in the context of this work? We answer this affirmatively. In what follows, we outline how this presentation can be constructed step by step.

**Step 1. Identifiability of ancestral relations among observed variables.** As a preliminary observation, we first note that the ancestral relations among observed variables $X$ are invariant across all equivalent digraphs (this follows from how admissible edge additions/deletions are defined, and the fact that cycle reversal does not alter the ancestral relations). Thus, presenting an arbitrary digraph in the equivalence class suffices to inform users the true ancestral relations among $X$. For applications such as experimental design involving observed variables, this alone is informative enough.

**Step 2. Unique maximal digraph within the class.** We show that under each cycle-reversal configuration, there exists a unique maximal digraph in the equivalence class such that every equivalent digraph is a subgraph of it. We further provide an explicit construction of this maximal digraph (Corollary 1), without needing to enumerate all equivalent digraphs and then take the maximal one. Analogous to the "largest chain graph" in Frydenberg (1990), this maximal digraph can serve as a basis of the presentation, informing users which causal relations are *guaranteed to be absent*.

**Step 3. Characterizing edges that must appear.** Building on the previous step, we also characterize edges that must be present in *all* equivalent digraphs. Again, these edges can be determined efficiently via a graphical condition (Corollary 2), without needing to enumerate all equivalent digraphs and then take the intersection. Note that it is an explicit construction, so iterative procedures (such as arrow propagation in Meek rules (Meek, 1995)) are not needed either. These edges can be visually highlighted on the basis maximal digraph, in analogous to the arrows in CPDAGs, or "visible edges" in PAGs (Zhang, 2008b). They inform users which causal relations *they can fully trust*.

Such presentation is formally defined in Theorem 4, and examples of it are shown in Figure 6.

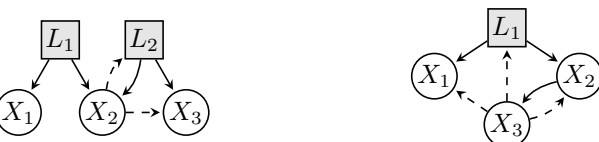

Figure 6: Illustrative presentations of equivalence classes. **Left:** Presentation of equivalent digraphs $\mathcal{G}_1, \mathcal{G}_2, \mathcal{G}_3$ under a same cycle-reversal configuration from Figure 3. The basis digraph shown is the unique maximal equivalent digraph (Step 2 above). In it, solid edges denote those that must appear in all equivalent digraphs, while dashed edges are those that can be removed (Step 3 above). One may use Corollaries 1 and 2 to check how they are determined. **Right:** A similar presentation for digraphs $\mathcal{G}_3, \mathcal{G}_4, \mathcal{G}_7, \mathcal{G}_{10}$ from Figure 5. **Remark:** One might ask why we present the equivalence class separately for each cycle-reversal configuration rather than for the entire class. The reason is that taking the union over all digraphs in the entire class can, unlike within one configuration, yield a supergraph that is itself out of the equivalence class, potentially producing misleading interpretations. In fact, this separation only leads to more informative presentations, shown by Theorem 4 below.

**Theorem 4 (Presentation of an equivalence class).** *For an irreducible model $(\mathcal{G}, X)$, we construct a digraph whose vertices are $V(\mathcal{G})$ and whose directed edges come in two types: solid and dashed. All edges are determined by Corollary 1, and among them the solid ones are determined by Corollary 2. We denote this digraph by $\mathrm{CP}(\mathcal{G})$, echoing the sense of "complete partial" as in CPDAGs.*

*For convenience, let $\mathcal{E}(\mathcal{G})$ be the whole equivalence class, that is, the set of all digraphs $\mathcal{H}$ on vertices $V(\mathcal{G})$ such that $\mathcal{H} \overset{X}{\sim} \mathcal{G}$. Let $\mathcal{F}(\mathcal{G})$ denote the set of all digraphs reachable from $\mathcal{G}$ via sequences of admissible edge addition/deletions as defined in Lemma 7. Clearly, $\mathcal{F}(\mathcal{G}) \subseteq \mathcal{E}(\mathcal{G})$.*

*Then, the presentation $\mathrm{CP}(\mathcal{G})$ enjoys the following properties:*

1. *$\mathrm{CP}(\mathcal{G}) \in \mathcal{F}(\mathcal{G})$;*

2. *For every $\mathcal{H} \in \mathcal{F}(\mathcal{G})$, the edge set of $\mathcal{H}$ is a subset of the edge set of $\mathrm{CP}(\mathcal{G})$;*

3. *The intersection of the edge sets of all $\mathcal{H} \in \mathcal{F}(\mathcal{G})$ equals the solid edges of $\mathrm{CP}(\mathcal{G})$;*

4. *For every $\mathcal{H} \in \mathcal{E}(\mathcal{G})$, let $\mathrm{CP}(\mathcal{H})$ be its own presentation. Then, $\mathrm{CP}(\mathcal{H})$ can be transformed into $\mathrm{CP}(\mathcal{G})$ via an L-relabel and a cycle reversal (alongside the solid/dashed edge types).*

It is worth noting that a dashed edge in a presentation means that there exists at least one equivalent digraph without this edge. However, it does not imply that dashed edges can be arbitrarily removed without affecting equivalence: they have to obey the rank constraints. This is in a similar spirit of undirected edges in a CPDAG: an undirected edge means that there exist at least two equivalent DAGs who have different orientations on this edge. However, it does not imply that undirected edges can be arbitrarily oriented: there are additional constraints like no new v-structures, no cycles, etc.

We are not sure whether such additional constraints can, or should, be also incorporated into the presentation, or at least summarized as a set of rules like Meek rules (especially given the availability of an interactive traversal tool). But in any case, we put it here as a possible future step:

**Step 4. Quantifying bounds on edges between vertex groups.** Extending step 3, one may describe bounds on the number of edges between vertex groups (e.g., "at least 2 and at most 4 edges from vertices $Y$ to vertices $Z$"). Such constraints may be presented like "underlined bows" in cyclic digraphs (Richardson, 1996) or "hyperedges" in mDAGs (Evans, 2016). We have not developed this result (though we hypothesize that they likely also follow from Theorem 2), because we are not sure how much practical informativeness it can offer to users.

Lastly, we also list the presentation of prior knowledge as a future step.

**Step 5. Incorporating additional prior knowledge.** As with other equivalence presentations, prior knowledge such as acyclicity, stable cycles, or certain causal orderings can further refine the equivalence class and its presentation (Perković et al., 2017). While we have not explored this part either, it motivates future theoretical developments, such as interventional equivalence classes, and parameter identifiability results based on the equivalence class established in this work.

### C.4 Examples of Non-Rank Constraints in Mixing Matrices

In Lemma 3 we have shown that path rank equivalence in mixing matrices sufficiently lead to distributional equivalence. However, this does not imply that there are no other constraints in mixing matrices. As an analogy, in the causally sufficient linear Gaussian system, CI equivalence (zero partial correlations in covariance matrices) sufficiently lead to distributional equivalence, but there are still other constraints, like the Tetrad constraints in the covariance matrices.

Below we give an example of non-rank constraints in mixing matrices.

Consider a digraph $\mathcal{G}$ with 4 vertices:

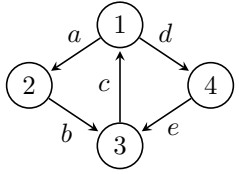

Its mixing matrix is:

$$
A \;=\; \begin{array}{c} \\ 1 \\ 2 \\ 3 \\ 4 \end{array} \begin{array}{cccc} 1 & 2 & 3 & 4 \\ \left[\begin{array}{cccc} 1 & bc & c & ce \\ a & 1-cde & ac & ace \\ ab+de & b & 1 & e \\ d & bcd & cd & 1-abc \end{array}\right] \end{array} \times \frac{1}{1-abc-cde} \;. \tag{C.1}
$$

We can verify the following constraint holds:

$$
\begin{aligned}
& A_{2,4}\,A_{3,2}\,A_{4,1} \;-\; A_{2,1}\,A_{3,4}\,A_{4,2} \\
&= ace \times b \times d \;-\; a \times e \times bcd \\
&= 0.
\end{aligned} \tag{C.2}
$$

Just like rank constraints, this constraint is also immune to arbitrary column scaling, that is, it also survives in the OICA estimated mixing matrix. However, this is not a rank constraint.

One may also verify some other non-rank constraints in this $A$, for example,

$$
\begin{aligned}
& A_{2,2}\,A_{3,4}\,A_{4,1} + A_{2,4}\,A_{3,1}\,A_{4,2} + A_{2,1}\,A_{3,2}\,A_{4,4} \\
&= 2\,A_{2,1}\,A_{3,4}\,A_{4,2} - A_{2,2}\,A_{3,1}\,A_{4,4},
\end{aligned} \tag{C.3}
$$

and

$$
\begin{aligned}
& A_{2,4}^2\,A_{3,1}\,A_{3,2}\,A_{4,2} + A_{2,1}\,A_{2,2}\,A_{3,4}^2\,A_{4,2} + A_{2,1}\,A_{2,4}\,A_{3,2}^2\,A_{4,4} \\
&= 2\,A_{2,1}\,A_{2,4}\,A_{3,2}\,A_{3,4}\,A_{4,2} + A_{2,2}\,A_{2,4}\,A_{3,1}\,A_{3,2}\,A_{4,4},
\end{aligned} \tag{C.4}
$$

both of which are also immune to column scaling.

We are not sure whether there are any specific geometry interpretations underlying these equality constraints. These examples are brutal-force searched from ideal elimination.

We notice that these equality constraints occur among the $\{2,3,4\}$ rows, meaning that when vertex $\{1\}$ is latent and $\{2,3,4\}$ observed, these constraints will also appear in the OICA mixing matrix. Fortunately, with Lemma 3, we know rank constraints alone can determine the distributional equivalence, so the equivalence among these constraints as well.

For example, one may verify that these constraints also occur in all 3 digraphs in the equivalence class, shown below, while this equivalence class is obtained only by the rank-based criterion (which is trivial in this case since only cycle reversals are applied).

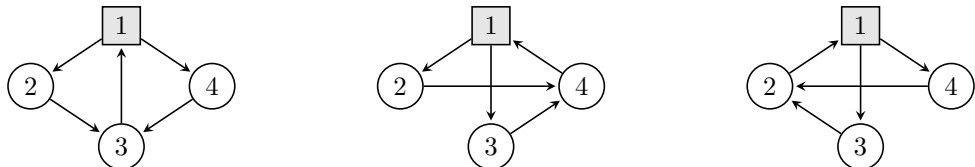

Note that the nice result of Lemma 3 only occurs at the linear non-Gaussian case, where path ranks are one-sided, so that it can be directly dualized to a transversal matroid that can be represented by vectors that lie in the faces of some simplex.

In the linear Gaussian case, with the two-sided path ranks in covariance matrices, there can be more constraints. In that setting, however, rank constraints equivalence do not necessarily imply distributional equivalence: there can be other unmatched equality constraints, e.g., the Pentad, Hexad constraints and beyond (Drton et al., 2007), let alone other inequality constraints.

## C.5 RELATED WORK

**Equivalence characterizations** We first review various approaches to characterize equivalence of causal models. At the same time, we summarize the multiple results developed in this work and situate them within this broader landscape of related literature.

Table 2: A side-by-side overview of representative works on equivalence characterizations across different settings using different approaches. The final column summarizes this work's contributions.

| Settings | | Markov equivalence in fully observed acyclic graphs | Markov equivalence in acyclic graphs with latents | Distributional equivalence in LiNG models with latents and cycles (this work) |
|---|---|---|---|---|
| **Structural** | Level 1 | "Same d-separations" | "Same d-separations" | "Same path/edge ranks up to permutation" (Lemmas 3 and 5) |
| | Level 2 | "Same adjacencies and minimal complexes/ v-structures" (Frydenberg, 1990; Verma & Pearl, 1991) | "Same FCI outputs" (Spirtes & Verma, 1992); "Same MAG adjacencies, v-structures, and colliders on discriminating paths" (Spirtes & Richardson, 1996; Richardson & Spirtes, 2002); "Same head and tails" (Hu & Evans, 2020) | "Same bases in children for $L$ itself and with each singleton $X_i$, up to permutation" (Theorem 2) |
| | Level 3 | "Maximal (deflagged) chain graphs; essential graphs" (Frydenberg, 1990; Andersson et al., 1997; Roverato et al., 2006); "CPDAGs" (Spirtes & Glymour, 1991; Meek, 1995) | "Arrowhead completeness" (Ali et al., 2005); "Full completeness of PAG orientations" (Zhang, 2008a); With background knowledge (still incomplete; (Andrews et al., 2020; Venkateswaran & Perković, 2024)) | "Unique maximal equivalent graph with edges that must always appear, up to cycle reversal" (Theorem 4) |
| **Transfor- mational** | | "Covered edge reversal (Meek conjecture)" (Chickering, 1995; Meek, 1997; Chickering, 2002); "Weakly covered edge reversal" (Markham et al., 2022) for unconditional equivalence | "Covered edge reversal" (Zhang & Spirtes, 2005; Tian, 2005; Ogarrio et al., 2016; Claassen & Bucur, 2022) | "Admissible edge additions/deletions and cycle reversals" (Theorem 3) |
| **Traversal algorithms** | | DAGs traversal within one CPDAG (Meek, 1995; Chickering, 1995; Wienöbst et al., 2023); CPDAGs traversal (Steinsky, 2003; Chen et al., 2016) | MAGs traversal within one PAG (Wang et al., 2024; 2025) | BFS/DFS by admissible transformations (Theorem 3), with additional parallel speedup by column decomposition (Lemmas 9 and 12) |

In general, approaches to characterize equivalence can be categorized into three types:

1. **Structural characterizations,** which provide conditions for determining equivalence between given graphs, and give rise to summary presentations. However, they do not directly lead to equivalence class traversal methods. They can be further stratified by their complexity, informativeness, or purpose, as follows:

- **Level 1.** Graphical conditions necessary and sufficient for determining equivalence, but more as definitions than practical criteria; usually require combinatorial complexities.
- **Level 2.** Practical graphical criteria for determining equivalence; still necessary and sufficient, but more efficient than Level 1.
- **Level 3.** Sound and complete conditions or presentations that summarize the equivalence class. While Level 2 criteria efficiently determine equivalence, they do not fully capture what can be identified; Level 3 addresses this gap. Level 3 can also be used for determining equivalence, but it will be less efficient than Level 2.

2. **Transformational characterizations,** which provide natural ways for traversing the equivalence class, and are useful for developing score-based algorithms. However, as a complement, they are not suited for directly determining equivalence between given graphs, or for developing summary presentations to the equivalence class.

3. **Traversal algorithms,** for enumerating, sampling, or counting elements of the equivalence class, where transformational characterizations are usually helpful.

We present in Table 2 a side-by-side overview of representative prior works and this work across these approaches in different settings. This unified view may help to better understand the contributions of this work, and as well to clarify the methodological implications among these approaches.

There is also a wide range of additional work characterizing equivalences under many other settings, which are not put in Table 2 due to space limit. These include efforts to develop Markov properties and establish Markov equivalence in fully observed models with cycles, such as in linear Gaussian settings (Richardson, 1996; Claassen & Mooij, 2023), discrete settings (Pearl & Dechter, 1996), and general nonlinear settings (Spirtes, 1994; Forré & Mooij, 2017; Mooij & Claassen, 2020), as well as nonlinear settings with latent variables and selection bias (Yao & Mooij, 2025). The distributional equivalence of fully observed linear Gaussian cyclic models has been studied in Ghassami et al. (2020); Drton et al. (2025b). Nonparametric equivalence with latent variables (usually referred to as semi-Markov models) has also been characterized in (Evans, 2018; Markham & Grosse-Wentrup, 2020; Jiang & Aragam, 2023; Richardson et al., 2023), beyond the original CI constraints (Richardson & Spirtes, 2002; Zhang, 2008a). When interventions or multi-domain data are involved, Markov equivalence (typically as CIs in each domain and invariant changes across domains) has been studied extensively in (Tian & Pearl, 2001; Eberhardt et al., 2005; Eberhardt & Scheines, 2007; Eberhardt, 2008; He & Geng, 2008; Hauser & Bühlmann, 2012; 2015; Zhang et al., 2015; Rothenhäusler et al., 2015; Meinshausen et al., 2016; Magliacane et al., 2016; Ghassami et al., 2017; Wang et al., 2017; Yang et al., 2018; Kocaoglu et al., 2019; Huang et al., 2020; Mooij et al., 2020; Squires et al., 2020; Zhang et al., 2023; Dai et al., 2025a; Luo et al., 2025; 2026; Yao et al., 2025), with various focuses spanning across latent variables, selection bias, unknown intervention targets, active experimental design, and so on. From a method view, transformational characterizations have gained increasing attention recently, including (Ghassami et al., 2020; Markham et al., 2022; Wang et al., 2024; 2025; Johnson & Semnani, 2025; Améndola et al., 2025).

Below, we then provide a more comprehensive review of the relevant literature on latent-variable causal discovery, in particular those under the linear non-Gaussian models (Shimizu et al., 2006).

**Parametric settings for latent-variable causal discovery**  A prosperous line of statistical tools beyond conditional independencies have been developed. These include rank constraints (Sullivant et al., 2010; Spirtes et al., 2000) and more general equality constraints (Drton, 2018) in the linear Gaussian setting; and high-order moment constraints (Xie et al., 2020; Adams et al., 2021; Robeva & Seby, 2021; Dai et al., 2022; 2024; Chen et al., 2024a; Xia et al., 2026), which exploit non-Gaussianity for identifiability. In addition to these, matrix decomposition methods (Anandkumar et al., 2013), copula-based constraints (Cui et al., 2018), and mixture oracles (Kivva et al., 2021) were also developed.

**Algorithms for latent-variable causal discovery**  Building on these statistical tools, many latent variable causal discovery algorithms have been proposed. Many of them fall within the constraint-based framework, by using CI tests and algebraic constraints to infer causal relations. Examples include those based on rank or tetrad constraints (Silva et al., 2003; 2006; Silva & Scheines, 2004;

Choi et al., 2011; Kummerfeld & Ramsey, 2016; Huang et al., 2022; Dong et al., 2024; 2025). Recent efforts have also attempted to formalize score-based methods for latent-variable causal discovery (Jabbari et al., 2017; Ng et al., 2024; Dong et al., 2026).

**Linear non-Gaussian models**    Thanks to the strong identifiability results given by OICA, the linear non-Gaussian models have received much attention for causal discovery with latent variables or cycles: (Améndola et al., 2023; Salehkaleybar et al., 2020; Wang & Drton, 2023; Maeda & Shimizu, 2020; Silva & Shimizu, 2017; Dai et al., 2024; Yang et al., 2022; 2024; Shimizu, 2022; Drton et al., 2025a; Liu et al., 2021; Schkoda et al., 2024; Tramontano et al., 2022; Rothenhäusler et al., 2015), together with those discussed in §1, and many more. Beyond structure learning, LiNG models also provide benign conditions for many other tasks, including causal effect identification (Tchetgen Tchetgen et al., 2024; Kivva et al., 2023; Xie et al., 2022; Tramontano et al., 2024; 2025), model selection (Schkoda & Drton, 2025), covariate selection (Zhang & Wiedermann, 2024), experimental design (Sharifian et al., 2025), etc.

Below, we also discuss how the results in this work, especially the edge rank tools and the motivation of a bipartite matching view, may be generalized to other parameter settings.

For the linear Gaussian setting, existing results in the literature can be directly translated into our edge rank language. Unlike the non-Gaussian setting where the mixing matrix is identifiable, in the Gaussian setting, only the covariance matrix is available. The graphical characterization of covariance matrix ranks, known as "trek-separation," has been established by Sullivant et al. (2010). Specifically, the concept of a bottleneck, which we term the "path rank" on the one-sided directed paths, is extended to the bottleneck along the two-sided directed paths, known as "treks". Since the duality between path ranks and edge ranks hold universally in graphs regardless of the parametric setting, the existing characterization on trek-based path ranks can be directly translated into trek-based edge rank language. As for the technical roadmap, one may first note that the non-Gaussian equivalence condition we build in this work is necessary but not sufficient for the Gaussian setting. That is, two graphs that are equivalent in non-Gaussian models are guaranteed to remain equivalent in Gaussian models; however, graphs that are distinguishable under non-Gaussianity may collapse into the same equivalence class under Gaussianity. We see the closing of this gap as the most immediate future direction for extending our current work.

For the discrete setting, our results are likely generalizable as well. Several recent works have explored path ranks in graphs from discrete data (Gu & Xu, 2023; Chen et al., 2024b; 2025; Lyu & Yang, 2025), where the algebraic counterpart becomes the tensor ranks in the contingency table (Teicher, 1967). However, a precise graphical characterization, analogous to "trek-separation" in the Gaussian case above, has yet to be developed. That said, such a characterization is promising, since the linear Gaussian and discrete models behave similarly in many aspects. For example, both are closed under marginalization and conditionalization; both admit a correspondence between Markov and distributional equivalence, in both cyclic and acyclic cases (Geiger & Meek, 1996; Pearl & Dechter, 1996). Motivated by these parallels already noted in literature, we believe our results can also extend to the discrete setting, and will be directly applicable once the corresponding graphical characterization is developed.

For nonlinear or even nonparametric settings, theoretical generalization remains possible. When the model is partially linear and partially nonlinear, low-dimensional bottlenecks in the linear component remain directly observable through covariance ranks (Spirtes, 2013). When the model is fully nonlinear or even nonparametric, there also exists prior results on the identifiability of latent-variable models (Hu, 2008; 2017). Although the techniques differ, the underlying motivation remains closely related to ranks, particularly those in the Jacobian matrix. However, despite the theoretical meaningfulness, the practical estimation and reliable testing of these ranks remain an open challenge. This challenge can be echoed by viewing rank constraints as generalizations of conditional independence constraints. In linear models, conditional independencies correspond to low ranks in the covariance matrix and can be directed tested via Fisher's Z test. In contrast, robust conditional independence tests in nonlinear settings are still under active development (Duong & Nguyen, 2024; Yang et al., 2025).

# D EVALUATION RESULTS

## D.1 QUANTIFYING THE SIZES OF EQUIVALENCE CLASSES

Table 3: For different numbers of vertices $n$ and latent vertices $l$, we report the total number of weakly connected digraphs with $n$ vertices, the subset that are irreducible when the first $l$ vertices are latent (both with and without $L$-isomorphic variants), and the corresponding numbers of distributional equivalence classes they fall into. The final two columns present statistics on how irreducible digraphs (both with and without $L$-isomorphic variants) are distributed among those equivalence classes.

| #Nodes | #All weakly connected digraphs | #Latents | #Irreducible digraphs (with $L$-isomorphic variants) | #Irreducible digraphs (unique up to $L$-relabeling) | #Distributional equivalence classes | Stats of #digraphs per class (with $L$-isomorphic variants) | Stats of #digraphs per class (unique up to $L$-relabeling) |
|---|---|---|---|---|---|---|---|
| 3 | 54 (18 DAGs) | 0 | 54 | 54 | 41 | | |
| | | 1 | 16 | 16 | 4 | | |
| 4 | 3,834 (446 DAGs) | 0 | 3,834 | 3,834 | 1,788 | | |
| | | 1 | 2,000 | 2,000 | 201 | | |
| | | 2 | 896 | 464 | 4 | | |
| 5 | 1,027,080 (26,430 DAGs) | 0 | 1,027,080 | 1,027,080 | 230,436 | | |
| | | 1 | 712,512 | 712,512 | 28,650 | | |
| | | 2 | 480,640 | 242,320 | 783 | | |
| | | 3 | 294,400 | 50,112 | 4 | | |
| 6 | 1,067,308,488 (3,596,762 DAGs) | 0 | 1,067,308,488 | 1,067,308,488 | 90,085,890 | | |
| | | 1 | 868,762,432 | 868,762,432 | 12,608,550 | | |
| | | 2 | 702,697,472 | 352,061,248 | 453,742 | | |
| | | 3 | 560,760,320 | 93,975,744 | 3,297 | | |
| | | 4 | 419,405,824 | 17,646,432 | 4 | | |

## D.2 Assessing glvLiNG Algorithm's Runtime

Table 4: Running time comparison between our glvLiNG algorithm and a mixed integer linear programming (MILP) baseline for constructing digraphs that satisfy the rank constraints of oracle OICA mixing matrices. Ground-truth graphs are generated from the Erdős–Rényi model with total number of vertices $n$ and average in-degree avgdeg, with $\ell$ vertices randomly designated as latent. Each entry reports the mean and standard deviation over 50 models (when completed); empty entries indicate runs that did not finish within 10 minutes. All times are reported in seconds. Experiments were run on an Apple M4 chip.

| $n$ | $\ell$ | avgdeg | MILP | glvLiNG | glvLiNG Phase 1 | glvLiNG Phase 2 |
|---|---|---|---|---|---|---|
| 5 | 1 | 1 | $0.045 \pm 0.013$ | $0.015 \pm 0.005$ | $0.014 \pm 0.005$ | $0.001 \pm 0.000$ |
| | | 3 | $0.112 \pm 0.008$ | $0.020 \pm 0.002$ | $0.019 \pm 0.002$ | $0.001 \pm 0.000$ |
| 7 | 1 | 1 | | $0.101 \pm 0.045$ | $0.098 \pm 0.044$ | $0.002 \pm 0.000$ |
| | | 3 | | $0.169 \pm 0.024$ | $0.165 \pm 0.024$ | $0.004 \pm 0.000$ |
| | 3 | 1 | $37.402 \pm 0.000$ | $0.048 \pm 0.013$ | $0.045 \pm 0.012$ | $0.003 \pm 0.001$ |
| | | 3 | | $0.083 \pm 0.014$ | $0.075 \pm 0.013$ | $0.007 \pm 0.001$ |
| 9 | 1 | 1 | | $0.691 \pm 0.304$ | $0.687 \pm 0.303$ | $0.004 \pm 0.001$ |
| | | 3 | | $1.129 \pm 0.191$ | $1.122 \pm 0.190$ | $0.007 \pm 0.001$ |
| | 3 | 1 | | $0.319 \pm 0.091$ | $0.308 \pm 0.090$ | $0.009 \pm 0.001$ |
| | | 3 | | $0.667 \pm 0.132$ | $0.634 \pm 0.128$ | $0.031 \pm 0.007$ |
| | 5 | 1 | | $0.082 \pm 0.023$ | $0.075 \pm 0.022$ | $0.005 \pm 0.001$ |
| | | 3 | | $0.260 \pm 0.057$ | $0.230 \pm 0.051$ | $0.023 \pm 0.005$ |
| 11 | 1 | 1 | | $3.637 \pm 1.533$ | $3.630 \pm 1.532$ | $0.007 \pm 0.000$ |
| | | 3 | | $7.706 \pm 0.883$ | $7.693 \pm 0.881$ | $0.013 \pm 0.002$ |
| | 3 | 1 | | $2.174 \pm 0.499$ | $2.142 \pm 0.500$ | $0.030 \pm 0.002$ |
| | | 3 | | $4.979 \pm 0.763$ | $4.873 \pm 0.748$ | $0.102 \pm 0.021$ |
| | 5 | 1 | | $0.530 \pm 0.111$ | $0.492 \pm 0.111$ | $0.032 \pm 0.002$ |
| | | 3 | | $2.348 \pm 0.426$ | $2.170 \pm 0.396$ | $0.159 \pm 0.045$ |
| 13 | 1 | 1 | | $22.838 \pm 9.667$ | $22.827 \pm 9.667$ | $0.011 \pm 0.001$ |
| | | 3 | | $38.173 \pm 7.725$ | $38.155 \pm 7.721$ | $0.017 \pm 0.005$ |
| | 3 | 1 | | $12.501 \pm 4.602$ | $12.404 \pm 4.593$ | $0.094 \pm 0.016$ |
| | | 3 | | $23.517 \pm 5.807$ | $23.362 \pm 5.751$ | $0.150 \pm 0.062$ |
| | 5 | 1 | | $4.277 \pm 1.430$ | $4.069 \pm 1.433$ | $0.190 \pm 0.009$ |
| | | 3 | | $13.150 \pm 4.009$ | $12.376 \pm 3.712$ | $0.723 \pm 0.310$ |

D.3  BENCHMARKING EXISTING METHODS UNDER ORACLE INPUTS

Table 5: Evaluation of existing methods under possible model misspecification on arbitrary latent-variable models. Ground-truth graphs are generated from the Erdős-Rényi model with total number of vertices $n$ and average in-degree avgdeg, with $\ell$ vertices randomly designated as latent. Only irreducible models are chosen. Each entry reports the mean and standard deviation of the structural Hamming distances (SHDs) between the result and truth over 50 random models.

Algorithms are provided with their oracle tests, that is, for them to directly query oracle generalized independent noise (GIN) conditions from the digraph. When the number of their identified latent variables is fewer than truth, we simply add isolated latent variables into the result. When the identified number of latents is larger (which seems not happened), we planned to choose the removal that leads to best result. Finally, the best possible result is reported, i.e., we choose the digraph in the ground-truth equivalence class that is closer to their output as the truth. The latent variables are viewed as unlabeled.

| $n$ | $\ell$ | avgdeg | PO-LiNGAM | LaHiCaSl |
|---|---|---|---|---|
| 10 | 3 | 1 | $15.48 \pm 2.75$ | $31.64 \pm 4.62$ |
|  |  | 2 | $24.30 \pm 4.41$ | $36.80 \pm 4.22$ |
|  |  | 3 | $35.40 \pm 3.61$ | $39.96 \pm 4.88$ |
|  |  | 4 | $45.22 \pm 3.53$ | $41.04 \pm 3.81$ |
|  | 5 | 1 | $18.04 \pm 3.99$ | $32.36 \pm 4.38$ |
|  |  | 2 | $28.44 \pm 3.48$ | $36.68 \pm 4.31$ |
|  |  | 3 | $39.18 \pm 3.60$ | $40.42 \pm 4.40$ |
|  |  | 4 | $50.00 \pm 3.45$ | $41.00 \pm 4.57$ |
| 15 | 3 | 1 | $31.10 \pm 4.87$ | $74.22 \pm 8.04$ |
|  |  | 2 | $48.26 \pm 6.67$ | $76.80 \pm 6.66$ |
|  |  | 3 | $64.12 \pm 5.59$ | $81.64 \pm 8.28$ |
|  |  | 4 | $84.60 \pm 5.73$ | $85.56 \pm 8.61$ |
|  | 5 | 1 | $35.02 \pm 5.74$ | $71.70 \pm 6.94$ |
|  |  | 2 | $54.84 \pm 6.04$ | $78.44 \pm 7.01$ |
|  |  | 3 | $72.96 \pm 6.62$ | $79.98 \pm 8.15$ |
|  |  | 4 | $92.28 \pm 7.71$ | $82.50 \pm 7.35$ |
|  | 7 | 1 | $36.44 \pm 5.63$ | $71.34 \pm 8.37$ |
|  |  | 2 | $58.00 \pm 6.24$ | $77.90 \pm 8.25$ |
|  |  | 3 | $79.90 \pm 7.34$ | $79.16 \pm 8.79$ |
|  |  | 4 | $101.04 \pm 5.90$ | $84.56 \pm 8.99$ |
| 20 | 3 | 1 | $48.60 \pm 6.12$ | $129.88 \pm 14.18$ |
|  |  | 2 | $76.92 \pm 8.29$ | $136.36 \pm 13.05$ |
|  |  | 3 | $103.04 \pm 8.30$ | $138.24 \pm 11.98$ |
|  |  | 4 | $129.14 \pm 10.46$ | $146.86 \pm 13.02$ |
|  | 5 | 1 | $54.04 \pm 5.23$ | $122.78 \pm 12.67$ |
|  |  | 2 | $84.10 \pm 8.44$ | $136.04 \pm 12.85$ |
|  |  | 3 | $115.72 \pm 8.72$ | $139.76 \pm 13.64$ |
|  |  | 4 | $146.44 \pm 9.10$ | $141.64 \pm 10.75$ |
|  | 7 | 1 | $58.70 \pm 6.36$ | $128.46 \pm 11.96$ |
|  |  | 2 | $92.86 \pm 8.41$ | $132.48 \pm 12.36$ |
|  |  | 3 | $124.00 \pm 9.60$ | $140.76 \pm 12.31$ |
|  |  | 4 | $155.48 \pm 7.83$ | $143.76 \pm 11.55$ |
|  | 9 | 1 | $64.40 \pm 7.38$ | $120.08 \pm 12.43$ |
|  |  | 2 | $98.04 \pm 10.43$ | $135.24 \pm 12.75$ |
|  |  | 3 | $134.16 \pm 10.34$ | $137.52 \pm 12.30$ |
|  |  | 4 | $167.86 \pm 11.28$ | $142.76 \pm 11.39$ |

### D.4 Evaluating glvLiNG's Performance with Existing Methods in Simulations

**OICA estimation part**    We first describe how we handle the OICA estimation part.

For our choice of OICA implementation, we have tried multiple options and find that overall, the MATLAB implementation[2] of SDP-ICA (Podosinnikova et al., 2019) tends to provide best estimated mixing matrices across multiple settings. We thus adopt it in our experiments.

For the number of latent variables, although theoretically identifiable, existing OICA implementations still require specifying this number as an input. Hence, following the common practice as in (Salehkaleybar et al., 2020), we test multiple candidate values and select the one minimizing the loss on a held out set.

**Handling empirical ranks in an OICA matrix**    We then explain how we process the mixing matrix estimated from OICA. Having obtained an OICA-estimated mixing matrix, the core task of glvLiNG becomes constructing a bipartite graph to realize the rank patterns in this mixing matrix, which define a transversal matroid. When OICA is not an oracle, these empirical ranks may violate matroid axioms, just like how conditional independencies in data may violate a graphoid in nonparametric settings.

To address this, in our implementation, we assign a "full-rank confidence score" to each relevant block of $A$. Specifically, let $\sigma_{\min}$ be $A$'s minimum singular value, we use the score $\frac{1}{1+\exp(-\alpha(\sigma_{\min}-\epsilon))}$, and in experiments, we set $\alpha = 25$ and $\epsilon = 0.02$. Then, in phase 1 (recovering latent outgoing edges), we approximate the closest valid transversal matroid that maximizes agreement with these scores. In phase 2 (recovering observed outgoing edges), for efficiency we simply threshold these scores to determine each variable's outgoing edges independently. We have simulated and verified that this procedure is robust to moderately noisy ranks, by e.g., assigning true full-rank blocks scores from $\mathcal{N}(0.75, 0.2)$ and others from $\mathcal{N}(0.25, 0.2)$, both $0, 1$ truncated.

**Simulation setup**    In simulation, we compare glvLiNG with existing methods including LaHiCaSl[3] (Xie et al., 2024) and PO-LiNGAM[4] (Jin et al., 2024). We generate random Erdős-Rényi model with total number of vertices $n$ from 5 to 13, number of latent variables $\ell$ from 1 to 5, average in-degree $d$ of 1 and 3, and sample size $N$ from $1,000$ to $200,000$. We sample data with linear causal weights uniformly from $[-2.5, -0.5] \cup [0.5, 2.5]$, and exogenous noise are sampled from a uniform distribution $[-0.5, 0.5]$, following (Podosinnikova et al., 2019). We calculate the minimum SHD between all graphs in the true equivalence class to the discovery output graph as the SHD result.

**Simulation results**    The results are presented in Figure 7. From it we have the following observations:

First of all, it is not surprising to see that LaHiCaSl and PO-LiNGAM perform better when the graph is sparser. For example, when $d = 1$, these two methods perform better than glvLiNG, though the difference remains modest. This is perhaps because, when the graph is sparser, maintaining irreducibility typically means more edges outgoing from latent variables, while edges from observed ones to others are fewer. This aligns well with the model assumptions of these two methods. For example, LaHiCaSl assumes a hierarchical latent-variable model in which all observed variables are leaf nodes. Given this additional benefit from their sparsity constraints, and the fact that both LaHiCaSl and PO-LiNGAM estimates ranks using the GIN condition which is more efficient than OICA, it is not surprising to see that they perform better in this setting.

However, when the graph is denser, glvLiNG performs particularly better. For example, when $d = 3$, glvLiNG consistently outperforms the other two methods, and the difference is considerable. This is perhaps because, when the graph is denser, more complex structures become common, including arbitrary edges between latent and observed variables, as well as cycles. Model assumptions of existing methods are more likely to be violated, making them less effective at recovering these structures. In contrast, with a structural-assumption-free design, glvLiNG avoids such model misspecification, and still allows the recovery of these structures.

---

[2]https://github.com/gilgarmish/oica
[3]https://github.com/jinshi201/LaHiCaSl
[4]https://github.com/Songyao-Jin/PO-LiNGAM

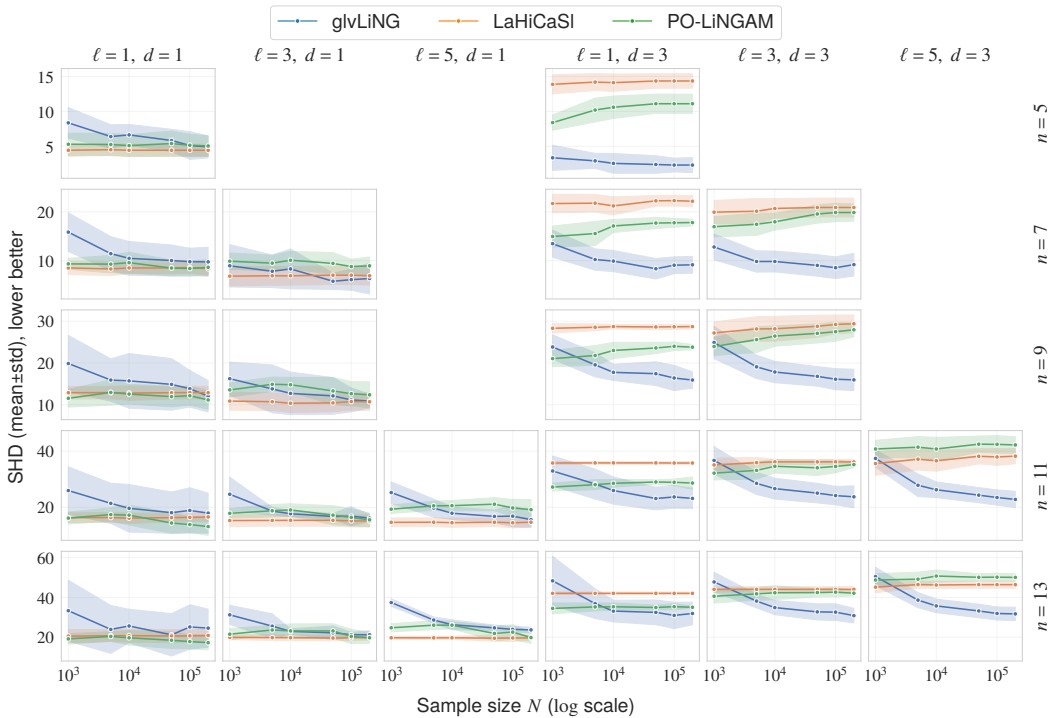

Figure 7: Simulation results comparing glvLiNG with existing methods with varying sample size $N$ (the global x-axis), and each subplot shows a setting under a specific number of total variables $n$, number of latent variables $\ell$, and the average in-degree $d$. Mean and standard deviation of SHD are calculated from 25 random irreducible models.

We also observe that glvLiNG tends to be more robust to latent dimensionality. For example, when $n = 13$ and $d = 3$, increasing the number of latents $\ell$ from 1 to 5, the average SHD of glvLiNG increases from 33.1 to 35.7, while other method, such as PO-LiNGAM, increases from 35.3 to 50.7. This is perhaps because glvLiNG jointly recovers all latent-outgoing edges at once, using an OICA mixing matrix whose dimensionality is already fixed. In contrast, the other two methods requires incrementally clustering and adding latent variables.

## D.5 ANALYZING A REAL-WORLD DATASET WITH GLVLING ALGORITHM

For the real-world experiment, we use a Hong Kong stock market dataset that involves the daily dividend/split-adjusted closing prices for 14 major stocks from January 4, 2000 to June 17, 2005 (1331 samples). These 14 stocks represent the dominant sectors of the market: 3 of them are on banking (HSBC Holdings, Hang Seng Bank, Bank of East Asia), 5 on real estate (Cheung Kong, Henderson Land, Hang Lung Properties, Sun Hung Kai Properties, Wharf Holdings), 3 on utilities (CLP Holdings, HK & China Gas, HK Electric), and 3 on commerce (Hutchison, Swire Pacific 'A', Cathay Pacific Airways). All of them were constituents of Hang Seng Index (HSI), and they were almost the largest companies of the Hong Kong stock market at the time.

By applying glvLiNG on this dataset, we recovered an equivalence class of causal graphs containing 2 latent variables. The presentation (see Appendix C.3) of this equivalence class is shown in Figure 8. Here is a summary: the class consists of 19,008 causal graphs with 16=14+2 vertices, and among them the numbers of edges range between 29 to 34. In the presentation, there are 20 "solid" (must appear) and 14 "dashed" (may appear) edges.

This result suggests several interesting observations, as follows:

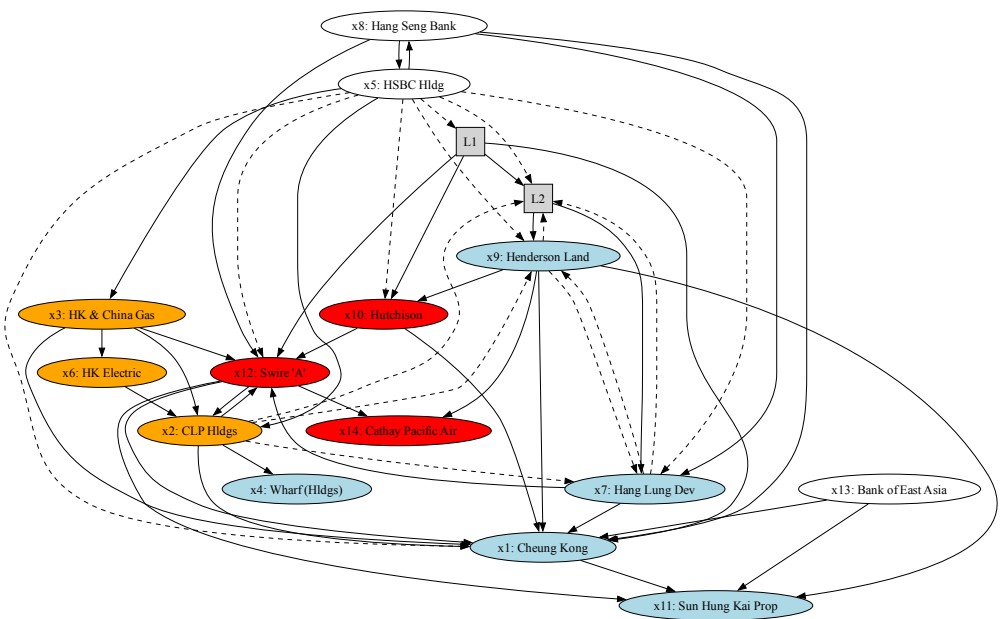

Figure 8: Presentation of the equivalence class that glvLiNG estimates from the stock market data. Different colors of nodes indicate different sectors. Solid and dashed edges indicate edges that must appear in all or at least one equivalent graph(s).

1. Large banks seem to be major upstream causes. For example, the two largest banks, HSBC Holdings and Hang Seng Bank, together form a 2-cycle that has 9 children across sectors, but there are no edges into them.

2. Real estates, in contrast, seem to be downstream effect receivers. For example, Cheung Kong has 10 parents, but only 1 edge pointing out from it.

3. Utilities are heavily involved in cycles. For example, among 17 simple cycles in the graph, CLP Holdings belongs to 11 of them. These cycles are often across sectors as utilities - real estate - commerce - utilities.

4. One latent variable seems interpretable. It has one parent HSBC Holdings, and three children (all with solid edges): Cheung Kong, Hutchison, and Swire Pacific 'A'. Among them, Cheung Kong and Hutchison were two core holdings of a same group.

5. Stocks under the same sector tend to be connected more closely.

