# OpenReview forum: "Distributional Equivalence in Linear Non-Gaussian Latent-Variable Cyclic Causal Models: Characterization and Learning"
_ICLR.cc/2026/Conference — ICLR 2026 Oral_

### Official Review · Reviewer_Czvx · 2025-10-27

**Soundness:** 4
**Presentation:** 4
**Contribution:** 4
**Rating:** 8
**Confidence:** 4

**Summary:**

In this paper, the authors propose a characterization of equivalence classes of causal graphs with both latent variables and cycles. To this end, they propose a novel graphical tool of edge ranks (which is dual to path ranks), which makes results cleaner and more intuitive than path ranks do. They further provide a straightforward algorithm to traverse this set of distributionally equivalent causal models.

**Strengths:**

1. The classification of equivalence classes of causal models with cycles and latent variables is an important topic.
2. Edge ranks seem to be a very useful tool for future results.
3. Despite the complexity of the topic, the paper is clearly written and examples are well-chosen.

**Weaknesses:**

1. It would be nice to have a little more description of the algorithm in the main paper, especially the decomposition of global rank constraints into local rank queries.
2. It is unclear how this method works when we do not have an OICA oracle, and in particular when we don't know the number of latent variables.

**Questions:**

1. Your paper currently focuses on minimal DGs, where minimality is measured in terms of numbers of variables. However, one might also be interested in minimally cyclic graphs, at the cost of more latent factors. Is there any way to establish results to determine equivalence classes of such graphs?
2. Your paper currently makes LiNG assumptions. However, low-dimensional bottlenecks should still be noticeable in linear Gaussian, or in nonlinear settings. What prevents us from generalizing the approach to these cases?
3. Given this classification of equivalence classes, is there any hope (and need) for a GES-like algorithm to learn the (equivalence class of) the causal graph? Would this provide any benefit?
4. Would it be possible to explain more about why certain graphs are not equivalent? For example, in the example in section C.2, Figure 5, one could think that the graph $G_1$, but with the edge $X_3 \rightarrow X_1$ removed (equivalently, $G_10$ with the edge $X_3 \rightarrow X_2$ removed) might lie in the same equivalence class.

As a side note, it appears that your sections D.1-D.3 and their contents are currently struggling a little with formatting.

---

> ### Author Response · Authors · 2025-11-22
> **Response to Reviewer Czvx - Q1**
>
> We sincerely appreciate the reviewer's constructive comments and insightful feedback. Please see below for our response.
>
> ---
>
>
> **(Q1)** The reviewer suggests including more description of the algorithm in the main text.
>
> **A:** Thank you for this helpful suggestion. In our original submission, we had to defer much of the algorithm detail to the Appendix due to page limit. In light of your suggestion, we have updated the manuscript to include **a more detailed description** of the algorithm in the main text.
>
> Specifically, we now highlight the efficient decomposition of global rank constraints into local rank queries by raising this result to **a standalone Theorem 2.** It is worth noting that beyond its use in the algorithm, this theorem also serves as an efficient criterion for determining equivalence between given graphs, in analogous to the "same adjacencies and v-structures" in the classical Markov equivalence setting for fully observed acyclic models.
>
> We hope the reviewer will find the updated manuscript more informative in this regard.

---

> ### Author Response · Authors · 2025-11-22
> **Response to Reviewer Czvx - Q2 (Part 1/3)**
>
> **(Q2)** The reviewer wonders how the algorithm works when we do not have an OICA oracle.
>
> **A:** Thank you for raising this point. Indeed, we do see the use of OICA as a major practical limitation of our glvLiNG algorithm, as noted in Section 5. In light of your comment, we have managed to **add a new empirical study** to evaluate glvLiNG's performance without an OICA oracle, in the finite sample settings with sample sizes varying between $1,000$ and $200,000$. We provide an overview of this study in what follows.
>
>
> **I. Let us first describe how we handle the OICA estimation part:**
>   - For our choice of OICA implementation, we have tried multiple options and find that overall, the [MATLAB implementation](https://github.com/gilgarmish/oica) of SDP-ICA (Podosinnikova et al., 2020) tends to provide best estimated mixing matrices across multiple settings. We thus adopt it in our experiments.
>   - For the number of latent variables, although theoretically identifiable, existing OICA implementations still require specifying this number as an input. Hence, following the common practice, we test multiple candidate values and select the one minimizing the loss.
>
>
> **II. We then explain how we process the mixing matrix estimated from OICA:**
>    - Having obtained an OICA-estimated mixing matrix, the core task of glvLiNG becomes constructing a bipartite graph to realize the rank patterns in this mixing matrix, which define a transversal matroid.
>    - When OICA is not an oracle, these empirical ranks may **violate matroid axioms,** just like how conditional independencies in data may violate a "graphoid" in nonparametric settings.
>    - To address this, in our implementation, we assign a "full-rank confidence score" to each relevant block of $A$. Then,
>      - In phase 1 (recovering latent outgoing edges), we **approximate the closest valid** transversal matroid that maximizes agreement with these scores.
>      - In phase 2 (recovering observed outgoing edges), for efficiency we simply threshold these scores to determine each variable's outgoing edges independently.
>    - We have simulated and verified that this procedure is robust to moderately noisy ranks, by e.g., assigning true full-rank blocks scores from $\mathcal{N}(0.75, 0.2)$ and others from $\mathcal{N}(0.25, 0.2)$, both $0,1$ truncated.
>
>
> **III. Next, we summarize the simulation setup and key results:**
>
>
> In simulation, we compare glvLiNG with existing methods including LaHiCaSl (Xie et al., 2024) and PO-LiNGAM (Jin et al., 2024). We generate random Erdős–Rényi irreducible graphs and varied the total number of variables $n$ from $5$ to $13$, number of latent variables $\ell$ from $1$ to $5$, average in-degree $d$ of $1$ and $3$, and sample size $N$ from $1,000$ to $200,000$. Full results and details are provided in Appendix D.4 of the updated manuscript.
>
> Below, we report the structural hamming distance (SHD; in this setting, we use the lowest SHD between the algorithm's output and all graphs in the true equivalence class) results for $N=10,000$, and discuss the effect of sample size.
>
> + SHDs (lower is better) when the average in-degree $d=1$:
>
>   |$n$|5|7||9||11|||13|||
>   |-|-|-|-|-|-|-|-|-|-|-|-|
>   |$\ell$|**1**|**1**|**3**|**1**|**3**|**1**|**3**|**5**|**1**|**3**|**5**|
>    |**glvLiNG**|6.6±1.5|10.5±3.4|8.3±4.0|15.7±6.4|12.7±5.0|19.6±8.3|17.7±1.0|17.9±2.2|25.6±8.2|22.9±1.2|26.3±2.4|
>   |**LaHiCaSl**|**4.4±0.9**|**8.5±1.2**|**6.9±1.9**|12.8±1.7|**10.4±2.2**|**16.1±1.8**|**15.4±2.1**|**14.5±1.6**|20.8±3.2|**19.8±1.4**|**19.7±0.8**|
>   |**PO-LiNGAM**|5.1±1.5|9.6±2.1|10.1±1.8|**12.6±2.3**|14.8±1.7|17.2±3.4|19.1±2.2|20.6±1.8|**19.7±3.6**|23.2±3.4|26.1±1.7|
>
> + SHDs (lower is better) when the average in-degree $d=3$:
>
>   |$n$|5|7||9||11|||13|||
>   |-|-|-|-|-|-|-|-|-|-|-|-|
>   |$\ell$|**1**|**1**|**3**|**1**|**3**|**1**|**3**|**5**|**1**|**3**|**5**|
>   |**glvLiNG**|**2.6±1.4**|**9.9±2.1**|**9.8±2.1**|**17.8±1.9**|**17.8±2.5**|**26.0±4.7**|**26.6±3.6**|**26.2±2.7**|**33.1±5.3**|**34.9±3.5**|**35.7±3.4**|
>   |**LaHiCaSl**|14.1±1.2|21.2±1.8|20.7±2.0|28.7±0.5|28.2±2.9|35.8±0.5|36.2±0.9|36.6±3.7|42.0±0.7|44.0±1.3|46.2±1.7|
>   |**PO-LiNGAM**|10.6±1.6|17.1±1.3|17.9±1.7|23.0±1.9|26.4±2.4|28.5±1.3|34.6±2.3|40.8±4.0|35.3±1.8|42.3±3.4|50.7±2.9|

---

> ### Author Response · Authors · 2025-11-22
> **Response to Reviewer Czvx - Q2 (Part 2/3)**
>
> The results suggest the following observations:
>
> 1. **First of all, it is not surprising to see that LaHiCaSl and PO-LiNGAM perform better when the graph is sparser.**
>    - For example, when $d=1$, these two methods perform better than glvLiNG, though the difference remains modest.
>    - This is perhaps because, when the graph is sparser, maintaining irreducibility typically means more edges outgoing from latent variables, while edges from observed ones to others are fewer.
>    - This aligns well with the model assumptions of these two methods. For example, LaHiCaSl assumes a hierarchical latent-variable model in which all observed variables are leaf nodes.
>    - Given this additional benefit from their sparsity constraints, and the fact that both LaHiCaSl and PO-LiNGAM estimates ranks using the GIN condition which is more efficient than OICA, it is not surprising to see that they perform better in this setting.
> 2. **However, when the graph is denser, glvLiNG performs particularly better.**
>    - For example, when $d=3$, glvLiNG consistently outperforms the other two methods, and the difference is considerable.
>    - This is perhaps because, when the graph is denser, more complex structures become common, including arbitrary edges between latent and observed variables, as well as cycles.
>    - Model assumptions of existing methods are more likely to be violated, making them less effective at recovering these structures.
>    - In contrast, with a structural-assumption-free design, glvLiNG avoids such model misspecification, and still allows the recovery of these structures.
> 3. **glvLiNG is also more robust to latent dimensionality.**
>    - For example, when $n=13$ and $d=3$, increasing the number of latents $\ell$ from $1$ to $5$, the average SHD of glvLiNG increases from 33.1 to 35.7, while other method, such as PO-LiNGAM, increases from 35.3 to 50.7.
>    - This is perhaps because glvLiNG jointly recovers all latent-outgoing edges at once, using an OICA mixing matrix whose dimensionality is already fixed.
>    - In contrast, the other two methods requires incrementally clustering and adding latent variables.
>
>
> Certainly, we fully acknowledge that a sample size of $N=10,000$ is relatively large, and as expected given OICA's sensitivity to sample size, glvLiNG performs much worse with smaller sample sizes such as $N=1,000$. That said, we still see an overall trend similar to the observations above. **Full results across different sample sizes** are provided in Figure 7 of Appendix D.4 in the updated manuscript.

---

> ### Author Response · Authors · 2025-11-22
> **Response to Reviewer Czvx - Q2 (Part 3/3)**
>
> **IV. We then demonstrate results on a real-world dataset:**
>
>
> For the real-world experiment, we use a Hong Kong stock market dataset that involves the daily dividend/split-adjusted closing prices for 14 major stocks from January 4, 2000 to June 17, 2005 (1331 samples). These 14 stocks represent the dominant sectors of the market:
>
> - 3 of them are on **banking** (HSBC Holdings, Hang Seng Bank, Bank of East Asia),
> - 5 on **real estate** (Cheung Kong, Henderson Land, Hang Lung Properties, Sun Hung Kai Properties, Wharf Holdings),
> - 3 on **utilities** (CLP Holdings, HK & China Gas, HK Electric), and
> - 3 on **commerce** (Hutchison, Swire Pacific 'A, Cathay Pacific Airways).
>
> All of them were constituents of Hang Seng Index (HSI), and they were almost the largest companies of the Hong Kong stock market at the time.
>
> By applying glvLiNG on this dataset, we recovered an equivalence class of causal graphs containing 2 latent variables. The CPDAG-like presentation (new result; see Theorem 4 in the updated manuscript) of this equivalence class is shown in **Figure 8 (Appendix D.5)**. Here is a summary:
>
> - The class consists of 19,008 causal graphs with 16=14+2 vertices, and among them the numbers of edges range between 29 to 34.
> - In the presentation, there are 20 "solid" edges, i.e., edges that must appear in all equivalent graphs, and 14 "dashed" edges, i.e., edges that appear at least in one equivalent graph.
>
> This result suggests several interesting observations, as follows:
>
> 1. Large banks seem to be major upstream causes. For example, the two largest banks, HSBC Holdings and Hang Seng Bank, together form a 2-cycle that has 9 children across sectors, but there are no edges into them.
> 2. Real estates, in contrast, seem to be downstream effect receivers. For example, Cheung Kong has 10 parents, but only 1 edge pointing out from it.
> 3. Utilities are heavily involved in cycles. For example, among 17 simple cycles in the graph, CLP Holdings belongs to 11 of them. These cycles are often across sectors as utilities - real estate - commerce - utilities.
> 4. One latent variable seems interpretable. It has one parent HSBC Holdings, and three children (all with solid edges): Cheung Kong, Hutchison, and Swire Pacific 'A'. Among them, Cheung Kong and Hutchison were two core holdings of a same group.
> 5. Stocks under the same sector tend to be connected more closely.
>
> More details on the above empirical study are included in Appendices D.4 and D.5 of the updated manuscript.
>
>
> **V. Finally, we offer some remarks on our algorithm and equivalence characterization beyond OICA:**
>
> 1. **There are promising directions for developing OICA-free algorithms.** Note that we do not need full parameters of the mixing matrix, but only its rank structure. For this purpose, several existing techniques can already allow partial access to rank information without explicitly solving OICA, such as cumulant-based methods (Robeva & Seby, 2021), independent subspace analyses (ISA; Dai et al 2022), and generalized independent noise condition (GIN; Xie et al., 2024). It would be interesting to see how these can be incorporated into glvLiNG in future work.
> 2. **Our core contribution lies in the theoretical characterization of equivalence.** In analogy with Markov equivalence in fully observed acyclic models, **this work has provided a comprehensive suite of results,** each targeting a different perspective of the problem, as follows:
>    1. **A definitional characterization of equivalence,** like "same d-separations," expressed in dual languages of path and edge ranks (Lemmas 3 and 5);
>    2. **An efficient graphical criterion for determining equivalence,** like "same adjacencies and v-structures," where importantly, one only need to check each single observed variable, instead of all subsets of them (Theorem 2);
>    3. **A transformational characterization** that allows for equivalence class traversal, like "covered edge reversal (Meek conjecture)" (Theorem 3), and
>    4. **An interpretable presentation of the equivalence class** that fully captures what is identifiable, like a "CPDAG" (Theorem 4; new result during the discussion phase; for more details, please kindly refer to our response to Reviewer FDU9's Q1).
>
>     As for the glvLiNG algorithm, instead of a main focus, it is intended more as a first baseline to show that latent-variable causal discovery is indeed possible without any structural assumptions.
>
>
> We hope the above helps address the reviewer's question, and we thank the reviewer again for helping improve the completeness of our submission.

---

> ### Author Response · Authors · 2025-11-22
> **Response to Reviewer Czvx - Q3**
>
> **(Q3)** The reviewer suggests also considering "minimally cyclic" graphs, even with potentially more latent variables.
>
> **A:** We appreciate this forward-looking suggestion.
>
> In the specific linear non-Gaussian setting we consider in this work, **fortunately, such a trade-off does not arise:**
>   - We show that for any reducible graph, applying the reduction procedure in Proposition 2 yields an irreducible graph that not only has fewer latent variables, but also guarantees **no increase in either the number of cycles or the number of edges.**
>   - In this sense, our reduction can be viewed as a "safe canonicalization," one that both preserves (or even reduces; see Figure 1 in our manuscript for example) cyclicity and minimizes latent dimensionality.
>   - Thanks to your comment, we have made this point explicit in the updated manuscript.
>
> That said, in more general settings, this trade-off suggested by the reviewer indeed arises, and **becomes an important subtlety:**
>    - Consider a classical nonparametric example with four observed variables $X_1, X_2, X_3, X_4$, among which the only (conditional) independencies are $X_1 \perp X_4 | X_2, X_3$ and $X_2 \perp X_3 | X_1, X_4$. No fully observed acyclic model can generate such data.
>    - One might try to explain this data using a fully observed model with a 4-cycle, e.g., $X_1 \to X_3 \to X_4 \to X_2 \to X_1$.
>    - Alternatively, one could also choose to believe the presence of selection bias rather than cycles, e.g., by replacing the edge between $X_2$ and $X_4$ in the 4-cycle with a hidden selection $X_2 \to S \leftarrow X_4$.
>    - In such cases, both models are plausible, and it would be inappropriate to claim either one as more "plausible/irreducible". **Both alternatives should be presented,** with the choice left to users.
>
> We thank the reviewer again for this insightful point, which helps guide the design of future work.

---

> ### Author Response · Authors · 2025-11-22
> **Response to Reviewer Czvx - Q4**
>
> **(Q4)** The reviewer wonders whether our results on "low-dimensional bottlenecks" can generalize beyond the linear non-Gaussian setting.
>
> **A:** We appreciate this insightful question. **The answer is yes,** and this is exactly one of our main contributions in introducing edge ranks, a new tool that fills a missing piece in the rank-based picture and thus expands the toolbox for latent-variable causal discovery.
>
> In what follows, we outline how this new tool may help generalize our results to different parametric settings, and discuss the specific remaining challenges in each setting.
>
> 1. **For the linear Gaussian setting, existing results in the literature can be directly translated into our edge rank language.**
>     - Unlike the non-Gaussian setting where the mixing matrix is identifiable, in the Gaussian setting, only the covariance matrix is available.
>     - The graphical characterization of covariance matrix ranks, known as "trek-separation," has been established by Sullivant et al., (2010). Specifically, the concept of a bottleneck, which we term the "path rank" on the one-sided directed paths, is extended to the bottleneck along the two-sided directed paths, known as **"treks"**.
>     - The condition for when two latent-variable models entail the same covariance ranks is yet to be developed (and known to be difficult). This is where edge ranks might help: since the duality between path ranks and edge ranks hold universally in graphs, the existing characterization on trek-based path ranks can be directly translated into trek-based edge rank language.
> 	- **This alternative is valuable:** it allows one to work with more local, edge-based graphical constraints and, algebraically, to use simpler binary adjacency matrices rather than weighted covariance matrices.
> 	- As for the technical roadmap, one may first note that the non-Gaussian equivalence condition we build in this work is necessary but not sufficient for the Gaussian setting. That is, two graphs that are equivalent in non‑Gaussian models are guaranteed to remain equivalent in Gaussian models; however, graphs that are distinguishable under non‑Gaussianity may collapse into the same equivalence class under Gaussianity.
> 	- We see the closing of this gap as the most immediate future direction from our current one.
>
> 2. **For the discrete setting, our results are likely generalizable as well.**
>     - Several recent works have explored path ranks in graphs from discrete data (Gu & Xu 2023; Chen et al., 2024), where the algebraic counterpart becomes the **tensor ranks in the contingency table.**
>     - However, a precise graphical characterization, analogous to "trek-sepration" in the Gaussian case above, has yet to be developed.
>     - That said, such a characterization is promising, since the linear Gaussian and discrete models behave similarly in many aspects.
>     - For example, both are closed under marginalization and conditionalization; both admit a correspondence between Markov and distributional equivalence, in both cyclic and acyclic cases (Geiger & Meek, 1996; Pearl & Dechter, 1996).
>     - Motivated by these parallels already noted in literature, we believe our results can also extend to the discrete setting, and will be directly applicable once the corresponding graphical characterization is developed.
>
> 3. **For nonlinear or even nonparametric settings, theoretical generalization remains possible.**
>    - When the model is partially linear and partially nonlinear, low-dimensional bottlenecks in the linear component remain directly observable through covariance ranks (Spirtes, 2013).
>    - When the model is fully nonlinear or even nonparametric, there also exists prior results on the identifiability of latent-variable models (Hu, 2008 & 2017). Although the techniques differ, the underlying motivation remains closely related to ranks, particularly those in the **Jacobian matrix.**
>    - However, despite the theoretical meaningfulness, the practical estimation and reliable testing of these ranks remain an open challenge.
>    - This challenge can be echoed by viewing rank constraints as generalizations of conditional independence constraints. In linear models, conditional independencies correspond to low ranks in the covariance matrix and can be directed tested via Fisher's Z test. In contrast, robust conditional independence tests in nonlinear settings are still under active development (Duong & Nguyen, 2024; Yang et al., 2025).
>
> In summary, while this work itself focuses on the linear non-Gaussian setting as a starting point, **the tools and intuitions we develop are broadly generalizable.** Just as the original LiNGAM (Shimizu et al., 2006) inspired a range of general causal discovery methods in the fully observed case, including the well-known additive noise models (Hoyer et al., 2008) and post-nonlinear models (Zhang & Hyvärinen, 2010), we hope this work could as well provide some groundwork for latent-variable causal discovery in more settings.

---

> ### Author Response · Authors · 2025-11-22
> **Response to Reviewer Czvx - Q5 & Q6**
>
> **(Q5)** The reviewer wonders whether a GES-like algorithm can, or should, be developed.
>
> **A:** We thank the reviewer again for an insightful question. **The answers to both are yes.** Specifically,
>
> 1. **A score-based algorithm is needed:**
>    - As we note in our response to your Q2, empirical OICA may produce ranks that violate matroid axioms, just as empirical conditional independencies can violate graphoid axioms in nonparametric settings.
>    - While our current implementation of glvLiNG uses some heuristics to handle it, there is no guarantee for a robust procedure.
>    - It still faces issues like **order dependence, error propagation, and thresholding sensitivity,** much like the PC algorithm.
>    - Note that in the nonparametric setting, these were effectively mitigated by score-based methods such as GES.
>    - Similarly, properly designed score-based future methods would also offer greater practical benefits than our current glvLiNG algorithm.
>
> 2. **A score-based algorithm is also promising to emerge from our results:**
>    - In history, transformational characterizations of equivalence have played a key role in developing score-based algorithms.
>    - For example, the "covered edge reversal (Meek conjecture)" criterion (Chickering, 1995; Meek, 1997) forms the direct foundation for GES (Chickering, 2002).
>    - Its generalization under the FCI setting (Zhang & Spirtes, 2005; Tian, 2005) again led to algorithms like GFCI (Ogarrio et al., 2016).
>    - In our work, we provide a transformation characterization (Theorem 3) similar to "Meek conjecture".
>    - Hence, we have the reason to believe that general GES-like latent-variable discovery algorithms **may also emerge from this work's results.**
>
> We appreciate the reviewer for highlighting this valuable direction.
>
>
>
> ---
>
> **(Q6)** The reviewer asks for more explanation about why certain graphs are not equivalent, particularly in an example shown in Figure 5 of Section C.2.
>
> **A:** We thank the reviewer for the careful reading.
>
> Let us use $\mathcal{G}$ to denote the graph $\mathcal{G}_1$ in Section C.2, Figure 5, which contains one latent variable and three observed variables, with the following five edges: $L\to X_1$, $L\to X_3$, $X_2\to L$, $X_3 \to X_1$, and $X_3 \to X_2$. Let $\mathcal{H}$ denote the same graph but with the edge $X_3 \to X_1$ removed.
>
> To see why $\mathcal{G}$ and $\mathcal{H}$ are **not** equivalent, i.e., why the edge $X_3 \to X_1$ **cannot be removed from** $\mathcal{G}$, or equivalently, why this edge **cannot be added into** $\mathcal{H}$, let us carefully examine the "admissible edge addition/deletion" criterion in Lemma 7.
>
> According to Lemma 7, to add an edge "tail to head", we need to check the matchings from "latent variables excluding this tail" to "this tail's current nonchildren," which of course include the "head". Applying this to graph $\mathcal{H}$, to consider whether we may add the edge $X_3 \to X_1$, it is to check the matchings from $\{L\}$ to $\{L, X_1\}$.
>
> We observe that there is a full matching of size $1$, where $L$ can be matched either to $X_1$ or $L$ itself via existing edges in $\mathcal{H}$. However, **there is no vertex that appears in both these destinations to serve as a "pillar".** In other words, $X_3$ cannot add an edge to either $X_1$ or $L$ without causing a noticeable change in the rank structure.
>
> To see this, suppose we add the edge $X_3 \to X_1$ into $\mathcal{H}$. In the original $\mathcal{H}$, the rank $r_{\mathcal{H}}(\{L, X_1\}, \{L, X_3\})$ is deficient, equaling $1$. After adding this edge, that is, in graph $\mathcal{G}$, this rank increases to the full value of $2$. Note that such a rank change cannot be compensated by any row permutation afterwards, since the isomorphic structure of the "children bases" of $\{L, X_3\}$ is fundamentally changed. This can also be verified using our **graphical criterion for determining equivalence (Theorem 2).**
>
>
> We hope the above helps clarify why certain graphs are not equivalent in light of the example.
>
> ---
>
> **We want to thank the reviewer again for all the valuable feedback.**
>
> _All references (name; year) appear in the updated manuscript; we omit re-listing them here._

---

### Official Review · Reviewer_Tajr · 2025-10-27

**Soundness:** 3
**Presentation:** 3
**Contribution:** 3
**Rating:** 8
**Confidence:** 2

**Summary:**

This paper studies a fundamental problem in causal discovery: the distributional equivalence criterian of causal structure from observational data when both latent confounders and cycles are allowed, in linear non-Gaussian models. It provides graphical characterization of equivalence class under arbitrary latent structures and loops via path and edge rank constraints. It also provides graphical transformation rules to traverse within equivalence class. An algorithm glvLiNG is proposed for structure learning from data.

**Strengths:**

- This work is the first equivalence characterization for causal models with latent variables and cycles, providing the foundation needed for future assumption-free latent causal discovery.
- It introduces the edge rank constraints, a new local graphical tool complementing path-rank constraints.
- It provides clean connection between algebraic rank constraints and graph operations.
- The examples provided in Figure 1-3 are helpful for readers to understand the concepts.

**Weaknesses:**

See questions.

**Questions:**

- Do the geometric equivalence results in this paper, such as Lemma 3, 5, 6, generalize beyond LiNG models? It seems they also hold in general models.
- What is formally the faithfulness assumption in Line 440-441 for the proposed algorithm to work?
- Alhough improving on Lemma 3 from paths to edges, it is still hard to check equivalence via Lemma 5, as one needs to verify for all sets of variables $Z,Y$. The condition is is still not as clean as simply stating something like "same skeleton + v-structures".

---

> ### Author Response · Authors · 2025-11-22
> **Response to Reviewer Tajr - Q1 (Part 1/2)**
>
> We sincerely appreciate the reviewer's constructive comments and helpful feedback. Please see below for our point-by-point response.
>
> ---
>
> **(Q1)** The reviewer wonders whether our geometric equivalence results hold in general models.
>
> **A:** We appreciate this thoughtful question. In what follows, let us address it in two parts: whether these results themselves directly hold, and whether they, together with the motivations and techniques behind them, can be generalized.
>
> First, regarding whether the equivalence results themselves directly hold in general models, **the answer is no.**
>  - For example, in linear Gaussian models, our equivalence characterization is still necessary but no longer sufficient. That is, two graphs that are equivalent in linear non‑Gaussian models (this work) are guaranteed to remain equivalent in Gaussian models; however, graphs that are distinguishable under non‑Gaussianity may collapse into the same equivalence class under Gaussianity.
>  - This is expected and aligns with many familiar cases. For instance, with no latents and no cycles, the directed acyclic graph (DAG) can be uniquely identified in linear non‑Gaussian models, while in linear Gaussian models, different DAGs may be equivalent, and thus only their equivalence class, represented by a completed partially directed acyclic graph (CPDAG), is identifiable.
>  - Broadly speaking, this occurs because different model classes permit different types and amounts of structural information that can be identified from data. In the comparison above, the mixing matrix in linear non-Gaussian models can be identified, while in linear Gaussian models, only the covariance matrix is available.
>
>
> Now that the equivalence results themselves do not directly hold in other settings, we come to the second regard: do they suggest potential generalizations? This time, **the answer is yes,** and this is exactly one of our main contributions in introducing edge ranks, a new tool that fills a missing piece in the rank-based picture and thus expands the toolbox for latent-variable causal discovery.
>
> In what follows, we outline how this expanded rank‑based toolbox can support the generalization of our results to different parametric settings.

---

> ### Author Response · Authors · 2025-11-22
> **Response to Reviewer Tajr - Q1 (Part 2/2)**
>
> 1. **For the linear Gaussian setting, existing results in the literature can be directly translated into our edge rank language.**
>     - Unlike the non-Gaussian setting where the mixing matrix is identifiable, in the Gaussian setting, only the covariance matrix is available.
>     - The graphical characterization of covariance matrix ranks, known as "trek-separation," has been established by Sullivant et al., (2010). Specifically, the concept of a bottleneck, which we term the "path rank" on the one-sided directed paths, is extended to the bottleneck along the two-sided directed paths, known as **"treks"**.
>     - However, the condition for when two latent-variable models entail the same covariance ranks is yet to be developed (and known to be difficult). This is where edge ranks might help: since the duality between path ranks and edge ranks hold universally in graphs regardless of the parametric setting, the existing characterization on trek-based path ranks can be directly translated into trek-based edge rank language.
> 	- **This alternative is valuable:** it allows one to work with more local, edge-based graphical constraints and, algebraically, to use simpler binary adjacency matrices rather than weighted covariance matrices.
> 	- As for the technical roadmap, one may first note that the non-Gaussian equivalence condition we build in this work is necessary but not sufficient for the Gaussian setting. That is, two graphs that are equivalent in non‑Gaussian models are guaranteed to remain equivalent in Gaussian models; however, graphs that are distinguishable under non‑Gaussianity may collapse into the same equivalence class under Gaussianity.
> 	- We see the closing of this gap as the most immediate future direction for extending our current work.
>
> 2. **For the discrete setting, our results are likely generalizable as well.**
>     - Several recent works have explored path ranks in graphs from discrete data (Gu & Xu 2023; Chen et al., 2024), where the algebraic counterpart becomes the **tensor ranks in the contingency table.**
>     - However, a precise graphical characterization, analogous to "trek-sepration" in the Gaussian case above, has yet to be developed.
>     - That said, such a characterization is promising, since the linear Gaussian and discrete models behave similarly in many aspects.
>     - For example, both are closed under marginalization and conditionalization; both admit a correspondence between Markov and distributional equivalence, in both cyclic and acyclic cases (Geiger & Meek, 1996; Pearl & Dechter, 1996).
>     - Motivated by these parallels already noted in literature, we believe our results can also extend to the discrete setting, and will be directly applicable once the corresponding graphical characterization is developed.
>
> 3. **For nonlinear or even nonparametric settings, theoretical generalization remains possible.**
>    - When the model is partially linear and partially nonlinear, low-dimensional bottlenecks in the linear component remain directly observable through covariance ranks (Spirtes, 2013).
>    - When the model is fully nonlinear or even nonparametric, there also exists prior results on the identifiability of latent-variable models (Hu, 2008 & 2017). Although the techniques differ, the underlying motivation remains closely related to ranks, particularly those in the **Jacobian matrix.**
>    - However, despite the theoretical meaningfulness, the practical estimation and reliable testing of these ranks remain an open challenge.
>    - This challenge can be echoed by viewing rank constraints as generalizations of conditional independence constraints. In linear models, conditional independencies correspond to low ranks in the covariance matrix and can be directed tested via Fisher's Z test. In contrast, robust conditional independence tests in nonlinear settings are still under active development (Duong & Nguyen, 2024; Yang et al., 2025).
>
>
> In summary, while this work itself focuses on the linear non-Gaussian setting, and the exact equivalence characterization may not hold in other settings, **the tools and intuitions we develop are broadly generalizable.** Just as the original LiNGAM (Shimizu et al., 2006) inspired a range of general causal discovery methods in the fully observed case, including the well-known additive noise models (Hoyer et al., 2008) and post-nonlinear models (Zhang & Hyvärinen, 2010), we hope this work could as well provide some groundwork for latent-variable causal discovery in more general settings.

---

> ### Author Response · Authors · 2025-11-22
> **Response to Reviewer Tajr - Q2 & Q3**
>
> **(Q2)** The reviewer asks for the formal definition of the faithfulness assumption in our algorithm.
>
> **A:** We appreciate the reviewer's careful reading. The formal definition is as follows:
>
> > Let $d_X \coloneqq |X|$ be the number of observed variables. We assume that in the true mixing matrix $A_{X,:}$ that generates data (as defined in Equation (5)), all $d_X \times d_X$ and $(d_X-1) \times (d_X-1)$ minors exhibit matrix ranks consistent with the corresponding path ranks entailed by the true causal graph (as characterized in Lemma 2).
>
> In other words, there is no coincidental parameter cancellation in the data generating process that would lead to matrix ranks lower than those structurally entailed by the graph. Note that such **faithfulness** assumption, often also referred to as the **genericity** assumption, is standard in the literature. It holds almost everywhere in the parameter space except for a Lebesgue measure zero set where coincidental lower ranks occur.
>
> Thanks to your comment, we have provided this formal definition in the updated manuscript. We hope the reviewer finds the revised version clearer in this regard.
>
> ---
>
> **(Q3)** The reviewer raises a concern about our Lemma 5's efficiency in determining equivalence.
>
> We appreciate this helpful point. Indeed, Lemma 5 is **not** meant to serve as an efficient criterion. It is **only an intermediate result.** Our final graphical criterion, which we introduce subsequently, is significantly more efficient. Specifically:
>
> + **Lemma 5 serves as a transitional result,** which directly translates the path ranks from Lemma 3 into the edge rank language. It operates more at the definitional level and is analogous to saying "same d-separations".
> + **The main efficiency improvement comes in the next step,** where with the help of edge ranks, Lemma 5 is further simplified to a more efficient criterion for determining equivalence, in analogous to saying "same adjacencies and v-structures". Importantly,
> + **One only need to check each single observed variable,** instead of checking all subsets of them, for determining whether given graphs are equivalent.
> + In light of your comment, we now present this efficient criterion as a standalone **Theorem 2** in the updated manuscript to make these connections clearer.
>
> We also thank the reviewer for mentioning the helpful analogy to Markov equivalence in fully observed acyclic models. Based on it, we wish to highlight that our results go beyond the above equivalence checking criterion. In fact, **we have provided a comprehensive suite of results,** each targeting a different perspective of the problem, as follows:
>
>   1. **A definitional characterization of equivalence,** like "same d-separations," expressed in dual languages of path and edge ranks (Lemmas 3 and 5);
>   2. **An efficient graphical criterion for determining equivalence,** like "same adjacencies and v-structures," where importantly, one only need to check each single observed variable, instead of all subsets of them (Theorem 2);
>   3. **A transformational characterization** that allows for equivalence class traversal, like "covered edge reversal (Meek conjecture)" (Theorem 3),
>   4. **An interpretable presentation of the equivalence class** that fully captures what is identifiable, like a "CPDAG" (Theorem 4; new result during the discussion phase; for more details, please kindly refer to our response to Reviewer FDU9's Q1), and
>   5. **A constraint-based algorithm (glvLiNG)** for recovering the true causal graph up to its equivalence class from data, like PC (Section 5).
>
> We hope the above helps address the reviewer's concern. We thank the reviewer for this helpful point that allows us to improve clarity.
>
>
> ---
>
> **We would like to thank the reviewer again for all the valuable feedback.**
>
> _All references (name; year) appear in the updated manuscript; we omit re-listing them here._

---

### Official Review · Reviewer_FDU9 · 2025-10-31

**Soundness:** 3
**Presentation:** 3
**Contribution:** 3
**Rating:** 8
**Confidence:** 3

**Summary:**

The paper provides a transformational characterization of observationally equivalent latent linear nonGaussian (possibly cyclyc) directed graphical models, including helpful tools (edge ranks, and an interactive equivalence class explorer) along the way, as well as an ICA-based learning algorithm.
I think this is a solid paper considering the foundation it provides for future work and the interesting mathematics it uses, but it perhaps falls short in terms of potential for direct impact, due to its theoretical focus.

**Strengths:**

- addresses an important open problem
- provides a rigorous, elegant solution
- overall well written, including helpful examples

**Weaknesses:**

The main weakness is the lack of immediate applicability to real problems. While the paper makes great progress on an extremely challenging problem, it remains unclear to me (and unsupported by the experiments) how useful the current implementation actually is. If the authors could provide a real-data application and show meaningful practical interpretation of the (accurately) learned equivalence class, I would change my mind about this.

**Questions:**

The main reason I didn't rate the paper higher is because of limited (direct) potential impact. I think this is inline with how the authors present the work in the paper (e.g., presenting the glvLiNG algorithm as a "proof of concept" rather than a practical tool for causal discovery from real data), but nevertheless answers to the following questions could lead me to increase my rating:
1. Do the authors have any real-data applications in mind, where the learned equivalence classes are sufficiently informative? I'm thinking of how a CPDAG can still be relatively useful in practice, while it's less clear how useful one of these equivalence classes would be.
2. Any ideas for an interpretable representative a given equivalance class (analogous to a CPDAG)?
3. Any ideas for how to refine these to interventional equivalence classes?

The following are suggestions I hope the authors will find useful but that don't require a response and don't affect my rating:
1. Without providing finite sample guarantees, it doesn't really make sense to talk about "identifying" a graph from data---the graph is rather identified from (parameters of) the data-generating distribution. In many but not all instances, the paper correctly uses "recover" (and "estimate" or "learn" would also be appropriate words) for this case, but there are still some places where "identify" is used incorrectly (including L015, L027, L064).
2. L211: "share the same latent variables" seems too strong; can the authors justify the step from the same *number* of latents in the previous sentence to the same latent variables here? or use more precise language here if nothing more is meant.
writing: L238 ("is" -> "are", or "constraints" -> "constraint"), L297 ("to the" -> "in"), L346 ("in edges term" -> "in terms of edge ranks").
3. reference formatting: ensure all words (e.g., 'Bayesian') are capitalized correctly; use the correct characters in Dénes Kőnig's name.
4. Consider adding additional citations, e.g., to work on transformational characterizations for other settings [1], [2], [3], [4], [5], and identifiability in nonparametric latent measurement models (without graphical assumptions beyond acyclicity) [6], [7], [8].
5. Remove the "if not all" part in the second sentence of the abstract. Papers [6], [7], and [8] (and I imagine others) don't make structural assumptions---they just use minimality, which is essentially the author's irreducibility but for identifiability up to nonparametric independence model equivalence rather than up to LiNG distributional equivalence.

[1] Zhang, J., & Spirtes, P. (2005). A transformational characterization of Markov equivalence for directed acyclic graphs with latent variables. In Proceedings of the Twenty-First Conference on Uncertainty in Artificial Intelligence (pp. 667-674).

[2] Chickering, D. M. (1995). A Transformational Characterization of Equivalent Bayesian Network Structures. In UAI (pp. 87-98).

[3] Johnson, J., & Semnani, P. (2025). Characteristic Imsets for Cyclic Linear Causal Models and the Chickering Ideal. arXiv preprint arXiv:2506.13407.

[4] Markham, A., Deligeorgaki, D., Misra, P., & Solus, L. (2022). A transformational characterization of unconditionally equivalent Bayesian Networks. In International Conference on Probabilistic Graphical Models (pp. 109-120). PMLR.

[5] Améndola, C., Boege, T., Hollering, B., & Misra, P. (2025). Structural Identifiability of Graphical Continuous Lyapunov Models. arXiv preprint arXiv:2510.04985.

[6] Markham, A., & Grosse-Wentrup, M. (2020). Measurement dependence inducing latent causal models. In Conference on Uncertainty in Artificial Intelligence (pp. 590-599). PMLR.

[7] Jiang, Y., & Aragam, B. (2023). Learning nonparametric latent causal graphs with unknown interventions. Advances in Neural Information Processing Systems, 36, 60468-60513.

[8] Pearl, J. and Verma, T. (1995). A theory of inferred causation. In Studies in Logic and the Foundations of Mathematics, volume 134, pages 789–811. Elsevier.

---

> ### Author Response · Authors · 2025-11-22
> **Response to Reviewer FDU9 - Q1 (Part 1/3)**
>
> We sincerely appreciate the reviewer's constructive comments and insightful feedback. Please see below for our response.
>
> ---
>
> **(Q1)** The reviewer wonders whether an interpretable presentation of an equivalence class (analogous to a CPDAG) is possible.
>
> **A:** Thank the reviewer for this especially constructive question that helps us deliver new results. **The answer is yes.** In what follows, we first present a preliminary observation that, while not directly used in our presentation, is helpful in its own right. We then formally outline how our presentation is constructed. Finally, we discuss how the results in this work offer a comprehensive view of equivalence that actually goes beyond a transformational criterion.
>
> **I. We begin with a preliminary observation on the identifiability of ancestral relations among observed variables.**
>
> - As a preliminary observation, we first note that the ancestral relations among observed variables $X$ are invariant across all equivalent graphs (this follows from the admissible moves).
> - Therefore, presenting any arbitrary graph in the equivalence class suffices to inform users the true ancestral relations  (causal ordering with cycles) among $X$.
>  - For applications such as experimental design involving observed variables, even this alone, without needing the presentation to be developed next, is already highly informative.
>
>
> **II. Next, we formally outline how our presentation of an equivalence class is constructed.**
>
> Just for clarity in the subsequent results, we first define **"subclasses"** within an entire equivalence class: each "subclass" consists of graphs that can reach each other via sequences of admissible edge additions/deletions, but without cycle reversals. With this notion in place, we now show how to construct **an informative presentation for each subclass,** in two steps:
>
> **Step 1. Constructing the unique maximal graph within the subclass.**
>   - We show that among all equivalent graphs within a subclass, there exists a unique maximal graph such that every other one is a subgraph of it.
>   - We further provide an **explicit construction of this maximal graph (Corollary 1)**, without needing to enumerate all equivalent graphs in the subclass and then take the maximal.
>   - Analogous to the "largest chain graph'" in Frydenberg (1990), this maximal graph serves as the backbone of the subclass presentation, informing users which causal relations are **guaranteed to be absent** in this subclass.
>
> **Step 2. Constructing edges that must appear in all graphs in the subclass.**
>    - Building on Step 1, we further characterize edges that must be present in **every** equivalent graphs within the subclass.
>    - Again, these edges can be determined efficiently via an **explicit graphical condition (Corollary 2)**, without needing subclass traversal or iterative procedures like Meek rules (Meek, 1995).
>    - On the maximal graph from Step 1, i.e., the backbone of the presentation, we highlight these "must" edges using a solid type, while the remaining edges are shown as dashed.
>    - These solid edges, in analogous to the arrows in CPDAGs or "visible edges'' in PAGs (Zhang, 2008b), inform users which causal relations **they can fully trust** in this subclass.
>
>
> With these two steps, we construct a complete presentation for any given subclass. One may then naturally wonder how such a subclass presentation relates to presenting the entire class. Interestingly, we show that a single subclass presentation is sufficiently informative to **present the entire class,** via the following final step:
>
> **Step 3. Presentations for all subclasses are connected via simple cycle reversals.**
>    - We show that the presentation of any subclass can be converted to that of another subclass by a **simple cycle reversal,** with the solid/dashed edge types carried along **(Theorem 4).**
>    - Following the intuition from Step 2, it is now straightforward to see the edges that remain invariant across the entire class: they are simply the solid edges in any subclass presentation that do not belong to or point into any cycles (following the definition of cycler reversals). These edges inform users which causal relations **they can fully trust** throughout the entire equivalence class.
>    - In contrast, the intuition from Step 1, i.e., displaying all possible edges across the entire equivalence class in one graph, is no longer applicable: the union of subclass presentations may be a graph that is not itself equivalent anymore, and thus would introduce more confusion than clarity. For this reason, we do not adopt it.
>    - Finally, since cycle reversals are intuitive and lightweight operations (Lacerda et al., 2008), we simply **present one arbitrary subclass presentation,** with a note about how cycle reversals can generate others, as an interpretable, complete, and compact presentation of the entire equivalence class.

---

> ### Author Response · Authors · 2025-11-22
> **Response to Reviewer FDU9 - Q1 (Part 2/3)**
>
> **In summary**, our presentation of an equivalence class is a single graph with solid and dashed edges, which, together with cycle reversals, fully captures the edges that must or must not appear in every member of the equivalence class.
>
> Examples of such presentations, together with the theoretical results and more discussions, are added to Appendix C.3 of the updated manuscript. We will as well make this presentation available on [our online demo website](https://equiv.cc) soon.
>
> **III. Finally, we discuss how the results in this work offer a comprehensive view of equivalence.**
>
> Having established the above new results on a CPDAG-like presentation, let us now have a reflection on all the results we have for now in characterizing equivalence. In particular, we wish to highlight that our results in fact go beyond a transformational criterion. For context, we first try to broadly categorize common approaches to equivalence into the following four types:
>
> 1. **Structural characterizations,** which provide conditions for determining equivalence between given graphs, and give rise to summary presentations. However, they do not directly lead to equivalence class traversal. They can be further stratified by their complexity, informativeness, or purpose, as follows:
>    + **Level 1.** Conditions necessary and sufficient for determining equivalence, but more as definitions than practical criteria due to high complexities.
>    + **Level 2.** Practical graphical criteria for determining equivalence, which remain necessary and sufficient but are more efficient than Level 1.
>    + **Level 3.** Presentations that summarize the equivalence class. While Level 2 criteria efficiently determine equivalence, they do not completely capture which causal relations are identifiable/invariant across the equivalence class (e.g., extra arrows in a CPDAG apart from v-structures). Level 3 addresses this gap.
> 2. **Transformational characterizations,** which provide natural ways for equivalence class traversal, and are useful for developing score-based algorithms. However, as a complement, they do not themselves provide equivalence determining criteria or summary presentations.
> 3. **Traversal algorithms,** for enumerating, sampling, or counting elements in the equivalence class, where transformational characterizations are typically used.
> 4. **Discovery algorithms,** which learn causal relations directly from data by applying empirical constraints to recover the corresponding equivalence class.
>
> We would like to highlight that our results **contribute to all four categories above.** To illustrate this, we include a side-by-side overview of representative works across two classical settings with ours, as follows:
>
>
> |Categories / Settings|Level|Markov equivalence in fully observed acyclic graphs|Markov equivalence in acyclic graphs with latents|Distributional equivalence in LiNG models with latents and cycles (this work)|
> |-|-|-|-|-|
> |**Structural characterizations**|**Level 1**|"Same d-separations"|"Same d-separations"|"Same path/edge ranks up to permutation" **(Lemmas 3 and 5)**|
> ||**Level 2**| "Same adjacencies and minimal complexes / v-structures" (Frydenberg 1990; Verma & Pearl 1991) | "Same FCI outputs" (Spirtes & Verma 1992); "Same MAG adjacencies, v-structures, and colliders on discriminating paths" (Spirtes & Richardson 1996; Richardson & Spirtes 2002); "Same heads and tails" (Hu & Evans 2020) | "Same children bases for $L$ itself and with each individual $X_i$, up to permutation" **(Theorem 2)** |
> ||**Level 3**| "Maximal chain graphs / essential graphs" (Frydenberg 1990; Andersson et al. 1997; Roverato et al. 2006); "CPDAGs" (Spirtes & Glymour 1991; Meek 1995)|"Arrowhead completeness" (Ali et al. 2005); "Full completeness of PAG orientations" (Zhang 2008a); With background knowledge (still incomplete; Andrews et al. 2020; Venkateswaran & Perković 2024) | "Unique maximal equivalent graph with edges that must always appear, up to cycle reversal" **(Theorem 4)** |
> |**Transformational characterizations**|-| "Covered edge reversal (Meek conjecture)" (Chickering 1995; Meek 1997); "Weakly covered edge reversal" (Markham et al. 2022) for unconditional equivalence|"Covered edge reversal" (Zhang & Spirtes 2005; Tian 2005; Ogarrio et al. 2016; Claassen & Bucur 2022) | "Admissible edge additions/deletions and cycle reversals" **(Theorem 3)** |
> |**Traversal algorithms**|-| DAG traversal within one CPDAG (Meek 1995; Chickering 1995; Wienöbst et al. 2023); CPDAG traversal (Steinsky 2003; Chen et al. 2016) | MAG traversal within one PAG (Wang et al. 2024; Wang et al. 2025) | BFS/DFS by admissible moves **(Theorem 3)**, with additional parallel speedup by column decomposition **(Lemmas 9 and 12)**; online interactive explorer also provided |
> |**Discovery algorithms**|-| PC (Spirtes 1991); GES (Chickering 2002); and many others | FCI (Spirtes 1992); RFCI (Colombo et al. 2012); GFCI (Ogarrio et al. 2016) | glvLiNG **(Section 5)** |

---

> ### Author Response · Authors · 2025-11-22
> **Response to Reviewer FDU9 - Q1 (Part 3/3)**
>
> We hope this table helps better position our contributions within the landscape. There is also a broad range of related work on equivalence in other settings, including the references suggested by the reviewer, which we could not list in this table simply due to space limit. We have **incorporated them all and added further discussion** in the updated manuscript. We have also reorganized the presentation to make these connections clearer.
>
> We sincerely thank the reviewer again, for this constructive question that has inspired the new result on equivalence class presentation to come during the discussion phase.

---

> ### Author Response · Authors · 2025-11-22
> **Response to Reviewer FDU9 - Q2**
>
> **(Q2)** The reviewer suggests to evaluate glvLiNG on real-world settings.
>
> **A:** Thank you for this valuable suggestion. In light of it, we have managed to apply glvLiNG to a **daily stock return dataset** from the Hong Kong stock market. To further assess glvLiNG's empirical utility, we also conducted a simulation study comparing with existing methods. Results from both suggest that despite the potential limitations of the current OICA's implementations, glvLiNG is still able to produce meaningful causal discovery results. We outline this study below.
>
>
>
> For the real-world experiment, we use a Hong Kong stock market dataset that involves the daily dividend/split-adjusted closing prices for 14 major stocks from January 4, 2000 to June 17, 2005 (1331 samples). These 14 stocks represent the dominant sectors of the market:
>
> + 3 of them are on **banking** (HSBC Holdings, Hang Seng Bank, Bank of East Asia),
> + 5 on **real estate** (Cheung Kong, Henderson Land, Hang Lung Properties, Sun Hung Kai Properties, Wharf Holdings),
> + 3 on **utilities** (CLP Holdings, HK & China Gas, HK Electric), and
> + 3 on **commerce** (Hutchison, Swire Pacific 'A, Cathay Pacific Airways).
>
> All of them were constituents of Hang Seng Index (HSI), and they were almost the largest companies of the Hong Kong stock market at the time.
>
> By applying glvLiNG on this dataset, we recovered an equivalence class of causal graphs containing 2 latent variables. The presentation (as developed in response to your Q1) of this equivalence class is shown in **Figure 8 (Appendix D.5)** of the updated manuscript. Here is a summary:
>
> + The class consists of 19,008 causal graphs with 16=14+2 vertices.
> + Among them the numbers of edges range between 29 to 34.
> + In the presentation, there are 20 "solid" edges and 14 "dashed" edges.
>
> This result suggests several interesting observations, as follows:
>
> 1. Large banks seem to be major upstream causes. For example, the two largest banks, HSBC Holdings and Hang Seng Bank, together form a 2-cycle that has 9 children across sectors, but there are no edges into them.
> 2. Real estates, in contrast, seem to be downstream effect receivers. For example, Cheung Kong has 10 parents, but only 1 edge pointing out from it.
> 3. Utilities are heavily involved in cycles. For example, among 17 simple cycles in the graph, CLP Holdings belongs to 11 of them. These cycles are often across sectors as utilities - real estate - commerce - utilities.
> 4. One latent variable seems interpretable. It has one parent HSBC Holdings, and three children (all with solid edges): Cheung Kong, Hutchison, and Swire Pacific 'A'. Among them, Cheung Kong and Hutchison were two core holdings of a same group.
> 5. Stocks under the same sector tend to be connected more closely.
>
>
> Of course, we acknowledge that these interpretations can be post-hoc, especially given that the recovered graph is relatively dense. To this regard, and with the helpful suggestions from other reviewers, we have also conducted a **simulation study** comparing glvLiNG with existing methods including LaHiCaSl (Xie et al., 2024) and PO-LiNGAM (Jin et al., 2024) across settings with varying numbers of observed and latent variables, graph density, and sample size. Full results and details are provided in Appendix D.4 of the updated manuscript; part of them can also be referred to in our response to Reviewer GjKP's Q2. Below, we just highlight some key observations:
>
>
> 1. When the graph is **sparse,** glvLiNG performs slightly worse than LaHiCaSl and PO-LiNGAM. This is perhaps because sparsity and irreducibility together imply more latent‑outgoing edges, which aligns well with these methods' model assumptions. Then, given the fact that both methods estimate ranks using the more efficient generalized independent noise (GIN) condition rather than OICA, it is not surprising to see that they perform better in this setting.
>
> 2. However, when the graph is **denser,** glvLiNG performs particularly better. This is perhaps because, when the graph is denser, more complex structures become common, including arbitrary edges between latent and observed variables, as well as cycles. Model assumptions of existing methods are more likely to be violated, making them less effective at recovering these structures. In contrast, with a structural-assumption-free design, glvLiNG avoids such model misspecification, and still allows the recovery of these structures.
>
> 3. glvLiNG is also **more robust to latent dimensionality.** This is perhaps because glvLiNG jointly recovers all latent-outgoing edges at once, using an OICA mixing matrix whose dimensionality is already fixed. In contrast, the other two methods require incrementally clustering and adding latent variables.
>
>
> More details on the above empirical study are included in Appendices D.4 and D.5 of the updated manuscript.
>
> We thank the reviewer again for helping improve the completeness of our submission.

---

> ### Author Response · Authors · 2025-11-22
> **Response to Reviewer FDU9 - Q3**
>
> **(Q3)** The reviewer asks about possible extensions to interventional equivalence classes.
>
> Thank you for this insightful question. We address it below in three parts:
>
> 1. **For hard interventions, the technical path looks clear:** We conjecture that our edge rank tools can be directly used to identify the causal parents (up to some indeterminacies) of the intervened variable, and thus refine the equivalenc class. Specifically,
>    - Let us define "hard interventions" first. By "hard intervention" on a variable $V_i$, we mean setting $V_i$ to a fixed value or exogenous noise, which graphically corresponds to removing all incoming edges to $V_i$. Let us use $p ^ {(0)}$ to denote the pure observational distribution, and $p ^ {(V_i)}$ the distribution under intervention on $V_i$.
>    - Recent work (Sharifian et al. 2025) has shown that when there are no latent variables, the true causal parents of $V_i$ are identifiable given access to $p ^ {(0)}$ and $p ^ {(V_i)}$, and thereby the equivalence class is refined. Interestingly, as suggested by Reviewer GjKP, this work also approaches from a bipartite matching view.
>    - When there are latent variables, **our edge rank framework extends naturally.** In matroid language, a hard intervention on $V_i$ corresponds to removing it from the transversal family or, dually, adding a sink to its associated gammoid. **Note that this intervention target $V_i$ can also be latent.**
>    - Then, since path/edge ranks remain well-defined both before and after the intervention, comparing $p ^ {(V_i)}$ against $p ^ {(0)}$ should reveal additional observable low-rank constraints. These ranks should help identify the parents of $V_i$ up to some latent-induced indeterminacies, and thereby refine the equivalence class.
>    - We believe the technical path for the above procedure is clear and follows directly from our edge rank framework. In particular, we anticipate that our Lemma 10, which is currently used to recover each observed variable's children in the glvLiNG algorithm, could be a useful starting point.
>
>
> 2. **For soft interventions, the behavior is more subtle:** While the idea of leveraging changes in "bottleneck sizes" is perhaps still natural, a formal characterization of ranks in linear mixed models is to be developed. Specifically,
>    - By "soft intervention" on a variable $V_i$, we mean altering the causal mechanism of $V_i$ from its parents without necessarily removing their effects. In the linear case, this corresponds to changing the linear coefficients into $V_i$ or changing $V_i$'s exogenous noise. Graphically, edges into $V_i$ are not necessarily removed.
>    - For nonparametric causal discovery, soft interventions typically do not pose more difficulties than hard interventions, since these "changes in causal mechanisms" can also be treated as conditional independencies that can be estimated from data (Newey & Powell, 2003).
>    - However, in the linear case, this becomes subtle. For example, $p ^ {(V_i)}$ and $p ^ {(0)}$ would yield the same rank patterns, since indeed the causal graph is not changed, but only parameters are.
>    - One possible solution, we conjecture, is to examine the ranks in the mixed distribution of $p ^ {(V_i)}$ and $p ^ {(0)}$. In such mixed data, the effects into $V_i$ admit a **linear mixed model**, where the nice correspondence between graphical quantities (e.g., path ranks) and algebraic quantities (e.g., matrix ranks) may be broken.
>    - In particular, one should expect additional high-rank patterns from this mixed data. These ranks might help further recover the structure. However, the formal characterization of these ranks in the mixed model is still to be developed.
>
> 3. **Towards practical intervention handling, there are more complications to consider:** Latent variables and cycles can pose additional complications in modeling interventions that go beyond the direct applicability of our current results. Specifically,
>    - Beyond hard/soft distinctions, real-world interventions can raise even more challenges, especially with latent variables and cycles.
>    - For example, in cyclic models, intervening a variable may not only alter the incoming edges to that variable, but also **shift the global equilibrium.** When stability constraints are considered, this complicates how interventions should be modeled (Mokhtarian et al., 2023).
>    - Addressing these issues in our setting likely require new tools, such as parameter identification results with latent variables and cycles.
>    - While these questions remain open, we hope our current edge rank framework and structure identification results can provide some groundwork for future progress.
>
>
> We hope the above points clarify the potential extensions to interventional equivalence classes. We thank the reviewer again for this insightful question.

---

> ### Author Response · Authors · 2025-11-22
> **Response to Reviewer FDU9 - Q4**
>
> **(Q4)** Other valuable comments.
>
> **A:** We sincerely thank the reviewer for the close reading and thoughtful suggestions.
>
> We have corrected the noted typos, expressions, and reference formatting issues. We have also added all the suggested references and additional related ones, and included further discussion in the updated manuscript.
>
> Regarding the phrase **"share the same latent variable labels,"** we simply meant that when the number of latent variables is the same, we may, without loss of generality, assign them the same name labels for notational ease. We have clarified this point in the manuscript.
>
> Regarding the use of **"identify,"** we have carefully taken your suggestion into account. We totally agree with the reviewer that greater care is needed in finite-sample contexts, where we instead use "estimate." There are however some places we have kept the "identify" to refer to asymptotic identifiability from the data generating distribution, as commonly used in the literature (Pearl, 2009). We have carefully checked such usage throughout.
>
> ---
>
> **We want to thank the reviewer again for all the valuable feedback.**
>
>
> _All references (name; year) appear in the updated manuscript; we omit re-listing them here._

---

### Official Review · Reviewer_GjKP · 2025-11-02

**Soundness:** 3
**Presentation:** 3
**Contribution:** 3
**Rating:** 8
**Confidence:** 4

**Summary:**

The paper studies characterization and learning of distributional equivalence classes in linear non-Gaussian models with latent confounding and cycles. It first gives a graphical condition under which a latent variable is not reducible and a reduction procedure to the irreducible form. It then relates the rank of submatrices of the mixing matrix $A$ to path rank (a global property that is costly to check) and introduces a local notion, edge rank, proving a duality between the two. Using edge ranks, the paper provides a graphical characterization of equivalence (Theorem 2) and a way to traverse the class. The learning pipeline runs OICA to estimate columns of $A$ up to scaling/permutation and then applies the graphical operations to recover the equivalence class.

**Strengths:**

- A local graphical notion (edge rank) that makes rank constraints easier to verify.

- A full graphical characterization of distributional equivalence with latent variables and cycles.

- A constructive traversal algorithm for the equivalence class with the admissible moves.

**Weaknesses:**

- As mentioned by the authors, the learning algorithm relies on solving the OICA problem, which, in general, is hard to solve with existing methods. That being said, the result of characterizing the equivalence class has a theoretical contribution regardless of the learning algorithm.

- The evaluation mostly counts class sizes and compares to a MILP baseline for rank-realization under oracle ranks. There is no empirical test in the finite sample case.

**Questions:**

- For linear non-Gaussian cyclic models without latent confounding, (Sharifian et al. 2025) characterized the distributional equivalence class and showed it corresponds to perfect matchings in a bipartite graph. It would be valuable to check whether your paper’s equivalence conditions (via edge rank) reduce to theirs in the no-latent case.

- In the acyclic setting with latent confounding, (Salehkaleybar et al. 2020) gave graphical conditions when a model is irreducible. It would be good to clarify that your irreducibility condition matches the one in there in acyclic models.

- Given only a recovered $A$ from OICA, how should we test the coloop condition in Lemma 7? How can we have a robust solution in the finite sample case?

- What is the computational complexity of the two phases in the learning algorithm?

Sharifian, E., Salehkaleybar, S., & Kiyavash, N. (2025). Near-Optimal Experiment Design in Linear Non-Gaussian Cyclic Models. arXiv:2509.21423.

Salehkaleybar, S., Ghassami, A., Kiyavash, N., & Zhang, K. (2020). Learning Linear Non-Gaussian Causal Models in the Presence of Latent Variables. Journal of Machine Learning Research, 21(39): 1–24.

---

> ### Author Response · Authors · 2025-11-22
> **Response to Reviewer GjKP - Q1**
>
> We sincerely appreciate the reviewer's constructive comments and helpful feedback. Please see below for our response.
>
> ---
>
> **(Q1)** The reviewer has concerns about the use of OICA in our learning algorithm.
>
> **A:** Thank you for raising this important practical issue. Indeed, we do see this as a major practical limitation of our glvLiNG algorithm, as noted in Section 5. At the same time, we would also like to mention the following observations, which have been included in the updated manuscript:
>
> 1. **The glvLiNG algorithm is still usable in practice.** In light of your suggestion, we have added a simulation study. We find that despite the potential limitations of the current OICA's implementations, glvLiNG still outperforms existing methods, perhaps because it avoids model misspecification by design. Interestingly, glvLiNG also runs faster than these methods. In addition, we also apply glvLiNG to a real-world dataset and obtain meaningful results. Please kindly refer to our response to your Q2 for details on these.
> 2. **There are promising directions for improvement.** Note that we do not need full parameters of the mixing matrix, but only its rank structure. For this purpose, several existing techniques can already allow partial access to rank information without explicitly solving OICA, such as cumulant-based methods (Robeva & Seby, 2021), independent subspace analyses (ISA; Dai et al 2022), and generalized independent noise condition (GIN; Xie et al., 2024). It would be interesting to see how these can be incorporated into glvLiNG in future work.
>
> We hope these observations help ease the reviewer's concern. We also appreciate the reviewer's recognition of our main contribution in the theoretical characterization of equivalence, "regardless of the learning algorithm". For glvLiNG, it is intended more as a first baseline to show that latent-variable causal discovery is indeed possible without any structural assumptions.
>
> Finally, we wish to highlight that our equivalence characterization in fact goes beyond a traversal criterion. In analogy with Markov equivalence in fully observed acyclic models, **we have provided a comprehensive suite of results,** each targeting a different perspective of the problem, as follows:
>
> 1. **A definitional characterization of equivalence,** like "same d-separations," expressed in dual languages of path and edge ranks (Lemmas 3 and 5);
> 2. **An efficient graphical criterion for determining equivalence,** like "same adjacencies and v-structures," where importantly, one only need to check each single observed variable, instead of all subsets of them (Theorem 2);
> 3. **A transformational characterization** that allows for equivalence class traversal, like "covered edge reversal (Meek conjecture)" (Theorem 3), and
> 4. **An interpretable presentation of the equivalence class** that fully captures what is identifiable, like a "CPDAG" (Theorem 4; new result during the discussion phase; for more details, please kindly refer to our response to Reviewer FDU9's Q1).
>
> We have reorganized the presentation in the updated manuscript to better show these results. For illustration, we provide a side-by-side overview (Table 2 in Appendix C.5) of representative works across various settings/approaches with ours, to better position our contributions and make the connections to literature clearer.

---

> ### Author Response · Authors · 2025-11-22
> **Response to Reviewer GjKP - Q2 (Part 1/2)**
>
> **(Q2)** The reviewer suggests more empirical validation in the finite sample case.
>
> **A:** Thank you for this constructive suggestion. In light of it, we have managed to add a new empirical study to evaluate glvLiNG's performance in the finite sample case. We provide an overview in what follows.
>
> **I. Let us first describe how we handle the OICA estimation part:**
>   - For our choice of OICA implementation, we have tried multiple options and find that overall, the [MATLAB implementation](https://github.com/gilgarmish/oica) of SDP-ICA (Podosinnikova et al., 2020) tends to provide best estimated mixing matrices across multiple settings. We thus adopt it in our experiments.
>   - For the number of latent variables, although theoretically identifiable, existing OICA implementations still require specifying this number as an input. Hence, following the common practice, we test multiple candidate values and select the one minimizing the loss.
>
> **II. Next, we summarize the simulation setup and key results:**
>
> In simulation, we compare glvLiNG with existing methods including LaHiCaSl (Xie et al., 2024) and PO-LiNGAM (Jin et al., 2024). We generate random Erdős–Rényi irreducible graphs and vary the total number of variables $n$ from $5$ to $13$, number of latent variables $\ell$ from $1$ to $5$, average in-degree $d$ of $1$ and $3$, and sample size $N$ from $1,000$ to $200,000$. Full results and details are provided in Appendix D.4 of the updated manuscript.
>
> Below, we report the structural hamming distance (SHD; in this setting, we use the lowest SHD between the algorithm's output and all graphs in the true equivalence class) results for $N=10,000$, and discuss the effect of sample size.
>
> + SHDs (lower is better) when the average in-degree $d=1$:
>
>   |$n$|5|7||9||11|||13|||
>   |-|-|-|-|-|-|-|-|-|-|-|-|
>   |$\ell$|**1**|**1**|**3**|**1**|**3**|**1**|**3**|**5**|**1**|**3**|**5**|
>    |**glvLiNG**|6.6±1.5|10.5±3.4|8.3±4.0|15.7±6.4|12.7±5.0|19.6±8.3|17.7±1.0|17.9±2.2|25.6±8.2|22.9±1.2|26.3±2.4|
>   |**LaHiCaSl**|**4.4±0.9**|**8.5±1.2**|**6.9±1.9**|12.8±1.7|**10.4±2.2**|**16.1±1.8**|**15.4±2.1**|**14.5±1.6**|20.8±3.2|**19.8±1.4**|**19.7±0.8**|
>   |**PO-LiNGAM**|5.1±1.5|9.6±2.1|10.1±1.8|**12.6±2.3**|14.8±1.7|17.2±3.4|19.1±2.2|20.6±1.8|**19.7±3.6**|23.2±3.4|26.1±1.7|
>
> + SHDs (lower is better) when the average in-degree $d=3$:
>
>   |$n$|5|7||9||11|||13|||
>   |-|-|-|-|-|-|-|-|-|-|-|-|
>   |$\ell$|**1**|**1**|**3**|**1**|**3**|**1**|**3**|**5**|**1**|**3**|**5**|
>   |**glvLiNG**|**2.6±1.4**|**9.9±2.1**|**9.8±2.1**|**17.8±1.9**|**17.8±2.5**|**26.0±4.7**|**26.6±3.6**|**26.2±2.7**|**33.1±5.3**|**34.9±3.5**|**35.7±3.4**|
>   |**LaHiCaSl**|14.1±1.2|21.2±1.8|20.7±2.0|28.7±0.5|28.2±2.9|35.8±0.5|36.2±0.9|36.6±3.7|42.0±0.7|44.0±1.3|46.2±1.7|
>   |**PO-LiNGAM**|10.6±1.6|17.1±1.3|17.9±1.7|23.0±1.9|26.4±2.4|28.5±1.3|34.6±2.3|40.8±4.0|35.3±1.8|42.3±3.4|50.7±2.9|
>
> The results suggest the following observations:
>
> 1. **First of all, it is not surprising to see that LaHiCaSl and PO-LiNGAM perform better when the graph is sparser.**
>    - For example, when $d=1$, these two methods perform better than glvLiNG, though the difference remains modest.
>    - This is perhaps because, when the graph is sparser, maintaining irreducibility typically means more edges outgoing from latent variables, while edges from observed ones to others are fewer.
>    - This aligns well with the model assumptions of these two methods. For example, LaHiCaSl assumes a hierarchical latent-variable model in which all observed variables are leaf nodes.
>    - Given this additional benefit from their sparsity constraints, and the fact that both LaHiCaSl and PO-LiNGAM estimates ranks using the GIN condition which is more efficient than OICA, it is not surprising to see that they perform better in this setting.
> 2. **However, when the graph is denser, glvLiNG performs particularly better.**
>    - For example, when $d=3$, glvLiNG consistently outperforms the other two methods, and the difference is considerable.
>    - This is perhaps because, when the graph is denser, more complex structures become common, including arbitrary edges between latent and observed variables, as well as cycles.
>    - Model assumptions of existing methods are more likely to be violated, making them less effective at recovering these structures.
>    - In contrast, with a structural-assumption-free design, glvLiNG avoids such model misspecification, and still allows the recovery of these structures.
> 3. **glvLiNG is also more robust to latent dimensionality.**
>    - For example, when $n=13$ and $d=3$, increasing the number of latents $\ell$ from $1$ to $5$, the average SHD of glvLiNG increases from 33.1 to 35.7, while other method, such as PO-LiNGAM, increases from 35.3 to 50.7.
>    - This is perhaps because glvLiNG jointly recovers all latent-outgoing edges at once, using an OICA mixing matrix whose dimensionality is already fixed.
>    - In contrast, two other methods require incrementally clustering and adding latent variables.

---

> ### Author Response · Authors · 2025-11-22
> **Response to Reviewer GjKP - Q2 (Part 2/2)**
>
> Certainly, we fully acknowledge that a sample size of $N=10,000$ is **relatively large,** and as expected given OICA's sensitivity to sample size, glvLiNG performs much worse with smaller sample sizes such as $N=1,000$. That said, we still see an overall trend similar to the observations above. **Full results across different sample sizes** are provided in Figure 7 of Appendix D.4 in the updated manuscript.
>
> **III. We then demonstrate results on a real-world dataset:**
>
> For the real-world experiment, we use a Hong Kong stock market dataset that involves the daily dividend/split-adjusted closing prices for 14 major stocks from January 4, 2000 to June 17, 2005 (1331 samples). These 14 stocks represent the dominant sectors of the market:
>
> + 3 of them are on **banking** (HSBC Holdings, Hang Seng Bank, Bank of East Asia),
> + 5 on **real estate** (Cheung Kong, Henderson Land, Hang Lung Properties, Sun Hung Kai Properties, Wharf Holdings),
> + 3 on **utilities** (CLP Holdings, HK & China Gas, HK Electric), and
> + 3 on **commerce** (Hutchison, Swire Pacific 'A, Cathay Pacific Airways).
>
> All of them were constituents of Hang Seng Index (HSI), and they were almost the largest companies of the Hong Kong stock market at the time.
>
> By applying glvLiNG on this dataset, we recovered an equivalence class of causal graphs containing 2 latent variables. The CPDAG-like presentation (new result; see Theorem 4 in the updated manuscript) of this equivalence class is shown in **Figure 8 (Appendix D.5)**. Here is a summary:
>
> + The class consists of 19,008 causal graphs with 16=14+2 vertices, and among them the numbers of edges range between 29 to 34.
> + In the presentation, there are 20 "solid" edges, i.e., edges that must appear in all equivalent graphs, and 14 "dashed" edges, i.e., edges that appear at least in one equivalent graph.
>
> This result suggests several interesting observations, as follows:
>
> 1. Large banks seem to be major upstream causes. For example, the two largest banks, HSBC Holdings and Hang Seng Bank, together form a 2-cycle that has 9 children across sectors, but there are no edges into them.
> 2. Real estates, in contrast, seem to be downstream effect receivers. For example, Cheung Kong has 10 parents, but only 1 edge pointing out from it.
> 3. Utilities are heavily involved in cycles. For example, among 17 simple cycles in the graph, CLP Holdings belongs to 11 of them. These cycles are often across sectors as utilities - real estate - commerce - utilities.
> 4. One latent variable seems interpretable. It has one parent HSBC Holdings, and three children (all with solid edges): Cheung Kong, Hutchison, and Swire Pacific 'A'. Among them, Cheung Kong and Hutchison were two core holdings of a same group.
> 5. Stocks under the same sector tend to be connected more closely.
>
> More details on the above empirical study are included in Appendices D.4 and D.5 of the updated manuscript.
>
> We thank the reviewer again for helping improve the completeness of our submission.

---

> ### Author Response · Authors · 2025-11-22
> **Response to Reviewer GjKP - Q3**
>
> **(Q3)** The reviewer asks for more details about the glvLiNG algorithm. Specifically,
>
> **(Q3.1)** How should we test the coloop condition in Lemma 7 given only an OICA recovered $A$ matrix?
>
> **A:** The coloop condition is **not used** during the algorithm. Specifically,
>   - The coloop condition serves only to characterize how one graph can be equivalently transformed to others.
>   - In the algorithm itself, we recover a single graph (which is also shown to be the maximal one) that satisfies the rank structure in $A$. This recovery relies on checking matrix ranks of $A$ with **another condition** (Lemma 10 in Appendix A).
>   - Once this single maximal graph is obtained, the entire equivalence class can then be traversed using the coloop condition.
>   - Thanks to your comment, we have clarified this in Section 5 in the updated manuscript.
>
>
> **(Q3.2)** How can we have a robust solution in the finite sample case?
>
> **A:** Thank you for this insightful point. Below we outline how we handle it in each step of the algorithm.
>   - For the first step involving OICA, there is nothing much we could do in this work. We adopt a most reliable OICA implementation that we could find in experiments, as described in our response to your Q2.
>   - Having obtained an OICA-estimated mixing matrix $A$, the core task of glvLiNG can then be formulated as constructing a bipartite graph to realize the rank patterns in $A$, which define a transversal matroid.
>   - Under finite samples, empirical ranks may **violate matroid axioms,** just like how conditional independencies in data may violate a "graphoid" in nonparametric settings.
>   - To address this, in our implementation, we assign a "full-rank confidence score" to each relevant block of $A$. Then,
>     - In phase 1 (recovering latent outgoing edges), we **approximate the closest valid** transversal matroid that maximizes agreement with these scores.
>     - In phase 2 (recovering observed outgoing edges), we simply threshold these scores to determine each $X_i$'s outgoing edges independently. This is mainly for efficiency consideration, just like comparing PC to exact search.
>   - We have simulated and verified that this procedure is robust to moderately noisy ranks, e.g., by assigning true full-rank blocks scores from $\mathcal{N}(0.75, 0.2)$ and others from $\mathcal{N}(0.25, 0.2)$, both $0,1$ truncated.
>   - We however do not provide formal finite sample guarantees for this procedure, since this work mainly focuses on the asymptotic case, and for clarity, it might be better to separate such analyses.
>
> **(Q3.3)** What is the computational complexity of the two phases in the learning algorithm?
>
> **A:** The overall complexity is **combinatorial.** Specifically, let $x$ and $v$ be the numbers of observed and total variables, so the OICA outputs an $x \times v$ matrix $A$. Then,
>   - The two phases involve checking ranks of certain minors of $A$:
>     - Phase 1 requires checking all $x\times x$ minors of $A$, totaling $v \choose x$ checks.
>     - Phase 2 requires checking all $(x-1)\times (x-1)$ minors of $A$, totaling $x {v \choose {x-1}}$ checks.
>   - Though we parallelize these checks in practice, and glvLiNG is empirically faster than the other two methods in our simulation, **we fully acknowledge this high complexity.** This is because the algorithm is based on matroids, where even the size to describe one matroid (e.g., by its set of bases) is already combinatorial.
>   - That said, we see meaningful room for future improvements. For instance, the admissible moves (Theorem 3) imply that the ancestral relations among observed variables are identifiable, which may help further prune the algorithm.

---

> ### Author Response · Authors · 2025-11-22
> **Response to Reviewer GjKP - Q4**
>
> **(Q4)** The reviewer suggests clarifying how our conditions connect to existing ones in special cases.
>
> **A:** We thank the reviewer for this valuable suggestion. We have checked that our irreducibility condition matches the "absorbability" condition given by (Salehkaleybar et al., 2020) in acyclic settings, and that our equivalence condition matches the one given by (Sharifian et al. 2025) in no-latent no-intervention settings. **We have highlighted our connections to both in the updated manuscript (lines 175 and 393).**
>
> In particular, we are grateful to the reviewer for pointing us to (Sharifian et al. 2025). It is exciting to see that linear non-Gaussian cyclic models have also recently been interpreted from a bipartite matching view, though with a different purpose of modeling interventions. We believe that by combining our edge rank tools and their interventional equivalence results, there may be promising opportunities for further progress in **latent-variable causal discovery with interventions.** Specifically,
>   - Their result of "each such experiment ... revealing the causal parents of the intervened variable," when translated into matroid language, corresponds to removing the intervened variable from the transversal family or, dually, adding a sink to its associated gammoid.
>   - Then, using our edge rank tools, this translation suggests that in the presence of latent variables (and even when latent variables themselves are intervened on), the parents of the intervened variable are still likely to be identifiable, up to some latent-induced indeterminacies.
>   - We hypothesize that the technical path for this is clear, and in particular, we anticipate that Lemma 10, which we currently use to recover each observed variable's children from data, could be a useful starting point.
>
> ---
>
> **We want to thank the reviewer again for all the valuable feedback.**
>
> _All references (name; year) appear in the updated manuscript; we omit re-listing them here._

---

### Author Response · Authors · 2025-12-01
**TL;DR Summary for the Area Chair**

Dear Area Chair and Reviewers,

We sincerely thank you for your extra time and efforts during the unusual circumstances. Although further discussion is unfortunately not possible, we hope the following summary provides a fair overview of the current discussion, and may assist in Area Chair's assessment.

---
We first summarize our main contributions. Drawing an analogy to Markov equivalence in fully observed acyclic models, we wish to highlight that this submission provides **a comprehensive suite of results,** each addressing a different aspect of distributional equivalence in linear non-Gaussian latent-variable cyclic models, as follows:

1. A clean connection between algebra in data and geometry in graphs, using edge ranks, like "d-separations imply conditional independencies" (Theorem 1; Lemma 4);
2. An efficient graphical criterion for determining equivalence, like "same adjacencies and v-structures" (Theorem 2);
3. A transformational characterization that allows for equivalence class traversal, like "covered edge reversal (Meek conjecture)" (Theorem 3);
4. An informative presentation of the equivalence class that fully captures what is identifiable, like a "CPDAG" (Theorem 4); and
5. A constraint-based algorithm (glvLiNG) for recovering the true causal graph up to its equivalence class from data, like PC (Section 5).

We are grateful that reviewers recognized our work as, for example:
- "a solid paper considering the foundation it provides for future work,"
- "addresses an important open problem; provides a rigorous, elegant solution,"
- "makes great progress on an extremely challenging problem,"
- "a very useful tool for future results," and
- "providing the foundation needed for future assumption-free latent causal discovery."

---
We then provide a brief overview of the reviewers' main concerns/questions and our responses.

**_Reviewer GjKP:_** The reviewer has concerns about the use of OICA in the glvLiNG algorithm, and suggests more empirical validation in the finite sample case.

**_Response summary:_**

- Following the reviewer's suggestion, we added a new simulation study comparing glvLiNG to existing methods across different settings.
- We find that despite the potential limitations of the current OICA's implementations, glvLiNG still outperforms existing methods, perhaps because it avoids model misspecification by design.
- We also applied glvLiNG to a real-world stock return dataset and obtained interesting results, including meaningful causal relations among stocks and interpretable latent variables.

---
**_Reviewer FDU9:_** The reviewer wonders whether a CPDAG-like presentation of an equivalence class is possible, and suggests providing a real-data application to demonstrate glvLiNG's direct impact. (As a side note, the reviewer mentions that "answers to the following questions could lead me to increase my rating.")

**_Response summary:_**

- Thanks to the reviewer's suggestion, we developed the new result of a CPDAG-like informative presentation for equivalence classes.
- This presentation is a single graph with solid and dashed edges, which fully capture edges that must or must not appear in every member of the equivalence class.
- We also developed graphical conditions to efficiently construct this presentation from any graph, without needing to traverse the entire equivalence class.
- For real-data application, we applied glvLiNG to a real-world daily stock return dataset and obtained interesting results.

---
**_Reviewer Tajr:_** The reviewer wonders whether our graphical equivalence results hold in general models, and raises a concern about our Lemma 5's efficiency in determining equivalence.

**_Response summary:_**

- We clarified that while the graphical results themselves do not directly hold in general models, the underlying rank-based tools may help extend the framework to many other settings (e.g., Gaussian, discrete, or nonlinear models).
- We noted that Lemma 5 is only an intermediate result. Our final graphical criterion (Theorem 2) is significantly more efficient.

---
**_Reviewer Czvx:_** The reviewer wonders how the glvLiNG algorithm works when we do not have an OICA oracle, and suggests also considering "minimally cyclic" graphs, even with potentially more latent variables.

**_Response summary:_**

- We elaborated on how we handle the OICA part and determine the number of latents in practice, and included results from both simulation and real data to demonstrate glvLiNG's empirical practicability.
- We noted that in the linear non-Gaussian setting, such a trade-off does not arise: irreducibility minimizes the number of latents, and also cycles and edges at the same time. That said, we appreciate the reviewer's comment as this subtlety becomes relevant in other settings.

---
There are also many other insightful comments, which we address in detail in our full responses.

We thank the Area Chair and all reviewers again for all the efforts and valuable feedback.

---

### Meta-Review · Area_Chair_uB5v · 2026-01-05

**Summary:**

This is an excellent theoretical contribution to causal discovery, and given the reviewers' consistent, unanimous, and positive evaluation, I recommend acceptance without hesitation. During the rebuttal, the authors clearly acknowledged all bottlenecks, provided sufficient clarification, and further strengthened the paper's theoretical contribution by presenting additional results. If any change had occurred, the reviewers' rating would've only increased. Below, I summarize the key technical points raised by the reviewers and the authors' responses.

**Reviewer Concerns:**

Across all reviews, the main technical questions were focused on the practicality of the framework. Some reviewers have raised questions about the glvLiNG algorithm relying on OICA, noting its complexity, sensitivity to sample size, unknown latent dimensionality, and lack of finite-sample guarantees. Also, some reviewers noted the empirical evaluation was limited. Finally, some reviewers have asked for clearer algorithmic exposition and a more intuitive equivalence criterion (analogous to “skeleton + v-structures”), as well as an interpretable representation of large equivalence classes (CPDAG-like). In their responses, the authors have acknowledged the practical limitations of OICA and finite-sample robustness. Furthermore, they have strengthened the paper's contribution by adding new finite-sample evaluations and comparisons with existing methods, demonstrating interpretable results.

With these additions and the clarifications on various technical points raised, I believe the reviewers' key concerns have been acknowledged and addressed.

**Reviewer Scores:**

I believe all the reviewers would have likely maintained their scores.

---

### Decision · Program_Chairs · 2026-01-26

Accept (Oral)